# Personalized Decentralized Bilevel Optimization over Stochastic and Directed Networks

## Abstract

While personalization in distributed learning has been extensively studied, existing approaches employ dedicated algorithms to optimize their specific type of parameters (e.g., client clusters or model interpolation weights), making it difficult to simultaneously optimize different types of parameters to yield better performance. Moreover, their algorithms require centralized or static undirected communication networks, which can be vulnerable to center-point failures or deadlocks. This study proposes optimizing various types of parameters using a single algorithm that runs on more practical communication environments. First, we propose a gradient-based bilevel optimization that reduces most personalization approaches to the optimization of client-wise hyperparameters. Second, we propose a decentralized algorithm to estimate gradients with respect to the hyperparameters, which can run even on stochastic and directed communication networks. Our empirical results demonstrated that the gradient-based bilevel optimization enabled combining existing personalization approaches which led to state-of-the-art performance, confirming it can perform on multiple simulated communication environments including a stochastic and directed network.

## 1 Introduction

In distributed learning, providing personally tuned models to clients, or personalization, has shown to be effective when the clients' data are heterogeneously distributed (Tan et al., 2022).

While various approaches have been proposed, they are dedicated to optimizing specific types of parameters for personalization. A typical example is clustering-based personalization (Sattler et al., 2020), which employs similarity-based clustering specifically for seeking client clusters. Another approach called model interpolation (Mansour et al., 2020; Deng et al., 2020) also specializes in optimizing interpolation weights between local and global models. These dedicated algorithms prevent developers from combining different personalization methods to achieve better performance.

Another limitation of previous personalization algorithms is that they can run only on centralized or static undirected networks. Most approaches for federated learning (Smith et al., 2017; Sattler et al., 2020; Jiang et al., 2019) require centralized settings in which a host server can communicate with any client. Although a few studies (Lu et al., 2022; Marfoq et al., 2021) consider fully-decentralized settings, they assume that the communication edge between any clients is static and undirected (i.e., synchronized). These commutation networks are known to be vulnerable to practical issues, such as bottlenecks or central point failures on the host servers (Assran et al., 2019), or failing nodes and deadlocks on the static undirected networks (Tsianos et al., 2012).

This study proposes optimizing various parameters for personalization using a single algorithm while allowing more practical communication environments. First, we propose a gradient-based *Personalized Decentralized Bilevel Optimization (PDBO)*, which reduces many personalization approaches to the optimization of hyperparameters possessed by each client. Second, we propose *Hyper-gradient Push (HGP)* that allows any client to solve PDBO by estimating the gradient with respect to its hyperparameters (*hyper-gradient*) via stochastic and directed communications, that are immune to the practical problems of centralized or static undirected communications (Assran et al., 2019). We also introduce a variance-reduced HGP to avoid estimation variance, which is particularly effective when communications are stochastic, providing its theoretical error bound.

We empirically demonstrated that the generality of our gradient-based PDBO enabled combining existing personalization approaches which led to state-of-the-art performance in a distributed classification task. We also demonstrated that the gradient-based PDBO succeeded in the personalization on multiple simulated communication environments including a stochastic and directed network.

Our contributions are summarized as follows:

- We propose a gradient-based PDBO that can solve existing personalization problems and their combinations as its special cases.
- We propose a decentralized hyper-gradient estimation algorithm called HGP which can run even on stochastic and directed networks. We also propose a variance-reduced HGP, which is particularly effective in stochastic communications, and provide its theoretical error bound.
- We empirically validated the advantages of the gradient-based PDBO with HGP; it enabled solving a combination of different personalization problems which led to state-of-the-art performance, and it performed on different communication environments including a stochastic directed network.

**Notation** $\langle \boldsymbol{A} \rangle_{ij}$ denotes the matrix at the $i$-th row and $j$-th column block of the matrix $\boldsymbol{A}$, and $\langle \boldsymbol{a} \rangle_i$ denotes the $i$-th block vector of the vector $\boldsymbol{a}$. For a function $f : \mathbb{R}^{d_1} \mapsto \mathbb{R}^{d_2}$, we denote its total and partial derivatives with respect to a vector $\boldsymbol{x} \in \mathbb{R}^{d_1}$ by $\mathrm{d}_{\boldsymbol{x}} f(\boldsymbol{x}) \in \mathbb{R}^{d_1 \times d_2}$ and $\partial_{\boldsymbol{x}} f(\boldsymbol{x}) \in \mathbb{R}^{d_1 \times d_2}$, respectively. We denote the product of matrices by $\prod_{s=0}^{m} \hat{\boldsymbol{A}}^{(s)} = \hat{\boldsymbol{A}}^{(m)} \cdots \hat{\boldsymbol{A}}^{(0)}$ and $\prod_{s=0}^{-1} \hat{\boldsymbol{A}}^{(s)} = \boldsymbol{I}$.

## 2 PRELIMINARIES

We formulate distributed learning (Li et al., 2014), communication networks, and stochastic gradient push (Nedić & Olshevsky, 2016, SGP) as a generalization of gradient-based distributed learning.

**Distributed learning** Distributed learning with $n$ clients is commonly formulated for all $i \in [n]$ as

$$\boldsymbol{x}_i^* = \arg\min_{\boldsymbol{x}_i} \frac{1}{n} \sum_{k \in [n]} \mathbb{E}_{\xi_k} \left[ f_k \left( \boldsymbol{x}_k, \boldsymbol{\lambda}_k; \xi_k \right) \right], \quad \text{s.t. } \boldsymbol{x}_i = \boldsymbol{x}_j, \ \forall j \in [n], \tag{1}$$

where, the $i$-th client pursues the optimal parameter $\boldsymbol{x}_i^* \in \mathbb{R}^{d_{\boldsymbol{x}}}$, that makes *consensus* ($\boldsymbol{x}_i = \boldsymbol{x}_j, \forall j \in [n]$) over all the clients, while minimizing its cost $f_i : \mathbb{R}^{d_{\boldsymbol{x}}} \times \mathbb{R}^{d_{\boldsymbol{\lambda}}} \mapsto \mathbb{R}$ for the input $\xi_i \in \mathcal{X}$ sampled from its local data distribution. We allow $f_i$ to take the hyperparameters $\boldsymbol{\lambda}_i \in \mathbb{R}^{d_{\boldsymbol{\lambda}}}$ as its argument. We further explain the examples of the choice of $\boldsymbol{\lambda}_i$ in Sections 3 and 5.

**Stochastic and directed communication network** In distributed learning, clients solve Eq. (1) by exchanging messages over a physical communication network. The type of edge connections categorizes the communication network: *static undirected* (Lian et al., 2017), which represents synchronization over all clients; *stochastic undirected* (Lian et al., 2018), which represents asynchronicity between different client pairs; and *stochastic directed* (Nedić & Olshevsky, 2016), which represents push communication where any message passing can be unidirectional.

This study considers distributed learning on stochastic and directed communication networks. Such a network has several desirable properties: robustness to failing clients and deadlocks (Tsianos et al., 2012), immunity to central failures, and small communication overhead (Assran et al., 2019). We model stochastic directed networks by letting communication edges be randomly realized, as simulated in Assran et al. (2019) and Nedić & Olshevsky (2016). Let $\delta_{i \to j}^{(t)} \in \{0, 1\}$ be a random variable where $\delta_{i \to j}^{(t)} = 1$ denotes that there is a communication channel from the $i$-th client to the $j$-th client at the time step $t$, and $\delta_{i \to j}^{(t)} = 0$ otherwise. We set $\delta_{i \to i}^{(t)} = 1$ for all $i \in [n]$ and $t \in \mathbb{N}$ allowing every client to send a message to itself at any time step. Note that the edge model above can recover the other fully-decentralized settings as its special cases; the symmetric edges ($\delta_{i \to j}^{(t)} = \delta_{j \to i}^{(t)}, \forall i, j \in [n], \forall t \in \mathbb{N}$) recover stochastic undirected networks, and the symmetric constant edges, which additionally require $\delta_{i \to j}^{(t)} = \delta_{j \to i}^{(t)} = \delta_{ij}$, recover static and undirected networks.

**Stochastic gradient push (SGP)** SGP (Nedić & Olshevsky, 2016) is one of the most general solvers of Eq. (1). This section formulates SGP with further generalization for its variants.

The $i$-th client in SGP updates its weight $\omega_i \in \mathbb{R}$ along with biased parameter $\boldsymbol{z}_i \in \mathbb{R}^{d_x}$ to obtain its debiased parameter $\boldsymbol{x}_i = \boldsymbol{z}_i/\omega_i$. Let $\boldsymbol{y}_i = [\boldsymbol{z}_i^\top \ \omega_i]^\top \in \mathbb{R}^{d_y}$ be a concatenated vector. At the $t$-th step, the $i$-th client samples its minibatch $\zeta_i^{(t)}$ and sending edges $\delta_i^{(t)} = \{\delta_{i\rightarrow 1}^{(t)} \ \cdots \ \delta_{i\rightarrow n}^{(t)}\}$, runs a local update $\boldsymbol{\psi}_i : \mathbb{R}^{d_y} \mapsto \mathbb{R}^{d_y}$ and message generator $\boldsymbol{\varphi}_i : \mathbb{R}^{d_y} \mapsto \mathbb{R}^{d_y}$, and updates $\boldsymbol{y}_i$ as

$$\boldsymbol{y}_i^{(t+1)} = \sum_{j \in [n]} p_{ji}(\delta_j^{(t)}) \boldsymbol{\varphi}_j\left(\boldsymbol{y}_j^{(t)}; \boldsymbol{\lambda}_j, \zeta_j^{(t)}\right) + \boldsymbol{\psi}_i\left(\boldsymbol{y}_i^{(t)}; \boldsymbol{\lambda}_i, \zeta_i^{(t)}\right), \tag{2}$$

$$\text{s.t.} \quad \sum_{k \in [n]} p_{ik}(\delta_i^{(t)}) = 1 \quad \text{and} \quad p_{ik}(\delta_i^{(t)}) = \delta_{i\rightarrow k}^{(t)} p_{ik}(\delta_i^{(t)}), \forall k \in [n], \tag{3}$$

where, $p_{ji} : \{0,1\}^n \mapsto [0,1]$ is a weight function that forms column stochastic matrix $\boldsymbol{P}^{(t)}$ such that $P_{ij}^{(t)} = p_{ji}(\delta_j^{(t)})$ to ensure the convergence of $\boldsymbol{x}_i$ to the consensus. Denoting the learning rate by $\alpha_i \in \mathbb{R}^+$, the following formulations of $\boldsymbol{\varphi}_i$ and $\boldsymbol{\psi}_i$ recover the two SGP variants:

$$\begin{cases} \boldsymbol{\varphi}_i\left(\boldsymbol{y}_i; \boldsymbol{\lambda}_i, \zeta_i\right) = \left[\boldsymbol{z}_i^\top - \frac{\alpha_i}{|\zeta_i|}\sum_{\xi \in \zeta_i} \partial_{\boldsymbol{x}_i} f_i\left(\frac{\boldsymbol{z}_i}{\omega_i}, \boldsymbol{\lambda}_i; \xi\right)^\top \ \omega_i\right]^\top, \ \boldsymbol{\psi}_i\left(\boldsymbol{y}_i; \boldsymbol{\lambda}_i, \zeta_i\right) = \left[\boldsymbol{0}_{d_x} \ 0\right]^\top & \text{(4a)} \\[2mm] \boldsymbol{\varphi}_i\left(\boldsymbol{y}_i; \boldsymbol{\lambda}_i, \zeta_i\right) = \left[\boldsymbol{z}_i^\top \ \omega_i^\top\right], \ \boldsymbol{\psi}_i\left(\boldsymbol{x}_i; \boldsymbol{\lambda}_i, \zeta_i\right) = \left[-\frac{\alpha_i}{|\zeta_i|}\sum_{\xi \in \zeta_i} \partial_{\boldsymbol{x}_i} f_i\left(\frac{\boldsymbol{z}_i}{\omega_i}, \boldsymbol{\lambda}_i; \xi\right)^\top \ 0\right]^\top, & \text{(4b)} \end{cases}$$

where Eq. (4a) and Eq. (4b) run local gradient descent with a minibatch before (Assran et al., 2019) and after (Nedić & Olshevsky, 2016) communication, respectively.

We can recover other popular distributed learning schemes as special cases of SGP. By making $p_{ji}(\delta_j^{(t)})$ form a doubly stochastic mixing matrix $\boldsymbol{P}^{(t)}$, Eq. (4a) and Eq. (4b) recover the decentralized stochastic gradient descent (DSGD) in Bianchi et al. (2013) and Lian et al. (2017), respectively. We can also recover FedAVG (McMahan et al., 2017) by choosing a fully-connected graph with averaging over all clients, i.e., $\delta_{i\rightarrow j}^{(t)} = 1$ and $p_{ij}(\delta_i^{(t)}) = 1/n$ for all $i, j \in [n]$ and $t \in \mathbb{N}$ in Eq. (4a)[1].

## 3 PERSONALIZED DECENTRALIZED BILEVEL OPTIMIZATION (PDBO)

We then propose the formulation of PDBO as a generalization of existing personalization problems. PDBO played by $n$ clients is formulated as follows:

$$\min_{\boldsymbol{\lambda}_1,\ldots,\boldsymbol{\lambda}_n} \frac{1}{n}\sum_{s \in [n]} F_s\left(\boldsymbol{x}_s^*\left(\boldsymbol{\lambda}_1,\ldots,\boldsymbol{\lambda}_n\right), \boldsymbol{\lambda}_s\right), \quad \text{s.t.} \ \boldsymbol{x}_i^* \ \text{satisfies Eq. (1)}, \ \forall i \in [n], \tag{5}$$

where the outer-problem (Eq. (5-left)) lets the $i$-th client find its optimal hyperparameter $\boldsymbol{\lambda}_i$ that minimizes the average of outer-cost $F_i : \mathbb{R}^{d_x} \times \mathbb{R}^{d_\lambda} \mapsto \mathbb{R}$ across all clients. Here, we write $\boldsymbol{x}_i^*\left(\boldsymbol{\lambda}_1,\ldots,\boldsymbol{\lambda}_n\right)$ to show its dependency to hyperparameters explicitly. The generality of Eq. (5) in personalization comes from the flexibility in the choice of $f_i, F_i, \boldsymbol{x}_i$, and $\boldsymbol{\lambda}_i$. For example, suppose that $f_i$ is the cross-entropy loss of a DNN with a feature extractor and classifier parameterized by $\boldsymbol{x}_i$ and $\boldsymbol{\lambda}_i$, respectively. By letting $F_i$ be a validation loss, we can recover a family of personalized layer scheme (Arivazhagan et al., 2019; Bui et al., 2019). See Section 5 for further examples.

We then reformulate PDBO by replacing Eq. (1) with the stationary point of an iteration as in Grazzi et al. (2020). Following the original works of SGP (Nedić & Olshevsky, 2014) and the push-sum (Bénézit et al., 2010), we introduce additional assumptions:

**Assumption 1.** For every $i \in [n]$, and for all $\boldsymbol{\lambda}_i \in \mathbb{R}^{d_\lambda}$ and $\xi_i \in \mathcal{X}$, $f_i\left(\cdot, \boldsymbol{\lambda}_i; \xi_i\right)$ is strongly convex.

**Assumption 2.** A graph with edge set $\{(i,j) \mid \mathbb{E}_{\delta_i}[p_{ij}(\delta_i)] > 0, \ i, j \in [n]\}$ is strongly connected.

Let $\delta = \{\delta_1,\ldots,\delta_n\}$, $\zeta = \{\zeta_1,\ldots,\zeta_n\}$. For every $i \in [n]$, the expectation of iteration Eq. (2) with Assumptions 1 and 2 admits the following unique stationary point which gives the optimum of Eq. (1) (Nedić & Olshevsky, 2014; Assran & Rabbat, 2020):

$$\boldsymbol{y}_i^* = \mathbb{E}_{\delta,\zeta}\left[\sum_{j=1}^n p_{ji}\left(\delta_j\right) \boldsymbol{\varphi}_j(\boldsymbol{y}_j^*; \boldsymbol{\lambda}_j, \zeta_j) + \boldsymbol{\psi}_i(\boldsymbol{y}_i^*; \boldsymbol{\lambda}_i, \zeta_i)\right] = [\boldsymbol{z}_i^{*\top} \ \omega_i^*]^\top \ \text{s.t.} \ \boldsymbol{x}_i^* = \boldsymbol{z}_i^*/\omega_i^*, \tag{6}$$

Replacing the inner-problem in Eq. (5) by Eq. (6) reformulates PDBO as

$$\min_{\boldsymbol{\lambda}} \bar{F}\left(\boldsymbol{x}^*\left(\boldsymbol{y}^*\left(\boldsymbol{\lambda}\right)\right), \boldsymbol{\lambda}\right), \quad \text{s.t. Eq. (6) is satisfied for all } i \in [n], \tag{7}$$

where, $\boldsymbol{\lambda} = [\boldsymbol{\lambda}_1^\top \ \cdots \ \boldsymbol{\lambda}_n^\top]^\top$, $\boldsymbol{x}^* = [\boldsymbol{x}_1^{*\top} \ \cdots \ \boldsymbol{x}_n^{*\top}]^\top$, and $\boldsymbol{y}^*\left(\boldsymbol{\lambda}\right) = [\boldsymbol{y}_1^{*\top} \ \cdots \ \boldsymbol{y}_n^{*\top}]^\top$ are concatenated parameters, and $\bar{F}\left(\boldsymbol{x}, \boldsymbol{\lambda}\right) := \frac{1}{n}\sum_{k \in [n]}[F_k\left(\boldsymbol{x}_k, \boldsymbol{\lambda}_k\right)]$ is the average outer-cost.

---

[1]This is a mathematical equivalence; FedAVG runs on a centralized network in practice.

# 4 HYPER-GRADIENT ESTIMATION OVER STOCHASTIC AND DIRECTED COMMUNICATION NETWORKS

To solve PDBO using gradient-based methods, this section introduces an empirical estimate of the hyper-gradient and its decentralized computation algorithm, which we named HGP.

## 4.1 EMPIRICAL ESTIMATE VIA APPROXIMATE IMPLICIT DIFFERENTIATION

Below, we derive the estimator of hyper-gradient following the recurrent backpropagation for approximate implicit differentiation (Grazzi et al., 2020; Lorraine et al., 2020). The hyper-gradient with respect to $\boldsymbol{\lambda}$ is written as $\mathrm{d}_{\boldsymbol{\lambda}} \bar{F}(\boldsymbol{x}^*(\boldsymbol{y}^*(\boldsymbol{\lambda})), \boldsymbol{\lambda})$ under Assumption 3.

**Assumption 3.** For all $i \in [n]$ and $\zeta_i, \boldsymbol{\varphi}_i(\boldsymbol{y}_i; \boldsymbol{\lambda}_i, \zeta_i)$ and $\boldsymbol{\psi}_i(\boldsymbol{y}_i; \boldsymbol{\lambda}_i, \zeta_i)$ are differentiable with respect to $\boldsymbol{y}_i$ and $\boldsymbol{\lambda}_i$, and $F_i(\boldsymbol{x}_i, \boldsymbol{\lambda}_i)$ is differentiable with respect to $\boldsymbol{x}_i$ and $\boldsymbol{\lambda}_i$.

**Estimator of hyper-gradient** We introduce Jacobian matrices $\boldsymbol{A}(\delta, \zeta)$ and $\boldsymbol{B}(\delta, \zeta)$ whose $(j, i)$ blocks are the partial derivative of Eq. (2) with respect to $\boldsymbol{y}_j$ and $\boldsymbol{\lambda}_j$ for $j, i \in [n]$, respectively:

$$\langle \boldsymbol{A}(\delta, \zeta) \rangle_{ji} = p_{ji}(\delta_j) \partial_{\boldsymbol{y}_j} \boldsymbol{\varphi}_j(\boldsymbol{y}_j^*; \boldsymbol{\lambda}_j, \zeta_j) + \mathbb{1}_{ji} \partial_{\boldsymbol{y}_j} \boldsymbol{\psi}_j(\boldsymbol{y}_j^*; \boldsymbol{\lambda}_j, \zeta_j) \in \mathbb{R}^{d_{\boldsymbol{y}} \times d_{\boldsymbol{y}}}, \quad (8\text{a})$$

$$\langle \boldsymbol{B}(\delta, \zeta) \rangle_{ji} = p_{ji}(\delta_j) \partial_{\boldsymbol{\lambda}_j} \boldsymbol{\varphi}_j(\boldsymbol{y}_j^*; \boldsymbol{\lambda}_j, \zeta_j) + \mathbb{1}_{ji} \partial_{\boldsymbol{\lambda}_j} \boldsymbol{\psi}_j(\boldsymbol{y}_j^*; \boldsymbol{\lambda}_j, \zeta_j) \in \mathbb{R}^{d_{\boldsymbol{\lambda}} \times d_{\boldsymbol{y}}}, \quad (8\text{b})$$

where, $\mathbb{1}_{ij}$ denotes the Kronecker delta. We introduce their expectations by $\bar{\boldsymbol{A}} := \mathbb{E}_{\delta, \zeta}[\boldsymbol{A}(\delta, \zeta)]$ and $\bar{\boldsymbol{B}} := \mathbb{E}_{\delta, \zeta}[\boldsymbol{B}(\delta, \zeta)]$ assuming the following:

**Assumption 4.** The largest singular value of $\bar{\boldsymbol{A}}$ is strictly smaller than one.

Let $\boldsymbol{c}^{\boldsymbol{y}} = \partial_{\boldsymbol{y}} \bar{F}(\boldsymbol{x}^*(\boldsymbol{y}^*(\boldsymbol{\lambda})), \boldsymbol{\lambda})$ and $\boldsymbol{c}^{\boldsymbol{\lambda}} = \partial_{\boldsymbol{\lambda}} \bar{F}(\boldsymbol{x}^*(\boldsymbol{y}^*(\boldsymbol{\lambda})), \boldsymbol{\lambda})$. Using Assumption 4, Eq. (6), and empirical estimates $(\hat{\boldsymbol{A}}^{(t)}, \hat{\boldsymbol{B}}^{(t)}) = (\boldsymbol{A}(\delta^{(t)}, \zeta^{(t)}), \boldsymbol{B}(\delta^{(t)}, \zeta^{(t)}))$, we obtain the estimator as

$$\mathrm{d}_{\boldsymbol{\lambda}} \bar{F}(\boldsymbol{x}^*(\boldsymbol{y}^*(\boldsymbol{\lambda})), \boldsymbol{\lambda}) \approx \bar{\boldsymbol{B}} \sum_{m=0}^{M-1} \bar{\boldsymbol{A}}^m \boldsymbol{c}^{\boldsymbol{y}} + \boldsymbol{c}^{\boldsymbol{\lambda}} \approx \sum_{m=0}^{M-1} \hat{\boldsymbol{B}}^{(2m)} \prod_{s=0}^{m-1} \hat{\boldsymbol{A}}^{(2m+1)} \boldsymbol{c}^{\boldsymbol{y}} + \boldsymbol{c}^{\boldsymbol{\lambda}} =: \widehat{\mathrm{d}_{\boldsymbol{\lambda}} \bar{F}}. \quad (9)$$

where, the first approximation is obtained from Grazzi et al. (2020, (Eq. (4), (5), (19))) and the second approximation simply replaces the expected Jacobians with their estimates, as in Ghadimi & Wang (2018, 3.62, 3.66). We estimate Jacobians in the odd- and even-rounds introducing the following assumption to ensure unbiasedness: $\bar{\boldsymbol{B}} \sum_{m=0}^{M-1} \bar{\boldsymbol{A}}^m = \mathbb{E}[\sum_{m=0}^{M-1} \hat{\boldsymbol{B}}^{(2m)} \prod_{s=0}^{m-1} \hat{\boldsymbol{A}}^{(2m+1)}]$. We include the complete derivation of the estimator in Appendix A.

**Assumption 5.** $\delta^{(t)}$ and $\zeta^{(t)}$ are independent across the time steps $t \in \mathbb{N}$.

**Recurrent backpropagation** We compute Eq. (9) using the fact that a finite number of recurrent backpropagation around the stationary point approximates the hyper-gradient (Lorraine et al., 2020, 4.2), which avoids the explicit computation of Jacobian matrices, $\hat{\boldsymbol{A}}^{(m)}$ and $\hat{\boldsymbol{B}}^{(m)}$. Let $\boldsymbol{u}^{(m)} = \prod_{s=0}^{m-1} \hat{\boldsymbol{A}}^{(2m+1)} \boldsymbol{c}^{\boldsymbol{y}}$ and $\boldsymbol{v}^{(m)} = \sum_{m'=0}^{m-1} \hat{\boldsymbol{B}}^{(2m')} \boldsymbol{u}^{(m')} + \boldsymbol{c}^{\boldsymbol{\lambda}}$. By initializing $\boldsymbol{u}^{(0)} \leftarrow \boldsymbol{c}^{\boldsymbol{y}}$ and $\boldsymbol{v}^{(0)} \leftarrow \boldsymbol{c}^{\boldsymbol{\lambda}}$, and by the following iterations for $m = 0, \ldots, M-1$,

$$\begin{cases} \boldsymbol{v}^{(m+1)} \leftarrow \hat{\boldsymbol{B}}^{(2m)} \boldsymbol{u}^{(m)} + \boldsymbol{v}^{(m)}, \\ \boldsymbol{u}^{(m+1)} \leftarrow \hat{\boldsymbol{A}}^{(2m+1)} \boldsymbol{u}^{(m)}, \end{cases} \quad (10)$$

we obtain the hyper-gradient estimate as $\widehat{\mathrm{d}_{\boldsymbol{\lambda}} \bar{F}} \leftarrow \boldsymbol{v}^{(M)}$. Eq. (10) only requires Jacobian-vector products $\hat{\boldsymbol{B}}^{(2m)} \boldsymbol{u}^{(m)}$ and $\hat{\boldsymbol{A}}^{(2m+1)} \boldsymbol{u}^{(m)}$ leading $O(n d_{\boldsymbol{x}} + n d_{\boldsymbol{\lambda}})$ and $O(n d_{\boldsymbol{x}})$ in time, respectively.

**Decentralizing backpropagation** Decentralized computation of Eq. (10) requires consideration of the data locality and the communication stochasticity. From the locality of $\zeta_i^{(m)}$, clients need to communicate because only the $i$-th client can compute Jacobian-vector products of the $i$-th row block of $\hat{\boldsymbol{A}}^{(m)}$ and $\hat{\boldsymbol{B}}^{(m)}$ from their definitions Eq. (8). Moreover, we need to design a decentralized algorithm so that any required communication can be performed on stochastic and directed networks.

## 4.2 HYPER-GRADIENT PUSH (HGP)

We propose HGP which enables any $i$-th client to update their hyperparameter $\boldsymbol{\lambda}_i$ by estimating its hyper-gradient $\widehat{\mathrm{d}_{\boldsymbol{\lambda}_i}\bar{F}} = \langle \widehat{\mathrm{d}_{\boldsymbol{\lambda}}\bar{F}} \rangle_i$ over stochastic directed networks. HGP runs an unbiased alternative of Eq. (10) based on our observation that the exact computation of Eq. (10) requires undirected edges.

**Exact backpropagation requires undirected edges**  Suppose the $i$-th client is responsible for computing the $i$-th block of $\boldsymbol{u}^{(m+1)}$ and $\boldsymbol{v}^{(m+1)}$, denoted by $\boldsymbol{u}_i^{(m+1)} \in \mathbb{R}^{d_{\boldsymbol{x}}}$ and $\boldsymbol{v}_i^{(m+1)} \in \mathbb{R}^{d_{\boldsymbol{\lambda}}}$, respectively. From Eq. (10), we obtain the following recursive iteration performed by the $i$-th client:

$$
\begin{cases}
\boldsymbol{v}_i^{(m+1)} \leftarrow \sum_{j=1}^n \delta_{i \to j}^{(2m)} \left\langle \hat{\boldsymbol{B}}^{(2m)} \right\rangle_{ij} \boldsymbol{u}_j^{(m)} + \boldsymbol{v}_i^{(m)}, \\
\boldsymbol{u}_i^{(m+1)} \leftarrow \sum_{j=1}^n \delta_{i \to j}^{(2m+1)} \left\langle \hat{\boldsymbol{A}}^{(2m+1)} \right\rangle_{ij} \boldsymbol{u}_j^{(m)},
\end{cases}
\tag{11}
$$

where, we use the equivalencies $\langle \hat{\boldsymbol{A}}^{(m)} \rangle_{ij} = \delta_{i \to j}^{(m)} \langle \hat{\boldsymbol{A}}^{(m)} \rangle_{ij}$ and $\langle \hat{\boldsymbol{B}}^{(m)} \rangle_{ij} = \delta_{i \to j}^{(m)} \langle \hat{\boldsymbol{B}}^{(m)} \rangle_{ij}$, as they are non-zeros only when $\delta_{i \to j}^{(m)} = 1$ from Eq. (8) and Eq. (3). To complete Eq. (11), the $i$-th client needs to receive $\boldsymbol{u}_j^{(m)}$ from all the $j$-th client with $\delta_{i \to j}^{(m)} = 1$, which is possible only when there is the communication channel from $j$ to $i$ (i.e., $\delta_{j \to i}^{(m)} = 1$). In other words, the exact computation of Eq. (11) is available only when the communications are undirected (i.e., $\delta_{i \to j}^{(m)} = \delta_{j \to i}^{(m)}, \forall m \in \mathbb{N}$).

**Unbiased estimation via directed edges**  To relax the undirected communication constraint to stochastic directed communication, we propose HGP as a simple yet effective alternative of Eq. (11).

We first assumes that the $i$-th client knows the receiving frequency $\bar{\delta}_{j \to i} = \mathbb{E}_\delta[\delta_{j \to i}]$ and expected sending weight $\bar{p}_{ij} = \mathbb{E}_\delta[p_{ij}(\delta_i)]$ for all $j \in [n]$. In practice, we can estimate them through $T$ rounds of SGP communication. We also adopt the following assumptions:

**Assumption 6.**  If $\bar{\delta}_{j \to i} > 0$, then $\bar{\delta}_{i \to j} > 0$ and vice versa.

**Assumption 7.**  The realization of $\delta_{i \to j}^{(m)}$ are independent over different $j$ and $i$ for all $m \in \mathbb{N}$.

The key idea of HGP is to replace the sending edges $\delta_{i \to j}^{(m)}$ in Eq. (11) with the debiased receiving edges $\delta_{j \to i}^{(m)}/\bar{\delta}_{j \to i}$. By initializing $\boldsymbol{u}_i^{(0)} \leftarrow \langle \boldsymbol{c}^{\boldsymbol{y}} \rangle_i = \frac{1}{n}\partial_{\boldsymbol{y}_i} F_i(\boldsymbol{x}_i^*, \boldsymbol{\lambda}_i)$ and $\boldsymbol{v}_i^{(0)} \leftarrow \langle \boldsymbol{c}^{\boldsymbol{\lambda}} \rangle_i = \frac{1}{n}\partial_{\boldsymbol{\lambda}_i} F_i(\boldsymbol{x}_i^*, \boldsymbol{\lambda}_i)$, we obtain the estimate as $\widehat{\mathrm{d}_{\boldsymbol{\lambda}_i}\bar{F}} \leftarrow \boldsymbol{v}_i^{(M)}$ after the following iterations for $m = 0, \dots, M-1$,

$$
\begin{cases}
\boldsymbol{v}_i^{(m+1)} \leftarrow \sum_{j=1}^n \frac{\delta_{j \to i}^{(2m)}}{\bar{\delta}_{j \to i}} \left\langle \tilde{\boldsymbol{B}}^{(2m)} \right\rangle_{ij} \boldsymbol{u}_j^{(m)} + \boldsymbol{v}_i^{(m)}, \\
\boldsymbol{u}_i^{(m+1)} \leftarrow \sum_{j=1}^n \frac{\delta_{j \to i}^{(2m+1)}}{\bar{\delta}_{j \to i}} \left\langle \tilde{\boldsymbol{A}}^{(2m+1)} \right\rangle_{ij} \boldsymbol{u}_j^{(m)},
\end{cases}
\tag{12}
$$

where, $\langle \tilde{\boldsymbol{A}}^{(m)} \rangle_{ij}$ and $\langle \tilde{\boldsymbol{B}}^{(m)} \rangle_{ij}$ are defined by replacing $p_{ij}(\delta_i^{(m)})$ in Eq. (8a) and Eq. (8b) with $\bar{p}_{ij}$, respectively. The iterations above are always computable even on stochastic directed networks because the $i$-th client needs to receive $\boldsymbol{u}_j^{(m)}$ from the clients with $\delta_{j \to i}^{(m)} = 1$, which is always possible. We also note that Assumption 6 ensures that $\langle \tilde{\boldsymbol{A}}^{(m)} \rangle_{ij}$ and $\langle \tilde{\boldsymbol{B}}^{(m)} \rangle_{ij}$ are unbiased: $\mathbb{E}_{\delta,\zeta}[\delta_{j \to i}^{(m)}/\bar{\delta}_{j \to i} \langle \tilde{\boldsymbol{A}}^{(m)} \rangle_{ij}] = \mathbb{E}_\delta[\delta_{j \to i}^{(m)}/\bar{\delta}_{j \to i}]\mathbb{E}_{\delta,\zeta}[\langle \hat{\boldsymbol{A}}^{(m)} \rangle_{ij}] = \langle \bar{\boldsymbol{A}} \rangle_{ij}$ and the same for $\hat{\boldsymbol{B}}^{(m)}$.

HGP enjoys the same complexity as SGP in both communication and computation. HGP exchanges only $\boldsymbol{u}_i^{(\cdot)}$ having $O(d_{\boldsymbol{y}})$ in communication. In practical cases where $d_{\boldsymbol{\lambda}} \ll d_{\boldsymbol{y}}$, the Jacobian-vector products $\langle \tilde{\boldsymbol{B}}^{(\cdot)} \rangle_{ij} \boldsymbol{u}_j^{(\cdot)}$ and $\langle \tilde{\boldsymbol{A}}^{(\cdot)} \rangle_{ij} \boldsymbol{u}_j^{(\cdot)}$ are computed in $O(d_{\boldsymbol{y}})$ time.

**Variance reduction**  We now introduce the variance-reduced version of HGP, which we call VR-HGP. The naive HGP above suffers from large variance because of $\delta_{j \to i}^{(m)}/\bar{\delta}_{j \to i}$, which can take a value far larger than one when $\bar{\delta}_{j \to i}$ is small. The multiplication of such values induces a high variance.

The idea of VR-HGP is to combine HGP with its following variant, where $\boldsymbol{w}_i^{(0)} \leftarrow \langle \boldsymbol{c^y} \rangle_i$,

$$
\begin{cases}
\boldsymbol{v}_i^{(m+1)} = \sum_{j=1}^{n} \frac{\delta_{j\to i}^{(2m)}}{\bar{\delta}_{j\to i}} \left\langle \tilde{\boldsymbol{B}}^{(2m)} \right\rangle_{ij} \boldsymbol{w}_j^{(m)}, \\
\boldsymbol{w}_i^{(m+1)} = \sum_{j=1}^{n} \frac{\delta_{j\to i}^{(2m+1)}}{\bar{\delta}_{j\to i}} \left\langle \tilde{\boldsymbol{A}}^{(2m+1)} \right\rangle_{ij} \boldsymbol{w}_j^{(m)} + \langle \boldsymbol{c^y} \rangle_i.
\end{cases}
\tag{13}
$$

Here, $\boldsymbol{w}^{(m)}$ corresponds to the estimator of $\sum_{m'=0}^{m-1} \boldsymbol{A}^{m'} \boldsymbol{c^y}$. Note that the weighted average of two different estimators results in an estimator with a smaller variance. By averaging Eq. (12) and Eq. (13) with weights $\alpha, \beta \in (0, 1)$, we obtain VR-HGP as the following iterations for $m = 0, \ldots, M - 1$:

$$
\begin{cases}
\boldsymbol{v}_i^{(m+1)} \leftarrow \alpha \left( \boldsymbol{v}_i^{(m)} + \sum_j \frac{\delta_{j\to i}^{(2m)}}{\bar{\delta}_{j\to i}} \left\langle \tilde{\boldsymbol{B}}^{(2m)} \right\rangle_{ij} \boldsymbol{u}_j^{(m)} \right) + (1-\alpha) \left( \sum_j \frac{\delta_{j\to i}^{(2m)}}{\bar{\delta}_{j\to i}} \left\langle \tilde{\boldsymbol{B}}^{(2m)} \right\rangle_{ij} \boldsymbol{w}_j^{(m)} \right), \\
\boldsymbol{u}_i^{(m+1)} \leftarrow \sum_j \frac{\delta_{j\to i}^{(2m+1)}}{\bar{\delta}_{j\to i}} \left\langle \tilde{\boldsymbol{A}}^{(2m+1)} \right\rangle_{ij} \boldsymbol{u}_j^{(m)}, \\
\boldsymbol{w}_i^{(m+1)} \leftarrow \beta \left( \sum_j \frac{\delta_{j\to i}^{(2m+1)}}{\bar{\delta}_{j\to i}} \left\langle \tilde{\boldsymbol{A}}^{(2m+1)} \right\rangle_{ij} \boldsymbol{w}_j^{(m)} + \langle \boldsymbol{c^y} \rangle_i \right) + (1-\beta) \left( \boldsymbol{w}_i^{(m)} + \boldsymbol{u}_i^{(m+1)} \right),
\end{cases}
$$

with $\boldsymbol{v}_i^{(0)} \leftarrow \boldsymbol{0}_{d_{\boldsymbol{\lambda}}}$, $\boldsymbol{u}_i^{(0)} \leftarrow \langle \boldsymbol{c^y} \rangle_i$, and $\boldsymbol{w}_i^{(0)} \leftarrow \langle \boldsymbol{c^y} \rangle_i$ having the estimate as $\widehat{\mathrm{d}_{\boldsymbol{\lambda}_i}} \bar{F} \leftarrow \boldsymbol{v}_i^{(M)} + \langle \boldsymbol{c^\lambda} \rangle_i$.

The following theorem provides the estimation error of the hyper-gradient using VR-HGP.

**Assumption 8.** $\exists \eta_A \in (0, 1), \eta_B \in (0, \infty)$ such that $\forall \boldsymbol{y}_i, \boldsymbol{\lambda}_i, \zeta_i$ and $\forall i$,

$$
\max \left\{ \|\partial_{\boldsymbol{y}_i} \boldsymbol{\psi}_i\|_2, \|\partial_{\boldsymbol{y}_i} \boldsymbol{\varphi}_i\|_2 \right\} \leq \frac{\eta_A}{2\kappa}, \quad \max \left\{ \|\partial_{\boldsymbol{\lambda}_i} \boldsymbol{\psi}_i\|_2, \|\partial_{\boldsymbol{\lambda}_i} \boldsymbol{\varphi}_i\|_2 \right\} \leq \frac{\eta_B}{2\kappa},
\tag{14}
$$

where $\kappa = \sum_{i,j} \frac{\bar{p}_{ji}}{\bar{\delta}_{i\to j}}$ and $\|\cdot\|_2$ denotes spectral norm.

**Theorem 1** (Estimation Error of VR-HGP). Suppose that Assumptions 1–8 hold true and $|\zeta_i^{(2m)}| = |\zeta_i^{(2m+1)}| = b$ for any $i$ and $m$. Then, for $\alpha, \beta \in (0, 1)$, with probability at least $1 - \epsilon$, we have

$$
\left\| \widehat{\mathrm{d}_{\boldsymbol{\lambda}_i}} \bar{F} - \mathrm{d}_{\boldsymbol{\lambda}} \bar{F} \left( \boldsymbol{x}^*, \boldsymbol{\lambda} \right) \right\| \leq \mu_{\alpha,\beta} \tau \sqrt{ \left( \sum_{i,j} \frac{\bar{p}_{ji}^2}{\bar{\delta}_{i\to j}^2} + \frac{4n}{b} \right) \log \frac{n(d_{\boldsymbol{y}} + d_{\boldsymbol{\lambda}})}{\epsilon} } + e^{-O(M)},
$$

where, $\|\cdot\|$ denotes $\ell_2$ norm, $e^{-O(M)}$ denotes the exponentially diminishing term over $M$, and

$$
\mu_{\alpha,\beta} = \sqrt{ 8 \frac{1-\alpha}{1+\alpha} \left( 1 + \frac{1 + \alpha(1 - \beta + \beta\eta_A)}{1 - \alpha(1 - \beta + \beta\eta_A)} \frac{\beta^2 \eta_A^2}{1 - (1 - \beta + \beta\eta_A)^2} \right) }, \quad \tau = \frac{\eta_B \|\boldsymbol{c^y}\|}{\kappa(1 - \eta_A)}.
$$

One can see that the coefficient $\mu_{\alpha,\beta}$ dominates the magnitude of the estimation error. Setting $\alpha, \beta \in (0, 1)$ that minimizes $\mu_{\alpha,\beta}$ can attain a small error.[2] The proof is provided in Appendix C.

## 5 RELATED WORK

**Personalization in federated learning** We compare our work to standard personalization methods by recovering them as special cases of PDBO and pointing their applicable communication networks.

Mansour et al. (2020) and Deng et al. (2020) propose model interpolation that provides personalized models as the optimal interpolation between local models and the global model, which is recovered by letting the inner-problem train the global model and the outer-problem optimize the interpolation weight. Federated multi-task learning (MTL) (Marfoq et al., 2021) obtains personalized models by allowing clients to tune the ensemble weights of the global base-predictors. In Section 6, we demonstrated that PDBO can recover the federated MTL by letting the inner-problem optimize the base-predictors and the outer-problem learn ensemble weights. We see the clustering personalization (Sattler et al., 2020) as a sub-problem of federated MTL from the empirical results (Marfoq et al.,

---

[2]Although setting $\alpha = 1$ makes $\mu_{\alpha,\beta} = 0$, the remaining error is no longer $e^{-O(M)}$ in that case. This observation implies that $\alpha$ slightly smaller than 1 is preferred. A similar analysis also shows that $\beta$ slightly larger than 0 is preferred. Our empirical results show that $(\alpha, \beta) = (0.9, 0.1)$ performs well (Appendix C.6).

2021, J.4) that demonstrated the personalized ensemble weights recover the client clusters. Data augmentation (Duan et al., 2019; Zhao et al., 2018) mitigates data heterogeneity by over- or under-sampling to train a generalized global model. This can be recovered by optimizing pseudo-sampling rates as hyperparameters. Furthermore, the generality of PDBO allows us to optimize different types of parameters simultaneously, which current personalization algorithms cannot handle.

For communication networks, most personalization schemes require a centralized network (Sattler et al., 2020; Jiang et al., 2019), which is vulnerable to a central point of failure (Assran et al., 2019). A few fully-decentralized algorithms (Marfoq et al., 2021; Lu et al., 2022) assume static undirected networks which are vulnerable to failing clients and deadlocks (Tsianos et al., 2012). While Vanhaesebrouck et al. (2017) and Zantedeschi et al. (2020) consider stochastic undirected settings, their applicability are limited to linear models or a linear combination of pre-trained models. Gradient-based PDBO can learn more complex models and run on stochastic directed networks, which are immune to practical problems in centralized and static undirected networks.

**Distributed bilevel optimization**   Distributed bilevel optimizations proposed in concurrent works differ from PDBO in formulations. We categorize them into consensus distributed bilevel optimization (CDBO) (Chen et al., 2022; Tarzanagh et al., 2022; Gao et al., 2022; Yang et al., 2022) and CDBO with the local inner-problem (CDBO-Local) (Li et al., 2022; Liu et al., 2022; Lu et al., 2022).

CDBO requires clients to make a consensus both on the outer- and inner-problem. Chen et al. (2022); Tarzanagh et al. (2022); Gao et al. (2022); Yang et al. (2022) applied CDBO to hyperparameter optimization, such as L2 regularization rates. While PDBO and CDBO are different tasks, both require hyper-gradient estimation over communications, which we discuss in the next paragraph. CDBO-Local (Lu et al., 2022) requires consensus in the outer-problem as in CDBO, whereas its inner-problem is a local optimization. Clients in CDBO-Local thus cannot benefit from others in the inner-loop for better generalization. In our PDBO, both outer- and inner-problems are optimized using global information; the inner-parameters are trained for consensus, and the outer-parameters are optimized to improve the total performance across all clients.

We highlight that our gradient-based PDBO recovers CDBO by running SGP using the estimated hyper-gradient for the outer-problem, and recovers CDBO-Local by using SGP for the outer-problem and designing $p_{ij}$ to form the self-loop topology in the inner SGP.

**Hyper-gradient estimation over communication Networks**   We compare our HGP with the other hyper-gradient estimation methods performed over communication networks.

Yang et al. (2022) proposes a hyper-gradient estimation algorithm in fully-decentralized settings. However, they assume static and undirected networks, and their algorithm is complex both in computation and communication as they involve computations and communications of full Jacobians and Hessians. Tarzanagh et al. (2022) considers the hyper-gradient estimation in centralized settings, which is typical in federated learning. While their approach is advantageous in complexity because clients only compute Jacobian-vector products and exchange $O(d_{\boldsymbol{x}})$ vectors, its applicability is tied to the centralized host-clients setting. Other CDBO methods (Chen et al., 2022; Gao et al., 2022) estimate different types of hyper-gradient. See Appendix E for further details.

Our HGP enjoys reasonable complexity in computation and communication, as stated in Section 4.2, and covers a wide range of communication networks, including stochastic and directed networks.

**Hyper-gradient estimation for single agent**   The hyper-gradient estimation approaches are categorized into iterative differentiation (ITD), and approximate implicit differentiation with recurrent backpropagation (AID-RB) and conjugated gradient (AID-CG) (Grazzi et al., 2020). We found that applying ITD or AID-CG to the hyper-gradient estimation on stochastic and directed communication networks is infeasible for the following reasons.

Applying the ITD variants, backward and forward mode (Franceschi et al., 2017), have limitations in communication; the backward mode requires static and undirected network, the forward mode requires all-to-all communication at the end of iteration and exchanging large $O(d_{\boldsymbol{y}} \times d_{\boldsymbol{\lambda}})$ sized matrices. A detailed discussion is provided in Appendix H. To apply AID-CG (Pedregosa, 2016), we need to solve $\min_{\boldsymbol{q} \in \mathbb{R}^{d_{\boldsymbol{y}}}} \frac{1}{2} \|(\boldsymbol{I} - \bar{\boldsymbol{A}})\boldsymbol{q} - \boldsymbol{c}^{\boldsymbol{y}}\|^2$. However, in our setting where $\boldsymbol{I} - \bar{\boldsymbol{A}}$ can be

asymmetric, AID-CG is slower than AID-RB (Grazzi et al., 2020). AID-RB only requires the network to be undirected and our HGP relaxes this limitation by simple and effective modification.

## 6 EXPERIMENTS

To demonstrate the generality in personalization and applicability to practical communication environments, we introduced three different personalization approaches as special cases of gradient-based PDBO and benchmarked them with baselines on four different communication networks.

### 6.1 SETTINGS

We followed the settings of EMNIST classification played by $n = 100$ clients in Marfoq et al. (2021) unless otherwise mentioned. The detailed experimental settings are described in Appendix D.

**Communication networks**   We simulated four communication networks: fully-connected (FC), static undirected (FixU), stochastic undirected (StoU), and stochastic directed (StoD).

FC allows clients to communicate with all the others at any time step, i.e. $\delta_{i \to j}^{(t)} = 1$ for all $i, j \in [n]$ and $t \in \mathbb{N}$. FixU is static undirected network simulated by a binomial Erdős-Rényi graph (Erdős & Rényi, 1959) with parameter $p = 0.4$ adding the self-loop edges. Following the setting in Marfoq et al. (2021), we generated a doubly stochastic mixing matrix using the fast-mixing Markov chain (Boyd et al., 2003) rule. StoU simulates stochastic undirected network by letting undirected edge $\bar{\delta}_{j \to i}^{(t)} = \delta_{i \to j}^{(t)}$ independently realize at each step with the probability $\bar{\delta}_{j \to i} \in [0, 1]$. In StoD, every direction of edges $\delta_{j \to i}^{(t)}$ is independently sampled at probability $\bar{\delta}_{j \to i}$, simulating a stochastic directed network. For all $i, j \in [n]$, $\bar{\delta}_{j \to i}$ was sampled from the uniform distribution with $[0.4, 0.8]$.

**Proposed approaches**   We introduce and evaluated three different personalization methods as special cases of PDBO, that are, PDBO-DA, PDBO-MTL, and PDBO-MTL&DA.

PDBO-DA optimizes the pseudo-sampling rates to recover the data-augmentation-based personalization (Duan et al., 2019; Zhao et al., 2018). PDBO-DA optimizes $\boldsymbol{\lambda}_i \in \mathbb{R}^C$ to obtain the label-wise weight vector $C \texttt{Softmax}(\boldsymbol{\lambda}_i) \in [0, C]^C$. In the inner-problem, the losses of the instances labeled as $c \in [C]$ are multiplied by the $c$-th element of the weight vector. PDBO-MTL is obtained by formulating FedEM (Marfoq et al., 2021) as PDBO. PDBO-MTL lets each client train an ensemble classifier that outputs weighted average predictions across $K = 3$ of CNNs. PDBO-MTL trains the CNN parameters as the inner-problem and optimizes the hyperparameters $\boldsymbol{\lambda}_i \in \mathbb{R}^K$ to obtain the ensemble weight vector $\texttt{Softmax}(\boldsymbol{\lambda}_i) \in [0, 1]^K$. PDBO-MTL&DA combines PDBO-DA and PDBO-MTL by optimizing $\boldsymbol{\lambda}_i \in \mathbb{R}^{C+K}$ to obtain both the label weight and ensemble weight vectors.

**Baseline approaches**   We compared our approaches with baselines on each communication setting. For FC and FixU settings, we compared several personalization approaches: a personalized model trained only on the local dataset (Local), FedAvg with local tuning (FedAvg+) (Jiang et al., 2019), Clustered-FL (Sattler et al., 2020), pFedMe (T Dinh et al., 2020), and centralized and decentralized versions of FedEM (Marfoq et al., 2021). We also trained the global models using SGP (Nedić & Olshevsky, 2016; Assran et al., 2019) and FedProx (Li et al., 2020). As SGP recovers FedAvg and DSGD on FC and FixU, respectively, we treat them as equivalent approaches. Among all approaches including ours, model architecture follows the setting in Marfoq et al. (2021).

**Training procedure**   We allowed every client to generate its local dataset which has its unique label distribution, following Marfoq et al. (2021), and split it into train, validation, and test datasets.

All baselines and PDBO inner-optimizations ran the distributed learning following Marfoq et al. (2021) on FC and FixU, and ran SGP of Eq. (4b) on StoU and StoD using the train dataset. In PDBO, any $i$-th client estimates $\bar{\delta}_{j \to i}, \bar{p}_{ij}$ for all $j \in [n]$ through communications in the inner-optimization, and approximates $\boldsymbol{y}_i^*$ by $\boldsymbol{y}_i^{(T)}$ obtained from the $T$ steps of inner-optimization. Theorem 11 in Appendix C.7 proves this approximation of $\boldsymbol{y}_i^*$ is reasonable when $\|\boldsymbol{y}_i^{(T)} - \boldsymbol{y}_i^*\|$ is sufficiently small.

Table 1: Test accuracy of personalized models on EMNIST (average clients / 10% percentile).

| | Method | Communication network | | | |
|---|---|---|---|---|---|
| | | FC | FixU | **StoU** | **StoD** |
| Global | SGP(FedAvg/DSGD) | 82.2 / 73.8 | 82.3 / 74.1 | 79.7 / 71.6 | 79.7 / 72.5 |
| | FedProx | 69.6 / 58.2 | n/a | n/a | n/a |
| Personalized | Local | 74.7 / 63.9 | 74.7 / 63.9 | 73.7 / 63.8 | 73.7 / 63.8 |
| | FedAvg+ | 83.0 / 75.1 | n/a | n/a | n/a |
| | Clustered-FL | 82.3 / 73.8 | n/a | n/a | n/a |
| | pFedMe | 76.2 / 65.7 | n/a | n/a | n/a |
| | FedEM | **83.9** / 75.9 | 83.8 / 75.9 | n/a | n/a |
| | **PDBO-DA** | 82.9 / 74.8 | 83.0 / 75.5 | 80.9 / 73.2 | 80.8 / 72.9 |
| | **PDBO-MTL** | **83.9** / **76.5** | 83.9 / 76.5 | 81.6 / 73.8 | 81.6 / **75.0** |
| | **PDBO-MTL&DA** | **83.9** / 76.2 | **84.0** / **77.3** | **83.0** / **76.3** | **82.2** / 74.5 |

PDBO outer-optimizations ran 20 outer-steps tracing the average validation accuracy, and we reported the average test accuracy at an outer-step that showed the best validation accuracy. Outer-steps were performed by Adam (Kingma & Ba, 2015) from the zeros initial hyperparameters $\mathbf{0}_{d_\lambda}$. To estimate the hyper-gradient for each outer-step, clients ran $M = 200$ HGP iterations with Eq. (4b) using the average cross-entropy on the train dataset as $F_i$. We adopted HGP for FC and FixU, and VR-HGP with $(\alpha, \beta) = (0.9, 0.1)$ for StoU and StoD. We also made a practical modification in HGP to sample $\tilde{\boldsymbol{A}}^{(m)}$ and $\hat{\boldsymbol{B}}^{(m)}$ together at the single $m$-th round, which leads to the same length of the Neumann series with the half sampling costs of the original HGP, while they are no longer unbiased.

## 6.2 RESULTS AND DISCUSSIONS

**Personalization performance** Table 1 shows the average test accuracy with weights proportional to local test dataset sizes. We observed that the ensemble-based approaches, FedEM, PDBO-MTL, and PDBO-MTL&DA performed the best on FC, and PDBO-MTL&DA outperformed on all fully-decentralized settings, that are, FixU, StoU, and StoD. Although PDBO-DA improved the average accuracy from SGP in all communication settings, it was especially effective when combined with PDBO-MTL. These results indicate that optimizing different parameters simultaneously, which is newly enabled by our PDBO, is advantageous to the personalization performance.

We also investigated whether the accuracy gain was shared among all clients. Table 1 shows the accuracy of the bottom 10% percentile of clients. All our approaches improved accuracy at the 10% percentile from global model approaches (SGP and FedProx) in all communication settings, confirming that the clients fairly benefited from our personalization.

**Applicability to stochastic communication networks** The communication network limits the available personalization methods, especially when the network is stochastic. Although FedEM is one of the few personalization methods feasible in fully-decentralized settings, it requires the doubly stochastic mixing matrix to be known, which is impractical on stochastic networks (Tsianos et al., 2012). As PDBO encompasses SGP and HGP can run on stochastic communication networks, our approaches succeeded in the personalization on StoU and StoD.

**Robustness to communication directionality** Our HGP and VR-HGP estimate hyper-gradient solely from the directed communication edges, rather than running the standard recurrent backpropagation which requires undirected edges. The improvement in our approaches on StoD demonstrated that VR-HGP estimated the hyper-gradient with sufficiently small errors to solve PDBO.

## 7 CONCLUSION

This study proposed a gradient-based PDBO, which reduces most personalization approaches to the optimization of hyperparameters possessed by each client. We also proposed HGP that estimates the hyper-gradient through communications over stochastic and directed communication networks. In addition, we introduced a variance-reduced HGP that mitigated the estimation variance caused by the stochasticity of communication edges and provided its theoretical error bound. Our empirical results demonstrated that our gradient-based PDBO with HGP enabled combining different personalization approaches which led to state-of-the-art performance, and it performed on different simulated communication environments including a stochastic and directed network.

## REPRODUCIBILITY STATEMENT

We provide the detailed experiment settings of Section 6 in Appendix D including any modification to the benchmark conducted by Marfoq et al. (2021) and will be releasing their implementations after the review process. Our theoretical contributions and required assumptions are stated in Section 4.2. The detailed derivations of our HGP and VR-HGP are provided in Appendix A and Appendix B, respectively. We also provide detailed proof of the estimation error bound of VR-HGP in Appendix C. The code for reproducing the results in Section 6 and Appendix G are provided by a separated supplement.

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

## A   ESTIMATION OF HYPER-GRADIENT

**Notation**   For a vector $v \in \mathbb{R}^d$, $\|v\| = \sqrt{\sum_{i=1}^d v_i^2}$ is its $\ell_2$ norm. For a matrix $V \in \mathbb{R}^{d_1 \times d_2}$, $\|V\|_2$ is its largest singular value.

### A.1   STATIONARITY OF SGP

We consider the generalized version of SGP over $n$ nodes as follows:

$$\boldsymbol{y}_i^{(t+1)} = \sum_{j=1}^n p_{ji}(\delta_j^{(t)}) \boldsymbol{\varphi}_j \left(\boldsymbol{y}_j^{(t)}; \boldsymbol{\lambda}_j, \zeta_j^{(t)}\right) + \boldsymbol{\psi}_i \left(\boldsymbol{y}_i^{(t)}; \boldsymbol{\lambda}_i, \zeta_i^{(t)}\right), \;\; p_{ji}(\delta_j^{(t)}) = \delta_{j \to i}^{(t)} p_{ji} \left(\delta_{j \to 1}^{(t)}, \dots, \delta_{j \to n}^{(t)}\right),$$

We set $\delta_{i \to i}^{(t)} = 1$ for all $i$ and $t$, i.e., every client can send a message to itself at any time step.

Assumptions 1 and 2 ensures existence of the unique stationary point $\boldsymbol{y}^*$.

$$\boldsymbol{y}_i^* = \mathbb{E}_{\delta, \zeta} \left[ \sum_{j=1}^n p_{ji} \boldsymbol{\varphi}_j \left(\boldsymbol{y}_j^*; \boldsymbol{\lambda}_j, \zeta_j\right) + \boldsymbol{\psi}_i \left(\boldsymbol{y}_i^*; \boldsymbol{\lambda}_i, \zeta_i\right) \right]$$

$$= \sum_{j=1}^n \bar{p}_{ji} \mathbb{E}_\zeta \left[ \boldsymbol{\varphi}_j \left(\boldsymbol{y}_j^*; \boldsymbol{\lambda}_j, \zeta_j\right) \right] + \mathbb{E}_\zeta \left[ \boldsymbol{\psi}_i \left(\boldsymbol{y}_i^*; \boldsymbol{\lambda}_i, \zeta_i\right) \right],$$

where $\bar{p}_{ji} = \mathbb{E}_\delta \left[ p_{ji} \right]$.

### A.2   HYPER-GRADIENT BY IMPLICIT DIFFERENTIATION

We adopt Assumption 3 so that $\boldsymbol{\varphi}$, $\boldsymbol{\psi}$, and $\bar{F}$ to be differentiable. The differentiation of $\boldsymbol{y}_i^*$ by $\boldsymbol{\lambda}_j$ is

$$d_{\boldsymbol{\lambda}_j} \boldsymbol{y}_i^* = \sum_{j=1}^n \bar{p}_{ji} \left( d_{\boldsymbol{\lambda}_j} \boldsymbol{y}_i^* \partial_{\boldsymbol{y}_i} \mathbb{E}_\zeta \left[ \boldsymbol{\varphi}_j \left(\boldsymbol{y}_j^*; \boldsymbol{\lambda}_j, \zeta_j\right) \right] + \partial_{\boldsymbol{\lambda}_j} \mathbb{E}_\zeta \left[ \boldsymbol{\varphi}_j \left(\boldsymbol{y}_j^*; \boldsymbol{\lambda}_j, \zeta_j\right) \right] \right)$$

$$+ \mathbb{1}_{ji} \left( d_{\boldsymbol{\lambda}_j} \boldsymbol{y}_i^* \partial_{\boldsymbol{y}_i} \mathbb{E}_\zeta \left[ \boldsymbol{\psi}_i \left(\boldsymbol{y}_i^*; \boldsymbol{\lambda}_i, \zeta_i\right) \right] + \partial_{\boldsymbol{\lambda}_i} \mathbb{E}_\zeta \left[ \boldsymbol{\psi}_i \left(\boldsymbol{y}_i^*; \boldsymbol{\lambda}_i, \zeta_i\right) \right] \right).$$

Let $\boldsymbol{y}^* = [\boldsymbol{y}_i^*]_i$ and $\boldsymbol{\lambda} = [\boldsymbol{\lambda}_i]_i$ be the concatenated parameters and hyperparameters, respectively. We can write the differentiation in the matrix form by

$$d_{\boldsymbol{\lambda}} \boldsymbol{y}^* = d_{\boldsymbol{\lambda}} \boldsymbol{y}^* \bar{\boldsymbol{A}} + \bar{\boldsymbol{B}},$$

where

$$\bar{\boldsymbol{A}} = \left[ \mathbb{1}_{ji} \bar{\boldsymbol{A}}_i^\psi + \bar{p}_{ji} \bar{\boldsymbol{A}}_j^\varphi \right]_{ji} \in \mathbb{R}^{nd_{\boldsymbol{y}} \times nd_{\boldsymbol{y}}},$$

$$\bar{\boldsymbol{A}}_i^\psi = \partial_{\boldsymbol{y}_i} \mathbb{E}_\zeta \left[ \boldsymbol{\varphi}_i(\boldsymbol{y}_i^*; \boldsymbol{\lambda}_i, \zeta_i) \right] \in \mathbb{R}^{d_{\boldsymbol{y}} \times d_{\boldsymbol{y}}}, \qquad \bar{\boldsymbol{A}}_j^\varphi = \partial_{\boldsymbol{y}_j} \mathbb{E}_\zeta \left[ \boldsymbol{\varphi}_j(\boldsymbol{y}_j^*; \boldsymbol{\lambda}_j, \zeta_j) \right] \in \mathbb{R}^{d_{\boldsymbol{y}} \times d_{\boldsymbol{y}}},$$

$$\bar{\boldsymbol{B}} = \left[ \mathbb{1}_{ji} \bar{\boldsymbol{B}}_i^\psi + \bar{p}_{ji} \bar{\boldsymbol{B}}_j^\varphi \right]_{ji} \in \mathbb{R}^{nd_{\boldsymbol{\lambda}} \times nd_{\boldsymbol{y}}},$$

$$\bar{\boldsymbol{B}}_i^\psi = \partial_{\boldsymbol{\lambda}_i} \mathbb{E}_\zeta \left[ \boldsymbol{\varphi}_i(\boldsymbol{y}_i^*; \boldsymbol{\lambda}_i, \zeta_i) \right] \in \mathbb{R}^{d_{\boldsymbol{\lambda}} \times d_{\boldsymbol{y}}}, \qquad \bar{\boldsymbol{B}}_j^\varphi = \partial_{\boldsymbol{\lambda}_j} \mathbb{E}_\zeta \left[ \boldsymbol{\varphi}_j(\boldsymbol{y}_j^*; \boldsymbol{\lambda}_j, \zeta_j) \right] \in \mathbb{R}^{d_{\boldsymbol{\lambda}} \times d_{\boldsymbol{y}}}.$$

Then, we have

$$d_{\boldsymbol{\lambda}} \boldsymbol{y}^* = \bar{\boldsymbol{B}} (\boldsymbol{I} - \bar{\boldsymbol{A}})^{-1}.$$

In particular, we have

$$d_{\boldsymbol{\lambda}_j} \boldsymbol{y}_i^* = \sum_k \langle \bar{\boldsymbol{B}} \rangle_{jk} \langle (\boldsymbol{I} - \bar{\boldsymbol{A}})^{-1} \rangle_{ki},$$

where $\langle \cdot \rangle_{jk}$ and $\langle \cdot \rangle_{ki}$ denotes the $(j, k)$-th and $(k, i)$-th block of the matrix.

The hyper-gradient of the objective function $F(\boldsymbol{x}^*, \boldsymbol{\lambda}) = \sum_i F_i(\boldsymbol{x}_i^*, \boldsymbol{\lambda}_i)$ is then given as

$$d_{\boldsymbol{\lambda}_j} F(\boldsymbol{y}^*, \boldsymbol{\lambda}) = \sum_i d_{\boldsymbol{\lambda}_j} \boldsymbol{y}_i^* \underbrace{\partial_{\boldsymbol{y}_i} F_i(\boldsymbol{y}_i^*, \boldsymbol{\lambda}_i)}_{\boldsymbol{c}_i^{\boldsymbol{y}}} + \underbrace{\partial_{\boldsymbol{\lambda}_j} F_j(\boldsymbol{y}_j^*, \boldsymbol{\lambda}_j)}_{\boldsymbol{c}_j^{\boldsymbol{\lambda}}}$$

$$= \sum_i d_{\boldsymbol{\lambda}_j} \boldsymbol{y}_i^* \boldsymbol{c}_i^{\boldsymbol{y}} + \boldsymbol{c}_j^{\boldsymbol{\lambda}}.$$

### A.3 ESTIMATION OF HYPERGRADIENT

In the remainder, we consider $\psi_i$ and $\varphi_j$ of the following forms:

$$\psi_i(\boldsymbol{y}_i; \boldsymbol{\lambda}_i, \zeta_i) = \frac{1}{|\zeta_i|} \sum_{\xi \in \zeta_i} g_i(\boldsymbol{y}_i; \boldsymbol{\lambda}_i, \xi)$$

$$\varphi_j(\boldsymbol{y}_j; \boldsymbol{\lambda}_j, \zeta_j) = \frac{1}{|\zeta_j|} \sum_{\xi \in \zeta_i} h_j(\boldsymbol{y}_j; \boldsymbol{\lambda}_j, \xi),$$

for some $g_i(\cdot; \boldsymbol{\lambda}_i, \xi) : \mathbb{R}^{d_{\boldsymbol{y}}} \to \mathbb{R}^{d_{\boldsymbol{y}}}$ and $h_j(\cdot; \boldsymbol{\lambda}_j, \xi) : \mathbb{R}^{d_{\boldsymbol{y}}} \to \mathbb{R}^{d_{\boldsymbol{y}}}$, which are true for SGP in Eq. (4a) and Eq. (4b). Assumption 3 ensures that $g_i$ and $h_j$ are differentiable with respect to both $\boldsymbol{y}$ and $\boldsymbol{\lambda}$.

#### A.3.1 ESTIMATION OF $\bar{\boldsymbol{A}}$ AND $\bar{\boldsymbol{B}}$

Because the matrices $\bar{\boldsymbol{A}}$ and $\bar{\boldsymbol{B}}$ are defined as the expectation over the data minibatch $\zeta_i, \zeta_j$ as well as the realization of communication network $\delta$, we estimate them from the observation as follows.

$$\hat{\boldsymbol{A}} = \left[ \mathbb{1}_{ji} \hat{\boldsymbol{A}}_i^{\psi} + \bar{p}_{ji} \hat{\boldsymbol{A}}_j^{\varphi} \right]_{ji} \in \mathbb{R}^{nd_{\boldsymbol{y}} \times nd_{\boldsymbol{y}}},$$

$$\hat{\boldsymbol{A}}_i^{\psi} = \frac{1}{|\zeta_i|} \sum_{\xi \in \zeta_i} \partial_{\boldsymbol{y}_i} g_i(\boldsymbol{y}_i^*; \boldsymbol{\lambda}_i, \xi) \in \mathbb{R}^{d_{\boldsymbol{y}} \times d_{\boldsymbol{y}}}, \quad \hat{\boldsymbol{A}}_j^{\varphi} = \frac{1}{|\zeta_j|} \sum_{\xi \in \zeta_j} \partial_{\boldsymbol{y}_j} h_j(\boldsymbol{y}_j^*; \boldsymbol{\lambda}_j, \xi) \in \mathbb{R}^{d_{\boldsymbol{y}} \times d_{\boldsymbol{y}}},$$

$$\hat{\boldsymbol{B}} = \left[ \mathbb{1}_{ji} \hat{\boldsymbol{B}}_i^{\psi} + \bar{p}_{ji} \hat{\boldsymbol{B}}_j^{\varphi} \right]_{ji} \in \mathbb{R}^{nd_{\boldsymbol{\lambda}} \times nd_{\boldsymbol{y}}},$$

$$\hat{\boldsymbol{B}}_i^{\psi} = \frac{1}{|\zeta_i|} \sum_{\xi \in \zeta_i} \partial_{\boldsymbol{\lambda}_i} h_i(\boldsymbol{y}_i^*; \boldsymbol{\lambda}_i, \xi) \in \mathbb{R}^{d_{\boldsymbol{\lambda}} \times d_{\boldsymbol{y}}}, \quad \hat{\boldsymbol{B}}_j^{\varphi} = \frac{1}{|\zeta_j|} \sum_{\xi \in \zeta_j} \partial_{\boldsymbol{\lambda}_j} \varphi_{ji}(\boldsymbol{y}_j^*; \boldsymbol{\lambda}_j, \xi) \in \mathbb{R}^{d_{\boldsymbol{\lambda}} \times d_{\boldsymbol{y}}}.$$

#### A.3.2 APPROXIMATION BY NEUMANN SERIES

With Assumption 4, we have $\left\| \bar{\boldsymbol{A}} \right\|_2 < 1$ We can thus approximate $(\boldsymbol{I} - \bar{\boldsymbol{A}})^{-1}$ by the truncated Neumann series up to the $M$-th term as

$$(\boldsymbol{I} - \bar{\boldsymbol{A}})^{-1} = \sum_{m=0}^{\infty} \bar{\boldsymbol{A}}^m \approx \sum_{m=0}^{M-1} \bar{\boldsymbol{A}}^m.$$

The approximation of the hyper-gradient could be expressed as

$$\mathrm{d}_{\boldsymbol{\lambda}} F(\boldsymbol{x}^*, \boldsymbol{\lambda}) \approx \bar{\boldsymbol{B}} \sum_{m=0}^{M-1} \bar{\boldsymbol{A}}^m \boldsymbol{c}_i^{\boldsymbol{y}} + \boldsymbol{c}_j^{\boldsymbol{\lambda}}.$$

By replacing $\bar{\boldsymbol{A}}$ and $\bar{\boldsymbol{B}}$ with the estimators $\hat{\boldsymbol{A}}$ and $\hat{\boldsymbol{B}}$, we have

$$\mathrm{d}_{\boldsymbol{\lambda}} \bar{F}(\boldsymbol{x}^*, \boldsymbol{\lambda}) \approx \sum_{m=0}^{M-1} \hat{\boldsymbol{B}}^{(2m)} \prod_{s=0}^{m-1} \hat{\boldsymbol{A}}^{(2s+1)} \boldsymbol{c}^{\boldsymbol{y}} + \boldsymbol{c}^{\boldsymbol{\lambda}},$$

where $\hat{\boldsymbol{A}}^{(2s+1)}$ and $\hat{\boldsymbol{B}}^{(2m)}$ denotes the estimators at the $2s+1$-th and the $2m$-th step of the communication round, respectively. In this estimator, we estimate $\bar{\boldsymbol{A}}$ in the odd-numbered steps and estimate $\bar{\boldsymbol{B}}$ in the even-numbered steps of the communication round, respectively.

### A.4 HYPER-GRADIENT PUSH (HGP)

We now present our proposed method, hyper-gradient push (HGP), which is a modified version of the recurrent backpropagation. HGP can run even on stochastic and directed networks while enjoying the same order of communication efficiency as SGP. In HGP, we adopt Assumptions 6 and 7, and assume that $\{\bar{\delta}_{j \to i} = \mathbb{E}_{\delta}[\delta_{j \to i}]\}_{j,i}$ and $\{\bar{p}_{ji}\}_{j,i}$ are known.

The idea of HGP is to use $\delta_{ij}\frac{\bar{p}_{ji}}{\bar{\delta}_{ij}}$ instead of $p_{ji}$ in $\hat{A}$ and $\hat{B}$ as follows.

$$\hat{A} = \left[\mathbb{1}_{ji}\hat{A}_i^\psi + \delta_{i\to j}\frac{\bar{p}_{ji}}{\bar{\delta}_{i\to j}}\hat{A}_j^\varphi\right]_{ji},$$

$$\hat{B} = \left[\mathbb{1}_{ji}\hat{B}_i^\psi + \delta_{i\to j}\frac{\bar{p}_{ji}}{\bar{\delta}_{i\to j}}\hat{B}_j^\varphi\right]_{ji}.$$

Under Assumption 5 where $\delta$ and $\zeta$ are independent, these are the unbiased estimators because

$$\mathbb{E}_{\delta,\zeta}\left[\hat{A}\right] = \left[\mathbb{1}_{ji}\underbrace{\mathbb{E}_{\zeta_i}\left[\hat{A}_i^\psi\right]}_{=\bar{A}_i^\psi} + \underbrace{\mathbb{E}_{\delta_{i\to j}}\left[\delta_{i\to j}\frac{\bar{p}_{ji}}{\bar{\delta}_{i\to j}}\right]}_{\bar{\delta}_{i\to j}\frac{\bar{p}_{ji}}{\bar{\delta}_{i\to j}}=\bar{p}_{ji}}\underbrace{\mathbb{E}_{\zeta_j}\left[\hat{A}_j^\varphi\right]}_{=\bar{A}_j^\varphi}\right]_{ji}$$

$$= \left[\mathbb{1}_{ji}\bar{A}_i^\psi + \bar{p}_{ji}\bar{A}_j^\varphi\right]_{ji} = \bar{A},$$

$$\mathbb{E}_{\delta,\zeta}\left[\hat{B}\right] = \left[\mathbb{1}_{ji}\underbrace{\mathbb{E}_{\zeta_i}\left[\hat{B}_i^\psi\right]}_{=\bar{B}_i} + \underbrace{\mathbb{E}_{\delta_{i\to j}}\left[\delta_{i\to j}\frac{\bar{p}_{ji}}{\bar{\delta}_{i\to j}}\right]}_{\bar{\delta}_{i\to j}\frac{\bar{p}_{ji}}{\bar{\delta}_{i\to j}}=\bar{p}_{ji}}\underbrace{\mathbb{E}_{\zeta_j}\left[\hat{B}_j^\varphi\right]}_{=\bar{B}_j^\varphi}\right]_{ji}$$

$$= \left[\mathbb{1}_{ji}\bar{B}_i^\psi + \bar{p}_{ji}\bar{B}_j^\varphi\right]_{ji} = \bar{B}.$$

Recall that the hyper-gradient can be approximated as

$$d_{\lambda_j}F(x^*,\lambda) \approx \sum_{m=0}^{M-1}\sum_k\langle\bar{B}\rangle_{jk}\sum_i\langle\bar{A}^m\rangle_{ki}c_i^y + c_j^\lambda. \tag{15}$$

By replacing the expectation with the above estimators $\hat{A}$ and $\hat{B}$, we have

$$\widehat{d_{\lambda_j}}F(x^*,\lambda) = \sum_{m=0}^{M-1}\sum_k\left\langle\hat{B}^{(2m)}\right\rangle_{jk}\sum_i\left\langle\prod_{s=0}^{m-1}\hat{A}^{(2s+1)}\right\rangle_{ki}c_i^y + c_j^\lambda. \tag{16}$$

Let $u_k^{(m)} = \sum_i\left\langle\prod_{s=0}^{m-1}\hat{A}^{(2s+1)}\right\rangle_{ki}c_i^y$. We note that $u_k^{(m+1)}$ can be computed recursively as

$$u_k^{(m+1)} = \sum_{k'}\left\langle\hat{A}^{(2m+1)}\right\rangle_{kk'}u_{k'}^{(m)}.$$

By using this fact, we can rewrite the estimator as

$$\widehat{d_{\lambda_j}}F(x^*,\lambda) = \sum_{m=0}^{M-1}\sum_k\left\langle\hat{B}^{(2m)}\right\rangle_{jk}\sum_{k'}\left\langle\hat{A}^{(2m-1)}\right\rangle_{kk'}\underbrace{\sum_i\left\langle\prod_{s=0}^{m-2}\hat{A}^{(2s+1)}\right\rangle_{k'i}c_i^y}_{u_{k'}^{(m-1)}} + c_j^\lambda$$

$$= \sum_{m=0}^{M-1}\sum_k\left\langle\hat{B}^{(2m)}\right\rangle_{jk}\underbrace{\sum_{k'}\left\langle\hat{A}^{(2m-1)}\right\rangle_{kk'}u_{k'}^{(m-1)}}_{u_k^{(m)}} + c_j^\lambda.$$

We can then derive the proposed algorithm, hyper-gradient push, as follows:

---

**Hyper-Gradient Push (HGP)**

$$\boldsymbol{u}_j^{(0)} \leftarrow \boldsymbol{c}_j^{\boldsymbol{y}}, \boldsymbol{v}_j^{(0)} \leftarrow \boldsymbol{c}_j^{\boldsymbol{\lambda}}$$

$$\begin{cases} \boldsymbol{v}_j^{(m+1)} \leftarrow \boldsymbol{v}_j^{(m)} + \sum_k \left\langle \hat{\boldsymbol{B}}^{(2m)} \right\rangle_{jk} \boldsymbol{u}_k^{(m)} \\ \boldsymbol{u}_k^{(m+1)} \leftarrow \sum_{k'} \left\langle \hat{\boldsymbol{A}}^{(2m+1)} \right\rangle_{kk'} \boldsymbol{u}_{k'}^{(m)} \end{cases}$$

$$\text{for } m = 0, 1, 2, \ldots, M-1$$

$$\widehat{\mathrm{d}_{\boldsymbol{\lambda}_i}} F(\boldsymbol{x}^*, \boldsymbol{\lambda}) \leftarrow \boldsymbol{v}_j^{(M)}$$

---

In HGP, the estimator could be obtained after the $2M$ rounds of communication. In each round of the communication, the clients communicate $\boldsymbol{u}_k^{(m)} \in \mathbb{R}^{d_{\boldsymbol{y}}}$ which is $O(d_{\boldsymbol{y}})$ parameters only, the same as the standard communication for SGP update.

## B  VARIANCE REDUCTION

We now introduce the variance-reduced version of HGP. The naive HGP above suffers from the large variance because of $\delta_{j\rightarrow i}^{(m)}/\bar{\delta}_{j\rightarrow i}$; this term can take a value far larger than one when $\bar{\delta}_{j\rightarrow i}$ is small. The multiplication of such values induces high variance.

Recall that, in HGP, we aim at approximating the estimator

$$\mathrm{d}_{\boldsymbol{\lambda}_j} F(\boldsymbol{x}^*, \boldsymbol{\lambda}) \approx \sum_{m=0}^{M-1} \sum_k \langle \bar{\boldsymbol{B}} \rangle_{jk} \sum_i \langle \bar{\boldsymbol{A}}^m \rangle_{ki} \boldsymbol{c}_i^{\boldsymbol{y}} + \boldsymbol{c}_j^{\boldsymbol{\lambda}}.$$

With $\boldsymbol{v}_j^{(0)} \leftarrow 0, \boldsymbol{u}_k^{(0)} \leftarrow C_k^{\boldsymbol{y}}$, HGP computes the first term of the right-hand-side by iterating

$$\boldsymbol{v}_j^{(m+1)} \leftarrow \boldsymbol{v}_j^{(m)} + \sum_k \langle \bar{\boldsymbol{B}} \rangle_{jk} \boldsymbol{u}_k^{(m)},$$

$$\boldsymbol{u}_k^{(m+1)} \leftarrow \sum_{k'} \langle \bar{\boldsymbol{A}} \rangle_{kk'} \boldsymbol{u}_{k'}^{(m)},$$

where $u_k^{(m+1)}$ is equivalent to $\sum_{k'} \langle \boldsymbol{A}^{m+1} \rangle_{kk'} \boldsymbol{c}_{k'}^{\boldsymbol{y}}$. We can also consider another way of computing the first term. With $\boldsymbol{v}_j^{(0)} \leftarrow \boldsymbol{0}_{d_{\boldsymbol{\lambda}}}, \boldsymbol{w}_k^{(0)} \leftarrow \boldsymbol{c}_k^{\boldsymbol{y}}$, we can compute

$$\boldsymbol{v}_j^{(m+1)} \leftarrow \sum_k \langle \bar{\boldsymbol{B}} \rangle_{jk} \boldsymbol{w}_k^{(m)},$$

$$\boldsymbol{w}_k^{(m+1)} \leftarrow \sum_{k'} \langle \bar{\boldsymbol{A}} \rangle_{kk'} \boldsymbol{w}_{k'}^{(m)} + \boldsymbol{c}_k^{\boldsymbol{y}},$$

where $\boldsymbol{w}_k^{(m+1)}$ is equivalent to $\sum_{m'=0}^{m+1} \sum_{k'} \left\langle \bar{\boldsymbol{A}}^{m'} \right\rangle_{kk'} \boldsymbol{c}_{k'}^{\boldsymbol{y}} = \sum_{m'=0}^{m+1} \boldsymbol{u}_k^{(m')} = \boldsymbol{w}_k^{(m)} + \boldsymbol{u}_k^{(m+1)}$.

By combining the above two formulas, we can derive the general expression of HGP as

$$\boldsymbol{v}_j^{(m+1)} \leftarrow \alpha \left( \boldsymbol{v}_j^{(m)} + \sum_k \langle \bar{\boldsymbol{B}} \rangle_{jk} \boldsymbol{u}_k^{(m)} \right) + (1-\alpha) \sum_k \langle \bar{\boldsymbol{B}} \rangle_{jk} \boldsymbol{w}_k^{(m)},$$

$$\boldsymbol{u}_k^{(m+1)} \leftarrow \sum_{k'} \langle \bar{\boldsymbol{A}} \rangle_{kk'} \boldsymbol{u}_{k'}^{(m)},$$

$$\boldsymbol{w}_k^{(m+1)} \leftarrow \beta \left( \sum_{k'} \langle \bar{\boldsymbol{A}} \rangle_{kk'} \boldsymbol{w}_{k'}^{(m)} + \boldsymbol{c}_k^{\boldsymbol{y}} \right) + (1-\beta) \left( \boldsymbol{w}_k^{(m)} + \boldsymbol{u}_k^{(m+1)} \right),$$

where $\alpha, \beta \in [0, 1]$ are the interpolation weights. By replacing $\bar{\boldsymbol{A}}, \bar{\boldsymbol{B}}$ by the empirical estimates $\hat{\boldsymbol{A}}$, $\hat{\boldsymbol{B}}$, we obtain the general expression of HGP as follows.

---

**General HGP for Variance Reduction**

$\boldsymbol{v}_j^{(0)} \leftarrow \boldsymbol{0}_{d_{\boldsymbol{\lambda}}}, \, \boldsymbol{u}_j^{(0)} \leftarrow \boldsymbol{c}_j^{\boldsymbol{y}}, \, \boldsymbol{w}_j^{(0)} \leftarrow \boldsymbol{c}_j^{\boldsymbol{y}}$

$$
\begin{cases}
\boldsymbol{v}_j^{(m+1)} \leftarrow \alpha \left( \boldsymbol{v}_j^{(m)} + \sum_k \left\langle \hat{\boldsymbol{B}}^{(2m)} \right\rangle_{jk} \boldsymbol{u}_k^{(m)} \right) + (1 - \alpha) \sum_k \left\langle \hat{\boldsymbol{B}}^{(2m)} \right\rangle_{jk} \boldsymbol{w}_k^{(m)} \\
\boldsymbol{u}_k^{(m+1)} \leftarrow \sum_{k'} \left\langle \hat{\boldsymbol{A}}^{(2m+1)} \right\rangle_{kk'} \boldsymbol{u}_{k'}^{(m)} \\
\boldsymbol{w}_k^{(m+1)} \leftarrow \beta \left( \sum_{k'} \left\langle \hat{\boldsymbol{A}}^{(2m+1)} \right\rangle_{kk'} \boldsymbol{w}_{k'}^{(m)} + \boldsymbol{c}_k^{\boldsymbol{y}} \right) + (1 - \beta) \left( \boldsymbol{w}_k^{(m)} + \boldsymbol{u}_k^{(m+1)} \right),
\end{cases}
$$
$\qquad$ for $m = 0, 1, 2, \ldots, M - 1$

$\widehat{\mathrm{d}_{\boldsymbol{\lambda}_i} F}(\boldsymbol{x}^*, \boldsymbol{\lambda}) \leftarrow \boldsymbol{v}_j^{(M)} + \boldsymbol{c}_j^{\boldsymbol{\lambda}}$

---

We note that this general HGP is the weighted average of the two different estimation algorithms, which results in an estimator with a smaller variance. That is, by choosing $\alpha, \beta \in [0, 1]$ appropriately, we can obtain an estimate of the hyper-gradient with a smaller variance. From the computational perspective, this general HGP has properties similar to the original HGP: it can be computed even on stochastic and directed networks; the estimator could be obtained after the $2M$ rounds of communication; and the clients communicate $O(d_{\boldsymbol{y}})$ parameters in each iteration.

## C  ESTIMATION ERROR OF HYPER-GRADIENT

In the following, we assume that the derivatives of $g_i$ and $h_j$ are bounded.

**Assumption 9.** $\exists \eta_A \in (0, 1), \eta_B \in (0, \infty)$ such that $\forall \xi$ and $\forall i, j$,

$$
\max \left\{ \sup_{\boldsymbol{y}_i, \boldsymbol{\lambda}_i, \xi} \left\| \partial_{\boldsymbol{y}_i} g_i(\boldsymbol{y}_i, \boldsymbol{\lambda}_i, \xi) \right\|_2, \sup_{\boldsymbol{y}_j, \boldsymbol{\lambda}_j, \xi} \left\| \partial_{\boldsymbol{y}_j} h_j(\boldsymbol{y}_j, \boldsymbol{\lambda}_j, \xi) \right\|_2 \right\} \leq \frac{\eta_A}{2 \sum_{i,j} \frac{\bar{p}_{ji}}{\bar{\delta}_{i \to j}}},
$$

$$
\max \left\{ \sup_{\boldsymbol{y}_i, \boldsymbol{\lambda}_i, \xi} \left\| \partial_{\boldsymbol{\lambda}_i} g_i(\boldsymbol{y}_i, \boldsymbol{\lambda}_i, \xi) \right\|_2, \sup_{\boldsymbol{y}_j, \boldsymbol{\lambda}_j, \xi} \left\| \partial_{\boldsymbol{\lambda}_j} h_j(\boldsymbol{y}_j, \boldsymbol{\lambda}_j, \xi) \right\|_2 \right\} \leq \frac{\eta_B}{2 \sum_{i,j} \frac{\bar{p}_{ji}}{\bar{\delta}_{i \to j}}}.
$$

Recall that $\sum_{i,j} \frac{\bar{p}_{ji}}{\bar{\delta}_{i \to j}} \geq n$ by the properties $\sum_{i=1}^n \bar{p}_{ji} = 1, \bar{\delta}_{i \to j} \in [0, 1]$. Assumption 9 implies

$$
\left\| \bar{\boldsymbol{A}} \right\|_2 \leq \sum_i \sup_{\boldsymbol{y}_i, \boldsymbol{\lambda}_i, \xi} \left\| \partial_{\boldsymbol{y}_i} g_i(\boldsymbol{y}_i, \boldsymbol{\lambda}_i, \xi) \right\|_2 + \sum_{i,j} \bar{p}_{ji} \sup_{\boldsymbol{y}_j, \boldsymbol{\lambda}_j, \xi} \left\| \partial_{\boldsymbol{y}_j} h_j(\boldsymbol{y}_j, \boldsymbol{\lambda}_i, \xi) \right\|_2
$$

$$
\leq \left( n + \sum_{i,j} \bar{p}_{ji} \right) \frac{\eta_A}{2 \sum_{i,j} \frac{\bar{p}_{ji}}{\bar{\delta}_{i \to j}}} \leq \eta_A,
$$

$$
\left\| \hat{\boldsymbol{A}} \right\|_2 \leq \sum_i \sup_{\boldsymbol{y}_i, \boldsymbol{\lambda}_i, \xi} \left\| \partial_{\boldsymbol{y}_i} g_i(\boldsymbol{y}_i, \boldsymbol{\lambda}_i, \xi) \right\|_2 + \sum_{i,j} \frac{\bar{p}_{ji}}{\bar{\delta}_{i \to j}} \sup_{\boldsymbol{y}_j, \boldsymbol{\lambda}_j, \xi} \left\| \partial_{\boldsymbol{y}_j} h_j(\boldsymbol{y}_j, \boldsymbol{\lambda}_i, \xi) \right\|_2
$$

$$
\leq \left( n + \sum_{i,j} \frac{\bar{p}_{ji}}{\bar{\delta}_{i \to j}} \right) \frac{\eta_A}{2 \sum_{i,j} \frac{\bar{p}_{ji}}{\bar{\delta}_{i \to j}}} \leq \eta_A,
$$

$$
\left\| \bar{\boldsymbol{B}} \right\|_2 \leq \sum_i \sup_{\boldsymbol{y}_i, \boldsymbol{\lambda}_i, \xi} \left\| \partial_{\boldsymbol{\lambda}_i} g_i(\boldsymbol{y}_i, \boldsymbol{\lambda}_i, \xi) \right\|_2 + \sum_{i,j} \bar{p}_{ji} \sup_{\boldsymbol{y}_j, \boldsymbol{\lambda}_j, \xi} \left\| \partial_{\boldsymbol{\lambda}_j} h_j(\boldsymbol{y}_j, \boldsymbol{\lambda}_i, \xi) \right\|_2
$$

$$
\leq \left( n + \sum_{i,j} \bar{p}_{ji} \right) \frac{\eta_B}{2 \sum_{i,j} \frac{\bar{p}_{ji}}{\bar{\delta}_{i \to j}}} \leq \eta_B,
$$

$$
\left\| \hat{\boldsymbol{B}} \right\|_2 \leq \sum_i \sup_{\boldsymbol{y}_i, \boldsymbol{\lambda}_i, \xi} \left\| \partial_{\boldsymbol{\lambda}_i} g_i(\boldsymbol{y}_i, \boldsymbol{\lambda}_i, \xi) \right\|_2 + \sum_{i,j} \frac{\bar{p}_{ji}}{\bar{\delta}_{i \to j}} \sup_{\boldsymbol{\lambda}_j, \boldsymbol{\lambda}_j, \xi} \left\| \partial_{\boldsymbol{y}_j} h_j(\boldsymbol{y}_j, \boldsymbol{\lambda}_i, \xi) \right\|_2
$$

$$
\leq \left( n + \sum_{i,j} \frac{\bar{p}_{ji}}{\bar{\delta}_{i \to j}} \right) \frac{\eta_B}{2 \sum_{i,j} \frac{\bar{p}_{ji}}{\bar{\delta}_{i \to j}}} \leq \eta_B.
$$

## C.1 PRELIMINARY LEMMAS

In this section, we present a few preliminary lemmas we use in the proof of the theorems.

We recall that we can express the general HGP using the concatenated vectors and matrices as

$$\boldsymbol{v}^{(m+1)} = \alpha \left( \boldsymbol{v}^{(m)} + \hat{\boldsymbol{B}}^{(2m)} \boldsymbol{u}^{(m)} \right) + (1 - \alpha) \hat{\boldsymbol{B}}^{(2m)} \boldsymbol{w}^{(m)}, \tag{17}$$

$$\boldsymbol{u}^{(m+1)} = \hat{\boldsymbol{A}}^{(2m+1)} \boldsymbol{u}^{(m)}, \tag{18}$$

$$\boldsymbol{w}^{(m+1)} = \beta \left( \hat{\boldsymbol{A}}^{(2m+1)} \boldsymbol{w}^{(m)} + \boldsymbol{c}^{\boldsymbol{y}} \right) + (1 - \beta) \left( \boldsymbol{w}^{(m)} + \boldsymbol{u}^{(m+1)} \right). \tag{19}$$

with the initial conditions $\boldsymbol{v}^{(0)} \leftarrow 0$, $\boldsymbol{u}^{(0)} \leftarrow \boldsymbol{c}^{\boldsymbol{y}}$, and $\boldsymbol{w}^{(0)} \leftarrow \boldsymbol{c}^{\boldsymbol{y}}$.

The following lemmas show explicit formula of $\boldsymbol{v}$ and $\boldsymbol{w}$ and their decomposition.

**Lemma 2** (Explicit Formula of $\boldsymbol{w}$).

$$\boldsymbol{w}^{(M)} = \prod_{m=0}^{M-1} \left( (1 - \beta)\boldsymbol{I} + \beta \hat{\boldsymbol{A}}^{(2m+1)} \right) \boldsymbol{c}^{\boldsymbol{y}}$$

$$+ \sum_{i=1}^{M} \left[ \prod_{m=i}^{M-1} \left( (1 - \beta)\boldsymbol{I} + \beta \hat{\boldsymbol{A}}^{(2m+1)} \right) \right] \left( \beta \boldsymbol{I} + (1 - \beta) \left[ \prod_{m=0}^{i-1} \hat{\boldsymbol{A}}^{(2m+1)} \right] \right) \boldsymbol{c}^{\boldsymbol{y}}, \tag{20}$$

where we define $\prod_{m \in \emptyset} (\cdot)_m = 1$ so that $\prod_{m=M}^{M-1} (\cdot)_m = \boldsymbol{I}$.

*Proof.* We prove the claim by induction. We first recall that

$$\boldsymbol{u}^{(M)} = \prod_{m=0}^{M-1} \hat{\boldsymbol{A}}^{(2m+1)} \boldsymbol{c}^{\boldsymbol{y}}. \tag{21}$$

By setting $m = 0$ in (19), we have

$$\boldsymbol{w}^{(1)} = \beta \left( \hat{\boldsymbol{A}}^{(1)} \boldsymbol{w}^{(0)} + \boldsymbol{c}^{\boldsymbol{y}} \right) + (1 - \beta) \left( \boldsymbol{w}^{(0)} + \boldsymbol{u}^{(1)} \right)$$

$$= \beta \left( \hat{\boldsymbol{A}}^{(1)} \boldsymbol{c}^{\boldsymbol{y}} + \boldsymbol{c}^{\boldsymbol{y}} \right) + (1 - \beta) \left( \boldsymbol{c}^{\boldsymbol{y}} + \hat{\boldsymbol{A}}^{(1)} \boldsymbol{c}^{\boldsymbol{y}} \right)$$

$$= \boldsymbol{c}^{\boldsymbol{y}} + \hat{\boldsymbol{A}}^{(1)} \boldsymbol{c}^{\boldsymbol{y}}.$$

By setting $M = 1$ in (20), we also have

$$\boldsymbol{w}^{(1)} = \left( (1 - \beta)\boldsymbol{I} + \beta \hat{\boldsymbol{A}}^{(1)} \right) \boldsymbol{c}^{\boldsymbol{y}} + \left( \beta \boldsymbol{I} + (1 - \beta) \hat{\boldsymbol{A}}^{(1)} \right) \boldsymbol{c}^{\boldsymbol{y}} = \boldsymbol{c}^{\boldsymbol{y}} + \hat{\boldsymbol{A}}^{(1)} \boldsymbol{c}^{\boldsymbol{y}},$$

which confirms that (20) is valid when $M = 1$.

Now, suppose that the statement is true for some $M \geq 1$. Then, by (19),

$$\boldsymbol{w}^{(M+1)} = \beta \left( \hat{\boldsymbol{A}}^{(2M+1)} w^{(M)} + C^{\boldsymbol{y}} \right) + (1 - \beta) \left( \boldsymbol{w}^{(M)} + \boldsymbol{u}^{(M+1)} \right)$$

$$= \beta \boldsymbol{c}^{\boldsymbol{y}} + (1 - \beta) \left[ \prod_{m=0}^{M} \hat{\boldsymbol{A}}^{(2m+1)} \right] \boldsymbol{c}^{\boldsymbol{y}} + \left( (1 - \beta)\boldsymbol{I} + \beta \hat{\boldsymbol{A}}^{(2M+1)} \right) \boldsymbol{w}^{(M)}$$

$$= \beta \boldsymbol{c^y} + (1 - \beta) \left[ \prod_{m=0}^{M} \hat{\boldsymbol{A}}^{(2m+1)} \right] \boldsymbol{c^y}$$

$$+ \left( (1 - \beta)\boldsymbol{I} + \beta \hat{\boldsymbol{A}}^{(2M+1)} \right) \left[ \prod_{m=0}^{M-1} \left( (1 - \beta)\boldsymbol{I} + \beta \hat{\boldsymbol{A}}^{(2m+1)} \right) \right] \boldsymbol{c^y}$$

$$+ \left( (1 - \beta)\boldsymbol{I} + \beta \hat{\boldsymbol{A}}^{(2M+1)} \right) \sum_{i=1}^{M} \left[ \prod_{m=i}^{M-1} \left( (1 - \beta)\boldsymbol{I} + \beta \hat{\boldsymbol{A}}^{(2m+1)} \right) \right] \left( \beta \boldsymbol{I} + (1 - \beta) \left[ \prod_{m=0}^{i-1} \hat{\boldsymbol{A}}^{(2m+1)} \right] \right) \boldsymbol{c^y}$$

$$= \left[ \prod_{m=0}^{M} \left( (1 - \beta)\boldsymbol{I} + \beta \hat{\boldsymbol{A}}^{(2m+1)} \right) \right] \boldsymbol{c^y}$$

$$+ 1 \times \left( \beta \boldsymbol{I} + (1 - \beta) \left[ \prod_{m=0}^{M} \hat{\boldsymbol{A}}^{(2m+1)} \right] \right) \boldsymbol{c^y}$$

$$+ \sum_{i=1}^{M} \left[ \prod_{m=i}^{M} \left( (1 - \beta)\boldsymbol{I} + \beta \hat{\boldsymbol{A}}^{(2m+1)} \right) \right] \left( \beta \boldsymbol{I} + (1 - \beta) \prod_{m=0}^{i-1} \hat{\boldsymbol{A}}^{(2m+1)} \right) \boldsymbol{c^y}$$

$$= \left[ \prod_{m=0}^{M} \left( (1 - \beta)\boldsymbol{I} + \beta \hat{\boldsymbol{A}}^{(2m+1)} \right) \right] \boldsymbol{c^y}$$

$$+ \sum_{i=1}^{M+1} \left[ \prod_{m=i}^{M} \left( (1 - \beta)\boldsymbol{I} + \beta \hat{\boldsymbol{A}}^{(2m+1)} \right) \right] \left( \beta \boldsymbol{I} + (1 - \beta) \prod_{m=0}^{i-1} \hat{\boldsymbol{A}}^{(2m+1)} \right) \boldsymbol{c^y},$$

where the last line follows from the fact that $\prod_{i=M+1}^{M} (\cdot)_m = \boldsymbol{I}$. $\qquad \square$

**Lemma 3** (Decomposition of $\boldsymbol{w}$).

$$\boldsymbol{w}^{(M)} - \sum_{i=0}^{M} \bar{\boldsymbol{A}}^i \boldsymbol{c^y}$$

$$= \left( \sum_{i=0}^{M-1} \hat{\boldsymbol{L}}_1^{(i,M)} (\hat{\boldsymbol{A}}^{(2i+1)} - \bar{\boldsymbol{A}}) \boldsymbol{R}_1^{(i)} + \hat{\boldsymbol{L}}_2^{(i,M)} (\hat{\boldsymbol{A}}^{(2i+1)} - \bar{\boldsymbol{A}}) \bar{\boldsymbol{A}}^i \right) \boldsymbol{c^y}, \qquad (22)$$

where

$$\hat{\boldsymbol{L}}_1^{(i,M)} = \beta \left[ \prod_{m=i+1}^{M-1} \left( (1 - \beta)\boldsymbol{I} + \beta \hat{\boldsymbol{A}}^{(2m+1)} \right) \right],$$

$$\hat{\boldsymbol{L}}_2^{(i,M)} = (1 - \beta) \sum_{j=i+1}^{M} \left[ \prod_{m=j}^{M-1} \left( (1 - \beta)\boldsymbol{I} + \beta \hat{\boldsymbol{A}}^{(2m+1)} \right) \right] \left[ \prod_{m=i+1}^{j-1} \hat{\boldsymbol{A}}^{(2m+1)} \right],$$

$$\boldsymbol{R}_1^{(i)} = \left( (1 - \beta)\boldsymbol{I} + \beta \bar{\boldsymbol{A}} \right)^i + \sum_{j=1}^{i} \left( (1 - \beta)\boldsymbol{I} + \beta \bar{\boldsymbol{A}} \right)^{i-j} \left( \beta \boldsymbol{I} + (1 - \beta) \bar{\boldsymbol{A}}^j \right).$$

*Proof.* We first recall that, as the corollary of Lemma 2,

$$\sum_{i=0}^{M} \bar{\boldsymbol{A}}^i \boldsymbol{c^y} = \left( (1 - \beta)\boldsymbol{I} + \beta \bar{\boldsymbol{A}} \right)^M \boldsymbol{c^y} + \sum_{i=1}^{M} \left( (1 - \beta)\boldsymbol{I} + \beta \bar{\boldsymbol{A}} \right)^{M-i} \left( \beta \boldsymbol{I} + (1 - \beta) \bar{\boldsymbol{A}}^i \right) \boldsymbol{c^y}.$$

By using Lemma 2, we can expand the difference as

$$
\boldsymbol{w}^{(M)} - \sum_{i=0}^{M} \bar{\boldsymbol{A}}^i \boldsymbol{c}^{\boldsymbol{y}}
$$

$$
= \left( \prod_{m=0}^{M-1} \left( (1-\beta)\boldsymbol{I} + \beta \hat{\boldsymbol{A}}^{(2m+1)} \right) - \left( (1-\beta)\boldsymbol{I} + \beta \bar{\boldsymbol{A}} \right)^M \right) \boldsymbol{c}^{\boldsymbol{y}}
$$

$$
+ \sum_{i=1}^{M} \left( \left[ \prod_{m=i}^{M-1} \left( (1-\beta)\boldsymbol{I} + \beta \hat{\boldsymbol{A}}^{(2m+1)} \right) \right] - \left( (1-\beta)\boldsymbol{I} + \beta \bar{\boldsymbol{A}} \right)^{M-i} \right) \left( \beta \boldsymbol{I} + (1-\beta)\bar{\boldsymbol{A}}^i \right) \boldsymbol{c}^{\boldsymbol{y}}
$$

$$
+ \sum_{i=1}^{M} \left[ \prod_{m=i}^{M-1} \left( (1-\beta)\boldsymbol{I} + \beta \hat{\boldsymbol{A}}^{(2m+1)} \right) \right] (1-\beta) \left( \prod_{m=0}^{i-1} \hat{\boldsymbol{A}}^{(2m+1)} - \bar{\boldsymbol{A}}^i \right) \boldsymbol{c}^{\boldsymbol{y}}
$$

$$
= \sum_{i=0}^{M-1} \left[ \prod_{m=i+1}^{M-1} \left( (1-\beta)\boldsymbol{I} + \beta \hat{\boldsymbol{A}}^{(2m+1)} \right) \right] \beta \left( \hat{\boldsymbol{A}}^{(2i+1)} - \bar{\boldsymbol{A}} \right) \left( (1-\beta)\boldsymbol{I} + \beta \bar{\boldsymbol{A}} \right)^i \boldsymbol{c}^{\boldsymbol{y}}
$$

$$
+ \sum_{j=1}^{M-1} \sum_{i=j}^{M-1} \left[ \prod_{m=i+1}^{M-1} \left( (1-\beta)\boldsymbol{I} + \beta \hat{\boldsymbol{A}}^{(2m+1)} \right) \right] \beta \left( \hat{\boldsymbol{A}}^{(2i+1)} - \bar{\boldsymbol{A}} \right) \left( (1-\beta)\boldsymbol{I} + \beta \bar{\boldsymbol{A}} \right)^{i-j}
$$

$$
\times \left( \beta \boldsymbol{I} + (1-\beta)\bar{\boldsymbol{A}}^j \right) \boldsymbol{c}^{\boldsymbol{y}}
$$

$$
+ \sum_{j=1}^{M} \left[ \prod_{m=j}^{M-1} \left( (1-\beta)\boldsymbol{I} + \beta \hat{\boldsymbol{A}}^{(2m+1)} \right) \right] (1-\beta) \sum_{i=0}^{j-1} \left[ \prod_{m=i+1}^{j-1} \hat{\boldsymbol{A}}^{(2m+1)} \right] \left( \hat{\boldsymbol{A}}^{(2i+1)} - \bar{\boldsymbol{A}} \right) \bar{\boldsymbol{A}}^i \boldsymbol{c}^{\boldsymbol{y}}
$$

$$
= \sum_{i=0}^{M-1} \left[ \prod_{m=i+1}^{M-1} \left( (1-\beta)\boldsymbol{I} + \beta \hat{\boldsymbol{A}}^{(2m+1)} \right) \right] \beta \left( \hat{\boldsymbol{A}}^{(2i+1)} - \bar{\boldsymbol{A}} \right) \left( (1-\beta)\boldsymbol{I} + \beta \bar{\boldsymbol{A}} \right)^i \boldsymbol{c}^{\boldsymbol{y}}
$$

$$
+ \sum_{i=1}^{M-1} \left[ \prod_{m=i+1}^{M-1} \left( (1-\beta)\boldsymbol{I} + \beta \hat{\boldsymbol{A}}^{(2m+1)} \right) \right] \beta \left( \hat{\boldsymbol{A}}^{(2i+1)} - \bar{\boldsymbol{A}} \right)
$$

$$
\times \sum_{j=1}^{i} \left( (1-\beta)\boldsymbol{I} + \beta \bar{\boldsymbol{A}} \right)^{i-j} \left( \beta \boldsymbol{I} + (1-\beta)\bar{\boldsymbol{A}}^j \right) \boldsymbol{c}^{\boldsymbol{y}}
$$

$$
+ \sum_{i=0}^{M-1} \sum_{j=i+1}^{M} \left[ \prod_{m=j}^{M-1} \left( (1-\beta)\boldsymbol{I} + \beta \hat{\boldsymbol{A}}^{(2m+1)} \right) \right] (1-\beta) \left[ \prod_{m=i+1}^{j-1} \hat{\boldsymbol{A}}^{(2m+1)} \right] \left( \hat{\boldsymbol{A}}^{(2i+1)} - \bar{\boldsymbol{A}} \right) \bar{\boldsymbol{A}}^i \boldsymbol{c}^{\boldsymbol{y}}
$$

$$
= \sum_{i=0}^{M-1} \beta \underbrace{\left[ \prod_{m=i+1}^{M-1} \left( (1-\beta)\boldsymbol{I} + \beta \hat{\boldsymbol{A}}^{(2m+1)} \right) \right] \left( \hat{\boldsymbol{A}}^{(2i+1)} - \bar{\boldsymbol{A}} \right)}_{=\hat{\boldsymbol{L}}_1^{(i,M)}}
$$

$$
\times \underbrace{\left( \left( (1-\beta)\boldsymbol{I} + \beta \bar{\boldsymbol{A}} \right)^i + \sum_{j=1}^{i} \left( (1-\beta)\boldsymbol{I} + \beta \bar{\boldsymbol{A}} \right)^{i-j} \left( \beta \boldsymbol{I} + (1-\beta)\bar{\boldsymbol{A}}^j \right) \right) \boldsymbol{c}^{\boldsymbol{y}}}_{=\boldsymbol{R}_1^{(i)}}
$$

$$
+ \sum_{i=0}^{M-1} (1-\beta) \underbrace{\sum_{j=i+1}^{M} \left[ \prod_{m=j}^{M-1} \left( (1-\beta)\boldsymbol{I} + \beta \hat{\boldsymbol{A}}^{(2m+1)} \right) \right] \left[ \prod_{m=i+1}^{j-1} \hat{\boldsymbol{A}}^{(2m+1)} \right] \left( \hat{\boldsymbol{A}}^{(2i+1)} - \bar{\boldsymbol{A}} \right) \bar{\boldsymbol{A}}^i \boldsymbol{c}^{\boldsymbol{y}}}_{=\hat{\boldsymbol{L}}_2^{(i,M)}}.
$$

$\square$

**Lemma 4** (Explicit Formula of $\boldsymbol{v}$).

$$\boldsymbol{v}^{(M+1)} = \sum_{i=0}^{M} \alpha^{M-i+1} \hat{\boldsymbol{B}}^{(2i)} \boldsymbol{u}^{(i)} + (1-\alpha) \sum_{i=0}^{M} \alpha^{M-i} \hat{\boldsymbol{B}}^{(2i)} \boldsymbol{w}^{(i)}. \tag{23}$$

*Proof.* We prove the claim by induction. By setting $m = 0$ in (17), we have

$$\boldsymbol{v}^{(1)} = \alpha \left( \boldsymbol{v}^{(0)} + \hat{\boldsymbol{B}}^{(0)} \boldsymbol{u}^{(0)} \right) + (1-\alpha) \hat{\boldsymbol{B}}^{(0)} \boldsymbol{w}^{(0)}$$

$$= \alpha \hat{\boldsymbol{B}}^{(0)} \boldsymbol{c}^{\boldsymbol{y}} + (1-\alpha) \hat{\boldsymbol{B}}^{(0)} \boldsymbol{c}^{\boldsymbol{y}} = \hat{\boldsymbol{B}}^{(0)} \boldsymbol{c}^{\boldsymbol{y}}.$$

By setting $M = 0$ in (23), we also have

$$\boldsymbol{v}^{(1)} = \alpha \hat{\boldsymbol{B}}^{(0)} \boldsymbol{c}^{\boldsymbol{y}} + (1-\alpha) \alpha \hat{\boldsymbol{B}}^{(0)} \boldsymbol{w}^{(0)} = \hat{\boldsymbol{B}}^{(0)} \boldsymbol{c}^{\boldsymbol{y}},$$

which confirms that (23) is valid when $M = 0$.

Now, suppose that the statement is true for some $M \geq 1$. Then, by (17),

$$\boldsymbol{v}^{(M+1)} = \alpha \left( \boldsymbol{v}^{(M)} + \hat{\boldsymbol{B}}^{(2M)} \boldsymbol{u}^{(M)} \right) + (1-\alpha) \hat{\boldsymbol{B}}^{(2M)} \boldsymbol{w}^{(M)}$$

$$= \alpha \left( \sum_{i=0}^{M-1} \alpha^{M-i} \hat{\boldsymbol{B}}^{(2i)} \boldsymbol{u}^{(i)} + (1-\alpha) \sum_{i=0}^{M-1} \alpha^{M-i-1} \hat{\boldsymbol{B}}^{(2i)} \boldsymbol{w}^{(i)} \right)$$

$$+ \alpha \hat{\boldsymbol{B}}^{(2M)} \boldsymbol{u}^{(M)} + (1-\alpha) \hat{\boldsymbol{B}}^{(2M)} \boldsymbol{w}^{(M)}$$

$$= \left( \sum_{i=0}^{M-1} \alpha^{M-i+1} \hat{\boldsymbol{B}}^{(2i)} \boldsymbol{u}^{(i)} + \alpha \hat{\boldsymbol{B}}^{(2M)} \boldsymbol{u}^{(M)} \right)$$

$$+ (1-\alpha) \left( \sum_{i=0}^{M-1} \alpha^{M-i} \hat{\boldsymbol{B}}^{(2i)} \boldsymbol{w}^{(i)} + \hat{\boldsymbol{B}}^{(2M)} \boldsymbol{w}^{(M+1)} \right)$$

$$= \sum_{i=0}^{M} \alpha^{M-i+1} \hat{\boldsymbol{B}}^{(2i)} \boldsymbol{u}^{(i)} + (1-\alpha) \sum_{i=0}^{M} \alpha^{M-i} \hat{\boldsymbol{B}}^{(2i)} \boldsymbol{w}^{(i)}.$$

$\square$

**Lemma 5** (Decomposition of $\boldsymbol{v}$).

$$\boldsymbol{v}^{(M+1)} - \bar{\boldsymbol{B}} \sum_{i=0}^{M} \bar{\boldsymbol{A}}^i \boldsymbol{c}^{\boldsymbol{y}}$$

$$= \sum_{i=0}^{M} (\hat{\boldsymbol{B}}^{(2i)} - \bar{\boldsymbol{B}}) \boldsymbol{R}_3^{(i,M)} \boldsymbol{c}^{\boldsymbol{y}} + \sum_{i=0}^{M-1} \left( \hat{\boldsymbol{L}}_4^{(i,M)} (\hat{\boldsymbol{A}}^{(2i+1)} - \bar{\boldsymbol{A}}) \bar{\boldsymbol{A}}^i + \hat{\boldsymbol{L}}_5^{(i,M)} (\hat{\boldsymbol{A}}^{(2i+1)} - \bar{\boldsymbol{A}}) \boldsymbol{R}_1^{(i)} \right) \boldsymbol{c}^{\boldsymbol{y}}, \tag{24}$$

where

$$\boldsymbol{R}_3^{(i,M)} = \alpha^{M-i+1} \bar{\boldsymbol{A}}^i + (1-\alpha) \alpha^{M-i} \sum_{j=0}^{i} \bar{\boldsymbol{A}}^j,$$

$$\hat{\boldsymbol{L}}_4^{(i,M)} = \sum_{j=i+1}^{M} \alpha^{M-j+1} \hat{\boldsymbol{B}}^{(2j)} \left[ \prod_{m=i+1}^{j-1} \hat{\boldsymbol{A}}^{(2m+1)} \right]$$

$$+ (1-\alpha)(1-\beta) \sum_{j=i+1}^{M} \alpha^{M-j} \hat{\boldsymbol{B}}^{(2j)} \sum_{k=i+1}^{j} \left[ \prod_{m=k}^{j-1} \left( (1-\beta)\boldsymbol{I} + \beta \hat{\boldsymbol{A}}^{(2m+1)} \right) \right] \left[ \prod_{m=i+1}^{k-1} \hat{\boldsymbol{A}}^{(2m+1)} \right],$$

$$\hat{\boldsymbol{L}}_5^{(i,M)} = (1-\alpha)\beta \sum_{j=i+1}^{M} \alpha^{M-j} \hat{\boldsymbol{B}}^{(2j)} \left[ \prod_{m=i+1}^{j-1} \left( (1-\beta)\boldsymbol{I} + \beta \hat{\boldsymbol{A}}^{(2m+1)} \right) \right].$$

*Proof.* We first recall that, as the corollary of Lemma 4,

$$\bar{\boldsymbol{B}}\sum_{i=0}^{M}\bar{\boldsymbol{A}}^i\boldsymbol{c^y}=\sum_{i=0}^{M}\alpha^{M-i+1}\bar{\boldsymbol{B}}\bar{\boldsymbol{A}}^i\boldsymbol{c^y}+(1-\alpha)\sum_{i=0}^{M}\alpha^{M-i}\bar{\boldsymbol{B}}\sum_{j=0}^{i}\bar{\boldsymbol{A}}^j\boldsymbol{c^y}.$$

By using Lemma 4 and Lemma 2, we can expand the difference as

$$\boldsymbol{v}^{(M+1)}-\bar{\boldsymbol{B}}\sum_{i=0}^{M}\bar{\boldsymbol{A}}^i\boldsymbol{c^y}$$

$$=\sum_{i=0}^{M}\alpha^{M-i+1}\left(\hat{\boldsymbol{B}}^{(2i)}\left[\prod_{m=0}^{i-1}\hat{\boldsymbol{A}}^{(2m+1)}\right]-\bar{\boldsymbol{B}}\bar{\boldsymbol{A}}^i\right)\boldsymbol{c^y}+(1-\alpha)\sum_{i=0}^{M}\alpha^{M-i}\left(\hat{\boldsymbol{B}}^{(2i)}\boldsymbol{w}^{(i)}-\bar{\boldsymbol{B}}\sum_{j=0}^{i}\bar{\boldsymbol{A}}^j\boldsymbol{c^y}\right)$$

$$=\sum_{i=0}^{M}\alpha^{M-i+1}\left(\hat{\boldsymbol{B}}^{(2i)}-\bar{\boldsymbol{B}}\right)\bar{\boldsymbol{A}}^i\boldsymbol{c^y}+\sum_{j=1}^{M}\alpha^{M-j+1}\hat{\boldsymbol{B}}^{(2j)}\sum_{i=0}^{j-1}\left[\prod_{m=i+1}^{j-1}\hat{\boldsymbol{A}}^{(2m+1)}\right]\left(\hat{\boldsymbol{A}}^{(2i+1)}-\bar{\boldsymbol{A}}\right)\bar{\boldsymbol{A}}^i\boldsymbol{c^y}$$

$$+(1-\alpha)\sum_{i=0}^{M}\alpha^{M-i}\left(\hat{\boldsymbol{B}}^{(2i)}-\bar{\boldsymbol{B}}\right)\sum_{j=0}^{i}\bar{\boldsymbol{A}}^j\boldsymbol{c^y}+(1-\alpha)\sum_{i=1}^{M}\alpha^{M-i}\hat{\boldsymbol{B}}^{(2i)}\left(\boldsymbol{w}^{(i)}-\sum_{j=0}^{i}\bar{\boldsymbol{A}}^j\boldsymbol{c^y}\right).$$

By substituting (22), we have

$$\boldsymbol{v}^{(M+1)}-\bar{\boldsymbol{B}}\sum_{i=0}^{M}\bar{\boldsymbol{A}}^i\boldsymbol{c^y}$$

$$=\sum_{i=0}^{M}\alpha^{M-i+1}\left(\hat{\boldsymbol{B}}^{(2i)}-\bar{\boldsymbol{B}}\right)\bar{\boldsymbol{A}}^i\boldsymbol{c^y}+\sum_{j=1}^{M}\alpha^{M-j+1}\hat{\boldsymbol{B}}^{(2j)}\sum_{i=0}^{j-1}\left[\prod_{m=i+1}^{j-1}\hat{\boldsymbol{A}}^{(2m+1)}\right]\left(\hat{\boldsymbol{A}}^{(2i+1)}-\bar{\boldsymbol{A}}\right)\bar{\boldsymbol{A}}^i\boldsymbol{c^y}$$

$$+(1-\alpha)\sum_{i=0}^{M}\alpha^{M-i}\left(\hat{\boldsymbol{B}}^{(2i)}-\bar{\boldsymbol{B}}\right)\sum_{j=0}^{i}\bar{\boldsymbol{A}}^j\boldsymbol{c^y}$$

$$+(1-\alpha)\sum_{j=1}^{M}\alpha^{M-j}\hat{\boldsymbol{B}}^{(2j)}\left(\sum_{i=0}^{j-1}\hat{\boldsymbol{L}}_1^{(i,j)}(\hat{\boldsymbol{A}}^{(2i+1)}-\bar{\boldsymbol{A}})\boldsymbol{R}_1^{(i)}+\hat{\boldsymbol{L}}_2^{(i,j)}(\hat{\boldsymbol{A}}^{(2i+1)}-\bar{\boldsymbol{A}})\bar{\boldsymbol{A}}^i\right)\boldsymbol{c^y}$$

$$=\sum_{i=0}^{M}\left(\hat{\boldsymbol{B}}^{(2i)}-\bar{\boldsymbol{B}}\right)\underbrace{\left(\alpha^{M-i+1}\bar{\boldsymbol{A}}^i+(1-\alpha)\alpha^{M-i}\sum_{j=0}^{i}\bar{\boldsymbol{A}}^j\right)}_{=\boldsymbol{R}_3^{(i,M)}}\boldsymbol{c^y}$$

$$+\sum_{i=0}^{M-1}\underbrace{\left(\sum_{j=i+1}^{M}\alpha^{M-j+1}\hat{\boldsymbol{B}}^{(2j)}\left[\prod_{m=i+1}^{j-1}\hat{\boldsymbol{A}}^{(2m+1)}\right]+(1-\alpha)\sum_{j=i+1}^{M}\alpha^{M-j}\hat{\boldsymbol{B}}^{(2j)}\hat{\boldsymbol{L}}_2^{(i,j)}\right)}_{=\hat{\boldsymbol{L}}_4^{(i,M)}}\left(\hat{\boldsymbol{A}}^{(2i+1)}-\bar{\boldsymbol{A}}\right)\bar{\boldsymbol{A}}^i\boldsymbol{c^y}$$

$$+\sum_{i=0}^{M-1}\underbrace{(1-\alpha)\sum_{j=i+1}^{M}\alpha^{M-j}\hat{\boldsymbol{B}}^{(2j)}\hat{\boldsymbol{L}}_1^{(i,j)}}_{=\hat{\boldsymbol{L}}_5^{(i,M)}}(\hat{\boldsymbol{A}}^{(2i+1)}-\bar{\boldsymbol{A}})\boldsymbol{R}_1^{(i)}\boldsymbol{c^y}.$$

By substituting $\boldsymbol{L}_2^{(i,j)}$, $\boldsymbol{L}_1^{(i,j)}$, we obtain the claim. $\qquad\square$

To bound the estimation error of hyper-gradient, we need to bound each term of (24). The following lemma gives the bounds for each coefficient matrices in (24).

**Lemma 6.** Under Assumption 9, we have

$$\left\|\boldsymbol{R}_3^{(i,M)}\right\|_2 \le \frac{1-\alpha}{1-\eta_A}\alpha^{M-i} + \frac{1}{1-\eta_A}\alpha^{M-i+1}\eta_A^i - \frac{1}{1-\eta_A}\alpha^{M-i}\eta_A^{i+1}, \tag{25}$$

$$\left\|\hat{\boldsymbol{L}}_4^{(i,M)}\right\|_2 \le \frac{\eta_B\alpha\beta}{\alpha-(1-\beta+\beta\eta_A)}\alpha^{M-i}$$
$$\qquad - \frac{\eta_B}{1-\eta_A}\eta_A^{M-i} - \frac{\eta_B}{1-\eta_A}\frac{1-\alpha}{\alpha-(1-\beta+\beta\eta_A)}(1-\beta+\beta\eta_A)^{M-i+1}, \tag{26}$$

$$\left\|\hat{\boldsymbol{L}}_5^{(i,M)}\right\|_2 \le \eta_B\frac{(1-\alpha)\beta}{\alpha-(1-\beta+\beta\eta_A)}\left(\alpha^{M-i} - (1-\beta+\beta\eta_A)^{M-i}\right), \tag{27}$$

$$\left\|\boldsymbol{R}_1^{(i)}\right\|_2 \le \frac{1-\eta_A^{i+1}}{1-\eta_A}. \tag{28}$$

*Proof.* Recall that Assumption 9 ensures $\left\|\bar{\boldsymbol{A}}\right\|_2 \le \eta_A$, $\left\|\hat{\boldsymbol{A}}\right\|_2 \le \eta_A$, $\left\|\bar{\boldsymbol{B}}\right\|_2 \le \eta_B$, $\left\|\hat{\boldsymbol{B}}\right\|_2 \le \eta_B$. Then, we have

$$\left\|\boldsymbol{R}_3^{(i,M)}\right\|_2 \le \alpha^{M-i+1}\left\|\bar{\boldsymbol{A}}\right\|_2^i + (1-\alpha)\alpha^{M-i}\sum_{j=0}^{i}\left\|\bar{\boldsymbol{A}}\right\|_2^j$$

$$\le \alpha^{M-i+1}\eta_A^i + (1-\alpha)\alpha^{M-i}\sum_{j=0}^{i}\eta_A^j$$

$$= \alpha^{M-i+1}\eta_A^i + (1-\alpha)\alpha^{M-i}\frac{1-\eta_A^{i+1}}{1-\eta_A}$$

$$= \alpha^{M-i+1}\eta_A^i + \frac{1-\alpha}{1-\eta_A}\alpha^{M-i} - \frac{1}{1-\eta_A}\alpha^{M-i}\eta_A^{i+1} + \frac{\eta_A}{1-\eta_A}\alpha^{M-i+1}\eta_A^i$$

$$= \frac{1-\alpha}{1-\eta_A}\alpha^{M-i} + \frac{1}{1-\eta_A}\alpha^{M-i+1}\eta_A^i - \frac{1}{1-\eta_A}\alpha^{M-i}\eta_A^{i+1},$$

$$\left\|\hat{\boldsymbol{L}}_4^{(i,M)}\right\|_2 \le \sum_{j=i+1}^{M}\alpha^{M-j+1}\left\|\hat{\boldsymbol{B}}^{(2j)}\right\|_2\prod_{m=i+1}^{j-1}\left\|\hat{\boldsymbol{A}}^{(2m+1)}\right\|_2$$

$$\qquad + (1-\alpha)(1-\beta)\sum_{j=i+1}^{M}\alpha^{M-j}\left\|\hat{\boldsymbol{B}}^{(2j)}\right\|_2\sum_{k=i+1}^{j}\prod_{m=k}^{j-1}\left\|(1-\beta)I + \beta\hat{\boldsymbol{A}}^{(2m+1)}\right\|_2\prod_{m=i+1}^{k-1}\left\|\hat{\boldsymbol{A}}^{(2m+1)}\right\|_2$$

$$\le \eta_B\sum_{j=i+1}^{M}\alpha^{M-j+1}\eta_A^{j-i-1} + \eta_B(1-\alpha)(1-\beta)\sum_{j=i+1}^{M}\alpha^{M-j}\sum_{k=i+1}^{j}(1-\beta+\beta\eta_A)^{j-k}\eta_A^{k-i-1}$$

$$= \eta_B\frac{\alpha}{\alpha-\eta_A}\left(\alpha^{M-i} - \eta_A^{M-i}\right) + \eta_B\frac{1-\alpha}{1-\eta_A}\sum_{j=i+1}^{M}\alpha^{M-j}\left((1-\beta+\beta\eta_A)^{j-i} - \eta_A^{j-i}\right)$$

$$= \eta_B\frac{\alpha}{\alpha-\eta_A}\left(\alpha^{M-i} - \eta_A^{M-i}\right)$$

$$\qquad + \eta_B\frac{1-\alpha}{1-\eta_A}\left(\frac{(1-\beta+\beta\eta_A)\left(\alpha^{M-i} - (1-\beta+\beta\eta_A)^{M-i}\right)}{\alpha-(1-\beta+\beta\eta_A)} - \frac{\eta_A\left(\alpha^{M-i} - \eta_A^{M-i}\right)}{\alpha-\eta_A}\right)$$

$$= \eta_B\frac{1}{1-\eta_A}\left(\alpha^{M-i} - \eta_A^{M-i}\right) + \eta_B\frac{1-\alpha}{1-\eta_A}\frac{1-\beta+\beta\eta_A}{\alpha-(1-\beta+\beta\eta_A)}\left(\alpha^{M-i} - (1-\beta+\beta\eta_A)^{M-i}\right)$$

$$= \frac{\eta_B\alpha\beta}{\alpha-(1-\beta+\beta\eta_A)}\alpha^{M-i} - \frac{\eta_B}{1-\eta_A}\eta_A^{M-i} - \frac{\eta_B}{1-\eta_A}\frac{1-\alpha}{\alpha-(1-\beta+\beta\eta_A)}(1-\beta+\beta\eta_A)^{M-i+1},$$

$$\left\|\hat{\boldsymbol{L}}_5^{(i,M)}\right\|_2 \leq (1-\alpha)\beta \sum_{j=i+1}^{M} \alpha^{M-j} \left\|\hat{\boldsymbol{B}}^{(2j)}\right\|_2 \prod_{m=i+1}^{j-1} \left\|(1-\beta)\boldsymbol{I} + \beta\hat{\boldsymbol{A}}^{(2m+1)}\right\|_2$$

$$\leq \eta_B(1-\alpha)\beta \sum_{j=i+1}^{M} \alpha^{M-j}(1-\beta+\beta\eta_A)^{j-i-1}$$

$$= \eta_B(1-\alpha)\beta \frac{\alpha^{M-i} - (1-\beta+\beta\eta_A)^{M-i}}{\alpha - (1-\beta+\beta\eta_A)},$$

$$\left\|\boldsymbol{R}_1^{(i)}\right\|_2 \leq \left\|(1-\beta)\boldsymbol{I} + \beta\bar{\boldsymbol{A}}\right\|_2^i + \sum_{j=1}^{i} \left\|(1-\beta)\boldsymbol{I} + \beta\bar{\boldsymbol{A}}\right\|_2^{i-j} \left(\beta + (1-\beta)\|\bar{\boldsymbol{A}}\|_2^j\right)$$

$$\leq (1-\beta+\beta\eta_A)^i + \beta\sum_{j=1}^{i}(1-\beta+\beta\eta_A)^{i-j} + (1-\beta)\sum_{j=1}^{i}(1-\beta+\beta\eta_A)^{i-j}\eta_A^j$$

$$= (1-\beta+\beta\eta_A)^i + \frac{1-(1-\beta+\beta\eta_A)^i}{1-\eta_A} + \eta_A\frac{(1-\beta+\beta\eta_A)^i - \eta_A^i}{1-\eta_A}$$

$$= \frac{1-\eta_A^{i+1}}{1-\eta_A}.$$

$\square$

## C.2 Decomposition of $\hat{\boldsymbol{A}}$, $\hat{\boldsymbol{B}}$

We can decompose the difference $\hat{\boldsymbol{A}} - \bar{\boldsymbol{A}}$ and $\hat{\boldsymbol{B}} - \bar{\boldsymbol{B}}$ as

$$\hat{\boldsymbol{A}} - \bar{\boldsymbol{A}} = \left[\mathbb{1}_{ji}(\hat{\boldsymbol{A}}_i^\psi - \bar{\boldsymbol{A}}_i^\psi) + \bar{p}_{ji}\left(\frac{\delta_{i\to j}}{\bar{\delta}_{i\to j}}\hat{\boldsymbol{A}}_j^\varphi - \bar{\boldsymbol{A}}_j^\varphi\right)\right]_{ji}$$

$$= \left[\left(\frac{\delta_{i\to j}}{\bar{\delta}_{i\to j}} - 1\right)\bar{p}_{ji}\hat{\boldsymbol{A}}_j^\varphi\right]_{ji} + \left[\mathbb{1}_{ji}(\hat{\boldsymbol{A}}_i^\psi - \bar{\boldsymbol{A}}_i^\psi) + \bar{p}_{ji}\left(\hat{\boldsymbol{A}}_j^\varphi - \bar{\boldsymbol{A}}_j^\varphi\right)\right]_{ji}$$

$$= \sum_{i,j=1}^{n} \boldsymbol{e}_j\boldsymbol{e}_i^\top \otimes \left(\frac{\delta_{i\to j}}{\bar{\delta}_{i\to j}} - 1\right)\bar{p}_{ji}\hat{\boldsymbol{A}}_j^\varphi$$

$$+ \sum_{i,j=1}^{n} \boldsymbol{e}_j\boldsymbol{e}_i^\top \otimes \left(\frac{\mathbb{1}_{ji}}{|\zeta_i|}\sum_{\xi\in\zeta_i}\left(\partial_{\boldsymbol{y}_i}g_i(\boldsymbol{y}_i^*,\boldsymbol{\lambda}_i,\xi) - \bar{\boldsymbol{A}}_i^\psi\right) + \frac{\bar{p}_{ji}}{|\zeta_j|}\sum_{\xi\in\zeta_j}\left(\partial_{\boldsymbol{y}_j}h_j(\boldsymbol{y}_j^*,\boldsymbol{\lambda}_j,\xi) - \bar{\boldsymbol{A}}_j^\varphi\right)\right),$$

$$\hat{\boldsymbol{B}} - \bar{\boldsymbol{B}} = \left[\mathbb{1}_{ji}(\hat{\boldsymbol{B}}_i^\psi - \bar{\boldsymbol{B}}_i^\psi) + \bar{p}_{ji}\left(\frac{\delta_{i\to j}}{\bar{\delta}_{i\to j}}\hat{\boldsymbol{B}}_j^\varphi - \bar{\boldsymbol{B}}_j^\varphi\right)\right]_{ji}$$

$$= \left[\left(\frac{\delta_{i\to j}}{\bar{\delta}_{i\to j}} - 1\right)\bar{p}_{ji}\hat{\boldsymbol{B}}_j^\varphi\right]_{ji} + \left[\mathbb{1}_{ji}(\hat{\boldsymbol{B}}_i^\psi - \bar{\boldsymbol{B}}_i^\psi) + \bar{p}_{ji}\left(\hat{\boldsymbol{B}}_j^\varphi - \bar{\boldsymbol{B}}_j^\varphi\right)\right]_{ji}$$

$$= \sum_{i,j=1}^{n} \boldsymbol{e}_j\boldsymbol{e}_i^\top \otimes \left(\frac{\delta_{i\to j}}{\bar{\delta}_{i\to j}} - 1\right)\bar{p}_{ji}\hat{\boldsymbol{B}}_j^\varphi$$

$$+ \sum_{i,j=1}^{n} \boldsymbol{e}_j\boldsymbol{e}_i^\top \otimes \left(\frac{\mathbb{1}_{ji}}{|\zeta_i|}\sum_{\xi\in\zeta_i}\left(\partial_{\boldsymbol{\lambda}_i}g_i(\boldsymbol{y}_i^*,\boldsymbol{\lambda}_i,\xi) - \bar{\boldsymbol{B}}_i^\psi\right) + \frac{\bar{p}_{ji}}{|\zeta_j|}\sum_{\xi\in\zeta_j}\left(\partial_{\boldsymbol{\lambda}_j}h_j(\boldsymbol{y}_j^*,\boldsymbol{\lambda}_j,\xi) - \bar{\boldsymbol{B}}_j^\varphi\right)\right),$$

where $\boldsymbol{e}_i, \boldsymbol{e}_j$ are $i$-th and $j$-th canonical basis vectors.

By using these expressions, we can rewrite Lemma 5 as

$$
\boldsymbol{v}^{(M+1)} - \bar{\boldsymbol{B}} \sum_{i=0}^{M} \bar{\boldsymbol{A}}^i \boldsymbol{c}^{\boldsymbol{y}}
$$

$$
= \sum_{i=0}^{M} \sum_{s,t=1}^{n} \boldsymbol{X}_{B,st}^{(i)} \boldsymbol{c}^{\boldsymbol{y}} + \sum_{i=0}^{M} \sum_{t=1}^{n} \sum_{\xi \in \zeta_s^{(2i)}} \boldsymbol{Y}_{B,t}^{(i,\xi)} \boldsymbol{c}^{\boldsymbol{y}} + \sum_{i=0}^{M-1} \sum_{s,t=1}^{n} \boldsymbol{X}_{A,st}^{(i)} \boldsymbol{c}^{\boldsymbol{y}} + \sum_{i=0}^{M-1} \sum_{t=1}^{n} \sum_{\xi \in \zeta_s^{(2i+1)}} \boldsymbol{Y}_{A,t}^{(i,\xi)} \boldsymbol{c}^{\boldsymbol{y}},
$$

where

$$
\boldsymbol{X}_{B,st}^{(i)} = \left( \boldsymbol{e}_t \boldsymbol{e}_s^\top \otimes \left( \frac{\delta_{s \to t}^{(2i)}}{\bar{\delta}_{s \to t}} - 1 \right) \bar{p}_{ts} \hat{\boldsymbol{B}}_t^{\varphi(2i)} \right) \boldsymbol{R}_3^{(i,M)},
$$

$$
\boldsymbol{Y}_{B,t}^{(i)} = \sum_{s=1}^{n} \left( \boldsymbol{e}_t \boldsymbol{e}_s^\top \otimes \frac{1}{|\zeta_t^{(2i)}|} \left( \mathbb{1}_{ts} \partial_{\boldsymbol{\lambda}_t} g_t(\boldsymbol{y}_t^*, \boldsymbol{\lambda}_t, \xi) + \bar{p}_{ts} \partial_{\boldsymbol{\lambda}_t} h_t(\boldsymbol{y}_t^*, \boldsymbol{\lambda}_t, \xi) - \mathbb{1}_{ts} \bar{\boldsymbol{B}}_t^\psi - \bar{p}_{ts} \bar{\boldsymbol{B}}_t^\varphi \right) \right) \boldsymbol{R}_3^{(i,M)},
$$

$$
\boldsymbol{X}_{A,st}^{(i)} = \hat{\boldsymbol{L}}_4^{(i,M)} \left( \boldsymbol{e}_t \boldsymbol{e}_s^\top \otimes \left( \frac{\delta_{s \to t}^{(2i+1)}}{\bar{\delta}_{s \to t}} - 1 \right) \bar{p}_{ts} \hat{\boldsymbol{A}}_t^{\varphi(2i+1)} \right) \bar{\boldsymbol{A}}^i
$$

$$
+ \hat{\boldsymbol{L}}_5^{(i,M)} \left( \boldsymbol{e}_t \boldsymbol{e}_s^\top \otimes \left( \frac{\delta_{s \to t}^{(2i+1)}}{\bar{\delta}_{s \to t}} - 1 \right) \bar{p}_{ts} \hat{\boldsymbol{A}}_t^{\varphi(2i+1)} \right) \boldsymbol{R}_1^{(i)},
$$

$$
\boldsymbol{Y}_{B,t}^{(i)} = \hat{\boldsymbol{L}}_4^{(i,M)} \sum_{s=1}^{n} \left( \boldsymbol{e}_t \boldsymbol{e}_s^\top \otimes \frac{1}{|\zeta_t^{(2i+1)}|} \left( \mathbb{1}_{ts} \partial_{\boldsymbol{y}_t} g_t(\boldsymbol{y}_t^*, \boldsymbol{\lambda}_t, \xi) + \bar{p}_{ts} \partial_{\boldsymbol{y}_t} h_t(\boldsymbol{y}_t^*, \boldsymbol{\lambda}_t, \xi) - \mathbb{1}_{ts} \bar{\boldsymbol{A}}_t^\psi - \bar{p}_{ts} \bar{\boldsymbol{A}}_t^\varphi \right) \right) \bar{\boldsymbol{A}}^i
$$

$$
+ \hat{\boldsymbol{L}}_5^{(i,M)} \sum_{s=1}^{n} \left( \boldsymbol{e}_t \boldsymbol{e}_s^\top \otimes \frac{1}{|\zeta_t^{(2i+1)}|} \left( \mathbb{1}_{ts} \partial_{\boldsymbol{y}_t} g_t(\boldsymbol{y}_t^*, \boldsymbol{\lambda}_t, \xi) + \bar{p}_{ts} \partial_{\boldsymbol{y}_t} h_t(\boldsymbol{y}_t^*, \boldsymbol{\lambda}_t, \xi) - \mathbb{1}_{ts} \bar{\boldsymbol{A}}_t^\psi - \bar{p}_{ts} \bar{\boldsymbol{A}}_t^\varphi \right) \right) \boldsymbol{R}_1^{(i)}.
$$

Here, we note that $\hat{\boldsymbol{L}}_4^{(i,M)}$ and $\hat{\boldsymbol{L}}_5^{(i,M)}$ depend only on $\hat{\boldsymbol{A}}^{(2i+3)}, \ldots, \hat{\boldsymbol{A}}^{(2M-1)}$ and $\hat{\boldsymbol{B}}^{(2i+2)}, \ldots, \hat{\boldsymbol{B}}^{(2M)}$. We therefore have

$$
\begin{aligned}
\mathbb{E}_{\delta_{s \to t}^{(2i+1)}, \zeta_t^{(2i+1)}} \left[ \boldsymbol{X}_{A,st}^{(i)} \mid \hat{\boldsymbol{A}}^{(2i+3)}, \ldots, \hat{\boldsymbol{A}}^{(2M-1)}, \hat{\boldsymbol{B}}^{(2i+2)}, \ldots, \hat{\boldsymbol{B}}^{(2M)} \right] = 0, \\
\mathbb{E}_{\delta_{s \to t}^{(2i+1)}, \zeta_t^{(2i+1)}} \left[ \boldsymbol{Y}_{A,t}^{(i,\xi)} \mid \hat{\boldsymbol{A}}^{(2i+3)}, \ldots, \hat{\boldsymbol{A}}^{(2M-1)}, \hat{\boldsymbol{B}}^{(2i+2)}, \ldots, \hat{\boldsymbol{B}}^{(2M)} \right] = 0,
\end{aligned}
\tag{29}
$$

by the independence of $\delta_{s \to t}^{(2i+1)}$ and $\zeta_t^{(2i+1)}$ in Assumption 5.

## C.3  BOUND FOR $\alpha \in (0,1)$ AND $\beta \in (0,1)$

We now derive the error bound of VR-HGP for the case when $\alpha, \beta \in (0,1)$. The error bound follows from the next bounds on $\boldsymbol{X}_{B,st}^{(i)}, \boldsymbol{Y}_{B,t}^{(i)}, \boldsymbol{X}_{A,st}^{(i)}$, and $\boldsymbol{Y}_{A,t}^{(i)}$.

**Lemma 7.** Under Assumption 9, when $\alpha, \beta \in (0,1)$ so that $1 - \beta + \beta\eta_A \in (\eta_A, 1)$, we have

$$
\left\| \boldsymbol{X}_{B,st}^{(i)} \right\|_2^2 \leq \frac{\eta_B^2}{\kappa^2} \frac{\bar{p}_{ts}^2}{\bar{\delta}_{s \to t}^2} \left( \frac{1-\alpha}{1-\eta_A} \right)^2 \alpha^{2(M-i)} + \exp(-O(M)),
\tag{30}
$$

$$
\left\| \boldsymbol{X}_{A,st}^{(i)} \right\|_2^2 \leq \frac{\eta_B^2}{\kappa^2} \frac{\bar{p}_{ts}^2}{\bar{\delta}_{s \to t}^2} \left( \frac{\eta_A}{1-\eta_A} \frac{(1-\alpha)\beta}{\alpha - (1-\beta+\beta\eta_A)} \right)^2 \left( \alpha^{M-i} - (1-\beta+\beta\eta_A)^{M-i} \right)^2 + \exp(-O(M)),
\tag{31}
$$

$$
\left\| \boldsymbol{Y}_{B,t}^{(i,\xi)} \right\|_2^2 \leq \frac{4\eta_B^2}{\kappa^2 |\zeta_t^{(2i)}|^2} \left( \frac{1-\alpha}{1-\eta_A} \right)^2 \alpha^{2(M-i)} + \exp(-O(M)),
\tag{32}
$$

$$
\left\| \boldsymbol{Y}_{A,t}^{(i,\xi)} \right\|_2^2 \leq \frac{4\eta_B^2}{\kappa^2 |\zeta_t^{(2i+1)}|^2} \left( \frac{\eta_A}{1-\eta_A} \frac{(1-\alpha)\beta}{\alpha - (1-\beta+\beta\eta_A)} \right)^2 \left( \alpha^{M-i} - (1-\beta+\beta\eta_A)^{M-i} \right)^2 + \exp(-O(M)),
\tag{33}
$$

where

$$\kappa = \sum_{s,t} \frac{\bar{p}_{ts}}{\bar{\bar{\delta}}_{s \to t}}. \tag{34}$$

*Proof.*

$$\left\| \boldsymbol{X}_{B,st}^{(i)} \right\|_2^2 \leq \underbrace{\left\{ 1 + \left( \frac{1}{\bar{\bar{\delta}}_{s \to t}} - 1 \right)^2 \right\}}_{\leq \frac{1}{\bar{\delta}_{s \to t}^2}} \bar{p}_{ts}^2 \left\| \hat{\boldsymbol{B}}_t^{\varphi(2i)} \right\|_2^2 \left\| \boldsymbol{R}_3^{(i,M)} \right\|_2^2$$

$$\leq \frac{\bar{p}_{ts}^2}{\bar{\delta}_{s \to t}^2} \left( \frac{\eta_B}{2\kappa} \right)^2 \left( \frac{1-\alpha}{1-\eta_A} \alpha^{M-i} + \exp(-O(M)) \right)^2$$

$$= \frac{\eta_B^2}{\kappa^2} \frac{\bar{p}_{ts}^2}{\bar{\delta}_{s \to t}^2} \left( \frac{1-\alpha}{1-\eta_A} \right)^2 \alpha^{2(M-i)} + \exp(-O(M)),$$

$$\left\| \boldsymbol{X}_{A,st}^{(i)} \right\|_2^2 \leq \left\{ 1 + \left( \frac{1}{\bar{\bar{\delta}}_{s \to t}} - 1 \right)^2 \right\} \bar{p}_{ts}^2 \left\| \hat{\boldsymbol{A}}_t^{\varphi(2i+1)} \right\|_2^2 \left( \left\| \hat{\boldsymbol{L}}_4^{(i,M)} \right\|_2 \eta_A^i + \left\| \hat{\boldsymbol{L}}_5^{(i,M)} \right\|_2 \left\| \boldsymbol{R}_1^{(i)} \right\|_2 \right)^2$$

$$\leq \frac{\bar{p}_{ts}^2}{\bar{\delta}_{s \to t}^2} \left( \frac{\eta_A}{2\kappa} \right)^2 \left( \frac{\eta_B}{1-\eta_A} \frac{(1-\alpha)\beta}{\alpha - (1-\beta+\beta\eta_A)} \left( \alpha^{M-i} - (1-\beta+\beta\eta_A)^{M-i} \right) + \exp(-O(M)) \right)^2$$

$$= \frac{\eta_B^2}{\kappa^2} \frac{\bar{p}_{ts}^2}{\bar{\delta}_{s \to t}^2} \left( \frac{\eta_A}{1-\eta_A} \frac{(1-\alpha)\beta}{\alpha - (1-\beta+\beta\eta_A)} \right)^2 \left( \alpha^{M-i} - (1-\beta+\beta\eta_A)^{M-i} \right)^2 + \exp(-O(M)),$$

$$\left\| \boldsymbol{Y}_{B,s}^{(i,\xi)} \right\|_2^2 \leq \frac{1}{|\zeta_t^{(2i)}|^2} \sum_{s=1}^n \left\| \mathbb{1}_{ts} \partial_{\boldsymbol{\lambda}_t} g_t(\boldsymbol{y}_t^*, \boldsymbol{\lambda}_t, \xi) + \bar{p}_{ts} \partial_{\boldsymbol{\lambda}_t} h_t(\boldsymbol{y}_t^*, \boldsymbol{\lambda}_t, \xi) - \mathbb{1}_{ts} \bar{\boldsymbol{B}}_t^\psi - \bar{p}_{ts} \bar{\boldsymbol{B}}_t^\varphi \right\|_2^2 \left\| \boldsymbol{R}_3^{(i,M)} \right\|_2^2$$

$$\leq \frac{1}{|\zeta_s^{(2t)}|^2} \left( \frac{\eta_B}{2\kappa} \right)^2 \left( 2 + 2 \underbrace{\sum_{s=1}^n \bar{p}_{ts}}_{=1} \right)^2 \left( \frac{1-\alpha}{1-\eta_A} \alpha^{M-i} + \exp(-O(M)) \right)^2$$

$$= \frac{4\eta_B^2}{\kappa^2 |\zeta_t^{(2i)}|^2} \left( \frac{1-\alpha}{1-\eta_A} \right)^2 \alpha^{2(M-i)} + \exp(-O(M)),$$

$$\left\| \boldsymbol{Y}_{A,s}^{(i,\xi)} \right\|_2^2 \leq \frac{1}{|\zeta_s^{(2i+1)}|^2} \sum_{s=1}^n \left\| \mathbb{1}_{ts} \partial_{\boldsymbol{y}_t} g_t(\boldsymbol{y}_t^*, \boldsymbol{\lambda}_t, \xi) + \bar{p}_{ts} \partial_{\boldsymbol{y}_t} h_t(\boldsymbol{y}_t^*, \boldsymbol{\lambda}_t, \xi) - \mathbb{1}_{ts} \bar{\boldsymbol{A}}_t^\psi - \bar{p}_{ts} \bar{\boldsymbol{A}}_t^\varphi \right\|_2^2$$

$$\times \left( \left\| \hat{\boldsymbol{L}}_4^{(i,M)} \right\|_2 \eta_A^i + \left\| \hat{\boldsymbol{L}}_5^{(i,M)} \right\|_2 \left\| \boldsymbol{R}_1^{(i)} \right\|_2 \right)^2$$

$$\leq \frac{1}{|\zeta_t^{(2i+1)}|^2} \left( \frac{\eta_A}{2\kappa} \right)^2 \left( 2 + 2 \underbrace{\sum_{s=1}^n \bar{p}_{ts}}_{=1} \right)^2$$

$$\times \left( \frac{\eta_B}{1-\eta_A} \frac{(1-\alpha)\beta}{\alpha - (1-\beta+\beta\eta_A)} \left( \alpha^{M-i} - (1-\beta+\beta\eta_A)^{M-i} \right) + \exp(-O(M)) \right)^2$$

$$= \frac{4\eta_B^2}{\kappa^2 |\zeta_t^{(2i+1)}|^2} \left( \frac{\eta_A}{1-\eta_A} \frac{(1-\alpha)\beta}{\alpha - (1-\beta+\beta\eta_A)} \right)^2 \left( \alpha^{M-i} - (1-\beta+\beta\eta_A)^{M-i} \right)^2 + \exp(-O(M)).$$

$\square$

**Theorem 8.** Suppose Assumptions 1–9 hold true, and $|\zeta_t^{(2i)}| = |\zeta_t^{(2i+1)}| = b$ for any $t$ and $i$. Then, with probability at least $1 - \epsilon$, we have

$$\left\| \boldsymbol{v}^{(M+1)} + \boldsymbol{c}^{\boldsymbol{\lambda}} - \mathrm{d}_{\boldsymbol{\lambda}_j} F(\boldsymbol{y}^*, \boldsymbol{\lambda}) \right\|$$

$$\leq \mu_{\alpha,\beta}\tau \sqrt{\left( \sum_{s,t=1}^n \frac{\bar{p}_{ts}^2}{\bar{\delta}_{s\rightarrow t}^2} + \frac{4n}{b} \right) \log \frac{n(d_{\boldsymbol{y}} + d_{\boldsymbol{\lambda}})}{\epsilon}} + \exp(-O(M)),$$

where

$$\mu_{\alpha,\beta} = \sqrt{8 \frac{1-\alpha}{1+\alpha}\left( 1 + \frac{1 + \alpha(1 - \beta + \beta\eta_A)}{1 - \alpha(1 - \beta + \beta\eta_A)}\frac{\beta^2\eta_A^2}{1 - (1 - \beta + \beta\eta_A)^2} \right)}, \quad \tau = \frac{\eta_B \|\boldsymbol{c}^{\boldsymbol{y}}\|}{\kappa(1 - \eta_A)}.$$

*Proof.* We first have

$$\left\| \boldsymbol{v}^{(M+1)} + \boldsymbol{c}^{\boldsymbol{\lambda}} - \mathrm{d}_{\boldsymbol{\lambda}_j} F(\boldsymbol{y}^*, \boldsymbol{\lambda}) \right\| \leq \left\| \boldsymbol{v}^{(M+1)} - \bar{\boldsymbol{B}} \sum_{i=0}^M \bar{\boldsymbol{A}}^i \boldsymbol{c}^{\boldsymbol{y}} \right\| + \left\| \bar{\boldsymbol{B}} \sum_{i=M+1}^\infty \bar{\boldsymbol{A}}^i \boldsymbol{c}^{\boldsymbol{y}} \right\|.$$

Here, we can bound the second term by

$$\left\| \bar{\boldsymbol{B}} \sum_{i=M+1}^\infty \bar{\boldsymbol{A}}^i \boldsymbol{c}^{\boldsymbol{y}} \right\| \leq \eta_B \|\boldsymbol{c}^{\boldsymbol{y}}\| \sum_{i=M+1}^\infty \eta_A^i \leq \frac{\eta_B \|\boldsymbol{c}^{\boldsymbol{y}}\|}{1 - \eta_A}\eta_A^{M+1} = \exp\left(-O(M)\right).$$

The conditions (29) ensure that we can bound the first term by using Matrix Azuma's inequality; with probability at least $1 - \epsilon$, we have

$$\left\| \boldsymbol{v}^{(M+1)} - \bar{\boldsymbol{B}} \sum_{i=0}^M \bar{\boldsymbol{A}}^i \boldsymbol{c}^{\boldsymbol{y}} \right\| \leq \sqrt{8\sigma^2 \frac{n(d_{\boldsymbol{y}} + d_{\boldsymbol{\lambda}})}{\epsilon}},$$

where

$$\frac{\sigma^2}{\|\boldsymbol{c}^{\boldsymbol{y}}\|^2} \leq \sum_{i=0}^M \sum_{s,t=1}^n \left\| \boldsymbol{X}_{B,st}^{(i)} \right\|_2^2 + \sum_{i=0}^{M-1} \sum_{s,t=1}^n \left\| \boldsymbol{X}_{A,st}^{(i)} \right\|_2^2$$

$$+ \sum_{i=0}^M \sum_{t=1}^n \sum_{\xi \in \zeta_t^{(2i)}} \left\| \boldsymbol{Y}_{B,t}^{(i,\xi)} \right\|_2^2 + \sum_{i=0}^{M-1} \sum_{t=1}^n \sum_{\xi \in \zeta_t^{(2i+1)}} \left\| \boldsymbol{Y}_{A,t}^{(i,\xi)} \right\|_2^2$$

$$\leq \sum_{i=0}^M \sum_{s,t=1}^n \frac{\eta_B^2}{\kappa^2}\frac{\bar{p}_{ts}^2}{\bar{\delta}_{s\rightarrow t}^2}\left( \frac{1-\alpha}{1-\eta_A} \right)^2 \alpha^{2(M-i)}$$

$$+ \sum_{i=0}^{M-1} \sum_{s,t=1}^n \frac{\eta_B^2}{\kappa^2}\frac{\bar{p}_{ts}^2}{\bar{\delta}_{s\rightarrow t}^2}\left( \frac{\eta_A}{1-\eta_A}\frac{(1-\alpha)\beta}{\alpha - (1 - \beta + \beta\eta_A)} \right)^2 \left(\alpha^{M-i} - (1 - \beta + \beta\eta_A)^{M-i}\right)^2$$

$$+ \sum_{i=0}^M \sum_{t=1}^n \sum_{\xi \in \zeta_t^{(2i)}} \frac{4\eta_B^2}{\kappa^2 |\zeta_t^{(2i)}|^2}\left( \frac{1-\alpha}{1-\eta_A} \right)^2 \alpha^{2(M-i)}$$

$$+ \sum_{i=0}^{M-1} \sum_{t=1}^n \sum_{\xi \in \zeta_t^{(2i+1)}} \frac{4\eta_B^2}{\kappa^2 |\zeta_t^{(2i+1)}|^2}\left( \frac{\eta_A}{1-\eta_A}\frac{(1-\alpha)\beta}{\alpha - (1 - \beta + \beta\eta_A)} \right)^2 \left(\alpha^{M-i} - (1 - \beta + \beta\eta_A)^{M-i}\right)^2$$

$$+ \exp(-O(M))$$

$$
= \frac{\eta_B^2}{\kappa^2} \left[ \sum_{s,t=1}^n \frac{\bar{p}_{ts}^2}{\bar{\delta}_{s \to t}^2} \right] \left( \frac{1-\alpha}{1-\eta_A} \right)^2 \frac{1-\alpha^{2(M+1)}}{1-\alpha^2}
$$

$$
+ \frac{\eta_B^2}{\kappa^2} \left[ \sum_{s,t=1}^n \frac{\bar{p}_{ts}^2}{\bar{\delta}_{s \to t}^2} \right] \left( \frac{\eta_A}{1-\eta_A} \frac{(1-\alpha)\beta}{\alpha - (1-\beta+\beta\eta_A)} \right)^2
$$

$$
\times \left( \frac{\alpha^2 - \alpha^{2(M+1)}}{1-\alpha^2} + \frac{(1-\beta+\beta\eta_A)^2 - (1-\beta+\beta\eta_A)^{2(M+1)}}{1-(1-\beta+\beta\eta_A)^2} - 2 \frac{\alpha(1-\beta+\beta\eta_A) - \alpha^{M+1}(1-\beta+\beta\eta_A)^{M+1}}{1-\alpha(1-\beta+\beta\eta_A)} \right)
$$

$$
+ \frac{\eta_B^2}{\kappa^2} \left[ \sum_{t=1}^n \frac{4}{|\zeta_t^{(2i)}|} \right] \left( \frac{1-\alpha}{1-\eta_A} \right)^2 \frac{1-\alpha^{2(M+1)}}{1-\alpha^2}
$$

$$
+ \frac{\eta_B^2}{\kappa^2} \left[ \sum_{s=1}^K \frac{4}{|\zeta_t^{(2i+1)}|} \right] \left( \frac{\eta_A}{1-\eta_A} \frac{(1-\alpha)\beta}{\alpha - (1-\beta+\beta\eta_A)} \right)^2
$$

$$
\times \left( \frac{\alpha^2 - \alpha^{2(M+1)}}{1-\alpha^2} + \frac{(1-\beta+\beta\eta_A)^2 - (1-\beta+\beta\eta_A)^{2(M+1)}}{1-(1-\beta+\beta\eta_A)^2} - 2 \frac{\alpha(1-\beta+\beta\eta_A) - \alpha^{M+1}(1-\beta+\beta\eta_A)^{M+1}}{1-\alpha(1-\beta+\beta\eta_A)} \right)
$$

$$
+ \exp(-O(M))
$$

$$
\leq \frac{\eta_B^2}{\kappa^2} \left[ \sum_{s,t=1}^n \frac{\bar{p}_{ts}^2}{\bar{\delta}_{s \to t}^2} + \sum_{t=1}^n \frac{4}{|\zeta_t^{(2i)}|} \right] \left( \frac{1-\alpha}{1-\eta_A} \right)^2 \frac{1}{1-\alpha^2}
$$

$$
+ \frac{\eta_B^2}{\kappa^2} \left[ \sum_{s,t=1}^n \frac{\bar{p}_{ts}^2}{\bar{\delta}_{s \to t}^2} + \sum_{t=1}^n \frac{4}{|\zeta_t^{(2i+1)}|} \right] \left( \frac{\eta_A}{1-\eta_A} \frac{(1-\alpha)\beta}{\alpha - (1-\beta+\beta\eta_A)} \right)^2
$$

$$
\times \left( \frac{\alpha^2}{1-\alpha^2} + \frac{(1-\beta+\beta\eta_A)^2}{1-(1-\beta+\beta\eta_A)^2} - 2 \frac{\alpha(1-\beta+\beta\eta_A)}{1-\alpha(1-\beta+\beta\eta_A)} \right)
$$

$$
+ \exp(-O(M))
$$

$$
= \frac{\eta_B^2}{\kappa^2} \left[ \sum_{s,t=1}^n \frac{\bar{p}_{ts}^2}{\bar{\delta}_{s \to t}^2} + \sum_{t=1}^n \frac{4}{|\zeta_t^{(2i)}|} \right] \left( \frac{1-\alpha}{1-\eta_A} \right)^2 \frac{1}{1-\alpha^2}
$$

$$
+ \frac{\eta_B^2}{\kappa^2} \left[ \sum_{s,t=1}^n \frac{\bar{p}_{ts}^2}{\bar{\delta}_{s \to t}^2} + \sum_{t=1}^n \frac{4}{|\zeta_t^{(2i+1)}|} \right] \left( \frac{\eta_A}{1-\eta_A} \frac{(1-\alpha)\beta}{\alpha - (1-\beta+\beta\eta_A)} \right)^2
$$

$$
\times \frac{(1+\alpha(1-\beta+\beta\eta_A))(\alpha-(1-\beta+\beta\eta_A))^2}{(1-\alpha^2)(1-(1-\beta+\beta\eta_A)^2)(1-\alpha(1-\beta+\beta\eta_A))}
$$

$$
+ \exp(-O(M))
$$

$$
= \frac{\eta_B^2}{\kappa^2} \left[ \sum_{s,t=1}^n \frac{\bar{p}_{ts}^2}{\bar{\delta}_{s \to t}^2} + \sum_{t=1}^n \frac{4}{|\zeta_t^{(2i)}|} \right] \left( \frac{1}{1-\eta_A} \right)^2 \frac{1-\alpha}{1+\alpha}
$$

$$
+ \frac{\eta_B^2}{\kappa^2} \left[ \sum_{s,t=1}^n \frac{\bar{p}_{ts}^2}{\bar{\delta}_{s \to t}^2} + \sum_{t=1}^n \frac{4}{|\zeta_t^{(2i+1)}|} \right] \left( \frac{1}{1-\eta_A} \right)^2 \frac{1-\alpha}{1+\alpha}
$$

$$
\times \frac{1+\alpha(1-\beta+\beta\eta_A)}{1-\alpha(1-\beta+\beta\eta_A)} \frac{\beta^2\eta_A^2}{1-(1-\beta+\beta\eta_A)^2}
$$

$$
+ \exp(-O(M)).
$$

When $|\zeta_t^{(2i)}| = |\zeta_t^{(2i+1)}| = b$ for any $t$ and $i$, we further have

$$
\sigma^2 \leq \frac{\eta_B^2 \|\boldsymbol{c^y}\|^2}{\kappa^2(1-\eta_A)^2} \left[ \sum_{s,t=1}^n \frac{\bar{p}_{ts}^2}{\bar{\delta}_{s \to t}^2} + \frac{4n}{b} \right]
$$

$$
\times \frac{1-\alpha}{1+\alpha} \left( 1 + \frac{1+\alpha(1-\beta+\beta\eta_A)}{1-\alpha(1-\beta+\beta\eta_A)} \frac{\beta^2\eta_A^2}{1-(1-\beta+\beta\eta_A)^2} \right) + \exp(-O(M)).
$$

$\square$

## C.4  Bound for $\alpha = 1$ and $\beta = 0$

Setting $\alpha = 1$ and $\beta = 0$ recovers naive HGP. Here, we derive the error bound for naive HGP.

**Lemma 9.** Under Assumption 9, when $\alpha = 1$ and $\beta = 0$ so that $1 - \beta + \beta\eta_A = 1$, we have

$$\left\|\boldsymbol{X}_{B,st}^{(i)}\right\|_2^2 \leq \frac{\eta_B^2}{\kappa^2}\frac{\bar{p}_{ts}^2}{\bar{\delta}_{s\to t}^2}\eta_A^{2i}, \tag{35}$$

$$\left\|\boldsymbol{X}_{A,st}^{(i)}\right\|_2^2 \leq \frac{\eta_B^2}{\kappa^2}\frac{\bar{p}_{ts}^2}{\bar{\delta}_{s\to t}^2}\left(\frac{\eta_A}{1-\eta_A}\right)^2\eta_A^{2i} + \exp\left(-O(M)\right), \tag{36}$$

$$\left\|\boldsymbol{Y}_{B,t}^{(i,\xi)}\right\|_2^2 \leq \frac{4\eta_B^2}{\kappa^2|\zeta_t^{(2i)}|^2}\eta_A^{2i}, \tag{37}$$

$$\left\|\boldsymbol{Y}_{A,t}^{(i,\xi)}\right\|_2^2 \leq \frac{4\eta_B^2}{\kappa^2|\zeta_t^{(2i+1)}|^2}\left(\frac{\eta_A}{1-\eta_A}\right)^2\eta_A^{2i} + \exp\left(-O(M)\right). \tag{38}$$

*Proof.*

$$\left\|\boldsymbol{X}_{B,st}^{(i)}\right\|_2^2 \leq \left\{1 + \left(\frac{1}{\bar{\delta}_{s\to t}} - 1\right)^2\right\}\bar{p}_{ts}^2\left\|\hat{\boldsymbol{B}}_t^{\varphi(2i)}\right\|_2^2\left\|\boldsymbol{R}_3^{(i,M)}\right\|_2^2$$

$$\leq \frac{\bar{p}_{ts}^2}{\bar{\delta}_{s\to t}^2}\left(\frac{\eta_B}{2\kappa}\right)^2\left(\eta_A^i\right)^2$$

$$= \frac{\eta_B^2}{\kappa^2}\frac{\bar{p}_{ts}^2}{\bar{\delta}_{s\to t}^2}\eta_A^{2i},$$

$$\left\|\boldsymbol{X}_{A,st}^{(i)}\right\|_2^2 \leq \left\{1 + \left(\frac{1}{\bar{\delta}_{s\to t}} - 2\right)^2\right\}\bar{p}_{ts}^2\left\|\hat{\boldsymbol{A}}_t^{\varphi(2i+1)}\right\|_2^2\left(\left\|\hat{\boldsymbol{L}}_4^{(i,M)}\right\|_2\eta_A^i + \left\|\hat{\boldsymbol{L}}_5^{(i,M)}\right\|_2\left\|\boldsymbol{R}_1^{(i)}\right\|_2\right)^2$$

$$\leq \frac{\bar{p}_{ts}^2}{\bar{\delta}_{s\to t}^2}\left(\frac{\eta_A}{2\kappa}\right)^2\left(\frac{\eta_B}{1-\eta_A}(\eta_A^i - \eta_A^M)\right)^2$$

$$= \frac{\eta_B^2}{\kappa^2}\frac{\bar{p}_{ts}^2}{\bar{\delta}_{s\to t}^2}\left(\frac{\eta_A}{1-\eta_A}\right)^2\eta_A^{2i} + \exp\left(-O(M)\right),$$

$$\left\|\boldsymbol{Y}_{B,s}^{(i,\xi)}\right\|_2^2 \leq \frac{1}{|\zeta_t^{(2i)}|^2}\sum_{s=1}^n\left\|\mathbb{1}_{ts}\partial_{\boldsymbol{\lambda}_t}g_t(\boldsymbol{y}_t^*,\boldsymbol{\lambda}_t,\xi) + \bar{p}_{ts}\partial_{\boldsymbol{\lambda}_t}h_t(\boldsymbol{y}_t^*,\boldsymbol{\lambda}_t,\xi) - \mathbb{1}_{ts}\bar{\boldsymbol{B}}_t^\psi - \bar{p}_{ts}\bar{\boldsymbol{B}}_t^\varphi\right\|_2^2\left\|\boldsymbol{R}_3^{(i,M)}\right\|_2^2$$

$$\leq \frac{1}{|\zeta_s^{(2t)}|^2}\left(\frac{\eta_B}{2\kappa}\right)^2\left(2 + 2\sum_{s=1}^n\bar{p}_{ts}\right)^2\left(\eta_A^i\right)^2$$

$$= \frac{4\eta_B^2}{\kappa^2|\zeta_t^{(2i)}|^2}\eta_A^{2i},$$

$$\left\|\boldsymbol{Y}_{A,s}^{(i,\xi)}\right\|_2^2 \leq \frac{1}{|\zeta_s^{(2i+1)}|^2}\sum_{s=1}^n\left\|\mathbb{1}_{ts}\partial_{\boldsymbol{y}_t}g_t(\boldsymbol{y}_t^*,\boldsymbol{\lambda}_t,\xi) + \bar{p}_{ts}\partial_{\boldsymbol{y}_t}h_t(\boldsymbol{y}_t^*,\boldsymbol{\lambda}_t,\xi) - \mathbb{1}_{ts}\bar{\boldsymbol{A}}_t^\psi - \bar{p}_{ts}\bar{\boldsymbol{A}}_t^\varphi\right\|_2^2$$

$$\times\left(\left\|\hat{\boldsymbol{L}}_4^{(i,M)}\right\|_2\eta_{\boldsymbol{A}}^i + \left\|\hat{\boldsymbol{L}}_5^{(i,M)}\right\|_2\left\|\boldsymbol{R}_1^{(i)}\right\|_2\right)^2$$

$$\leq \frac{1}{|\zeta_t^{(2i+1)}|^2}\left(\frac{\eta_A}{2\kappa}\right)^2\left(2 + 2\sum_{s=1}^n\bar{p}_{ts}\right)^2\left(\frac{\eta_B}{1-\eta_A}(\eta_A^i - \eta_A^M)\right)^2$$

$$= \frac{4\eta_B^2}{\kappa^2|\zeta_t^{(2i+1)}|^2}\left(\frac{\eta_A}{1-\eta_A}\right)^2\eta_A^{2i} + \exp\left(-O(M)\right).$$

$\square$

**Theorem 10.** Suppose Assumptions 1–9 hold true, and $|\zeta_t^{(2i)}| = |\zeta_t^{(2i+1)}| = b$ for any $t$ and $i$. When $\alpha = 1, \beta = 0$, with probability at least $1 - \epsilon$, we have

$$\left\| \boldsymbol{v}^{(M+1)} - \mathrm{d}_{\boldsymbol{\lambda}_j} F(\boldsymbol{y}^*, \boldsymbol{\lambda}) \right\| \leq \mu_{1,0}\tau \sqrt{\left( \sum_{s,t=1}^n \frac{\bar{p}_{ts}^2}{\bar{\delta}_{s \to t}^2} + \frac{4n}{|\zeta|} \right) \log \frac{n(d_{\boldsymbol{y}} + d_{\boldsymbol{\lambda}})}{\epsilon}} + \exp(-O(M)),$$

where

$$\mu_{1,0} = \sqrt{8 \frac{\eta_A^2 + (1 - \eta_A)^2}{1 - \eta_A^2}}, \quad \tau = \frac{\eta_B \|\boldsymbol{c}^{\boldsymbol{y}}\|}{\kappa(1 - \eta_A)}.$$

*Proof.* We first have

$$\left\| \boldsymbol{v}^{(M+1)} - \mathrm{d}_{\boldsymbol{\lambda}_j} F(\boldsymbol{y}^*, \boldsymbol{\lambda}) \right\| \leq \left\| \boldsymbol{v}^{(M+1)} - \bar{\boldsymbol{B}} \sum_{i=0}^M \bar{\boldsymbol{A}}^i \boldsymbol{c}^{\boldsymbol{y}} \right\| + \left\| \bar{\boldsymbol{B}} \sum_{i=M+1}^\infty \bar{\boldsymbol{A}}^i \boldsymbol{c}^{\boldsymbol{y}} \right\|.$$

Here, we can bound the second term by

$$\left\| \bar{\boldsymbol{B}} \sum_{i=M+1}^\infty \bar{\boldsymbol{A}}^i \boldsymbol{c}^{\boldsymbol{y}} \right\| \leq \eta_B \|\boldsymbol{c}^{\boldsymbol{y}}\| \sum_{i=M+1}^\infty \eta_A^i \leq \frac{\eta_B \|\boldsymbol{c}^{\boldsymbol{y}}\|}{1 - \eta_A} \eta_A^{M+1} = \exp\left(-O(M)\right).$$

We can bound the first term by using Matrix Azuma's inequality; with probability at least $1 - \epsilon$, we have

$$\left\| \boldsymbol{v}^{(M+1)} - \bar{\boldsymbol{B}} \sum_{i=0}^M \bar{\boldsymbol{A}}^i \boldsymbol{c}^{\boldsymbol{y}} \right\| \leq \sqrt{8\sigma^2 \frac{n(d_{\boldsymbol{y}} + d_{\boldsymbol{\lambda}})}{\epsilon}},$$

where

$$\frac{\sigma^2}{\|\boldsymbol{c}^{\boldsymbol{y}}\|^2} \leq \sum_{i=0}^M \sum_{s,t=1}^n \left\| \boldsymbol{X}_{B,st}^{(i)} \right\|_2^2 + \sum_{i=0}^{M-1} \sum_{s,t=1}^n \left\| \boldsymbol{X}_{A,st}^{(i)} \right\|_2^2$$

$$+ \sum_{i=0}^M \sum_{t=1}^n \sum_{\xi \in \zeta_t^{(2i)}} \left\| \boldsymbol{Y}_{B,t}^{(i,\xi)} \right\|_2^2 + \sum_{i=0}^{M-1} \sum_{t=1}^n \sum_{\xi \in \zeta_t^{(2i+1)}} \left\| \boldsymbol{Y}_{A,t}^{(i,\xi)} \right\|_2^2$$

$$\leq \sum_{i=0}^M \sum_{s,t=1}^n \frac{\eta_B^2}{\kappa^2} \frac{\bar{p}_{ts}^2}{\bar{\delta}_{s \to t}^2} \eta_A^{2i} + \sum_{i=0}^{M-1} \sum_{s,t=1}^n \frac{\eta_B^2}{\kappa^2} \frac{\bar{p}_{ts}^2}{\bar{\delta}_{s \to t}^2} \left( \frac{\eta_A}{1 - \eta_A} \right)^2 \eta_A^{2i}$$

$$+ \sum_{i=0}^M \sum_{t=1}^n \sum_{\xi \in \zeta_t^{(2i)}} \frac{4\eta_B^2}{\kappa^2 |\zeta_t^{(2i)}|^2} \eta_A^{2i} + \sum_{i=0}^{M-1} \sum_{t=1}^n \sum_{\xi \in \zeta_t^{(2i+1)}} \frac{4\eta_B^2}{\kappa^2 |\zeta_t^{(2i+1)}|^2} \left( \frac{\eta_A}{1 - \eta_A} \right)^2 \eta_A^{2i} + \exp(-O(M))$$

$$= \frac{\eta_B^2}{\kappa^2} \left[ \sum_{s,t=1}^n \frac{\bar{p}_{ts}^2}{\bar{\delta}_{s \to t}^2} + \sum_{t=1}^n \frac{4}{|\zeta_t^{(2i)}|} \right] \frac{1 - \eta_A^{2(M+1)}}{1 - \eta_A^2}$$

$$+ \frac{\eta_B^2}{\kappa^2} \left[ \sum_{s,t=1}^n \frac{\bar{p}_{ts}^2}{\bar{\delta}_{s \to t}^2} + \sum_{t=1}^n \frac{4}{|\zeta_t^{(2i+1)}|} \right] \left( \frac{\eta_A}{1 - \eta_A} \right)^2 \frac{1 - \eta_A^{2M}}{1 - \eta_A^2} + \exp(-O(M))$$

$$= \frac{\eta_B^2}{\kappa^2} \left[ \sum_{s,t=1}^n \frac{\bar{p}_{ts}^2}{\bar{\delta}_{s \to t}^2} + \sum_{t=1}^n \frac{4}{|\zeta_t^{(2i)}|} \right] \frac{1}{1 - \eta_A^2}$$

$$+ \frac{\eta_B^2}{\kappa^2} \left[ \sum_{s,t=1}^n \frac{\bar{p}_{ts}^2}{\bar{\delta}_{s \to t}^2} + \sum_{t=1}^n \frac{4}{|\zeta_t^{(2i+1)}|} \right] \left( \frac{\eta_A}{1 - \eta_A} \right)^2 \frac{1}{1 - \eta_A^2} + \exp(-O(M)).$$

When $|\zeta_t^{(2i)}| = |\zeta_t^{(2i+1)}| = b$ for any $t$ and $i$, we further have

$$\sigma^2 \leq \frac{\eta_B^2 \|\boldsymbol{c}^{\boldsymbol{y}}\|^2}{\kappa^2(1 - \eta_A^2)} \left( 1 + \frac{\eta_A^2}{(1 - \eta_A)^2} \right) \left[ \sum_{s,t=1}^n \frac{\bar{p}_{ts}^2}{\bar{\delta}_{s \to t}^2} + \frac{4n}{b} \right] + \exp(-O(M)).$$

$\square$

## C.5  COMPARISON OF $\mu_{\alpha,\beta}$ AND $\mu_{0,1}$

The estimation errors of VR-HGP and naive HGP are dominated by their scaling factors.

$$\mu_{\alpha,\beta} = \sqrt{8\frac{1-\alpha}{1+\alpha}\left(1 + \frac{1+\alpha(1-\beta+\beta\eta_A)}{1-\alpha(1-\beta+\beta\eta_A)}\frac{\beta^2\eta_A^2}{1-(1-\beta+\beta\eta_A)^2}\right)},$$

$$\mu_{1,0} = \sqrt{8\frac{\eta_A^2 + (1-\eta_A)^2}{1-\eta_A^2}}.$$

Figure 1 shows that $\mu_{\alpha,\beta}$ is a few times smaller than $\mu_{1,0}$ for any $\eta_A \in (0,1)$ if we choose $\alpha$ close to one and $\beta$ close to zero. This result indicates that the error of VR-HGP can be a few times smaller than the one of naive HGP for sufficiently large $M$ where the diminishing term $\exp(-O(M))$ is negligibly small.

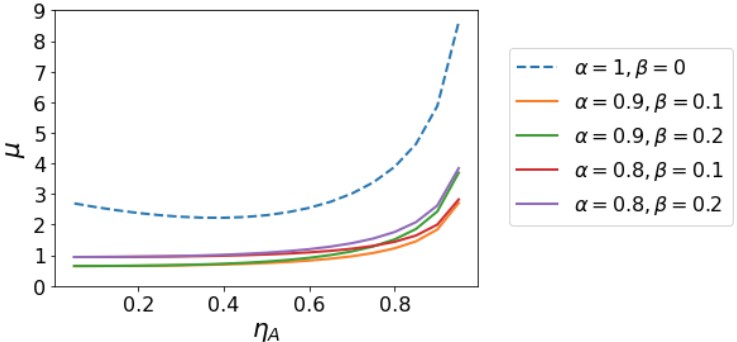

Figure 1: Comparisons of $\mu_{\alpha,\beta}$ and $\mu_{1,0}$ for $\eta_A \in (0,1)$.

## C.6  COMPARISON OF $\alpha$ AND $\beta$

We empirically evaluated the advantages of VR-HGP in stochastic communications as well as found that $(\alpha,\beta) = (0.9, 0.1)$ performed well in practice.

We compared the $\ell_2$ norm between of the hyper-gradient estimation $\boldsymbol{v}^{(m)}$ at the $m$-th round of HGP and the true hyper-gradient $d_{\boldsymbol{\lambda}}\bar{F}(\boldsymbol{x}^*, \boldsymbol{\lambda})$ which computed using the explicit $(\boldsymbol{I} - \bar{\boldsymbol{A}})^{-1}$. We made a synthetic one-dimensional dataset with two classes by randomly selecting two digits from MNIST and averaging the inputs of each sample. We let $n = 3$ clients performed 500 iterations of Eq. (4b) ensuring the convergence of SGP. For all $i \in [n]$, we used the binary cross-entropy loss for $f_i$ and $F_i$ computed on local training and validation datasets with 100 samples, respectively. We adopted `StoU` communication network presented in Section 6. In order to purely evaluate the effect of edge stochasticity $\delta_{j\to i}^{(m)}/\bar{\delta}_{j\to i}$, which we pointed the source of the high variance in Section 4.2, we excluded the randomness of minibatches $\zeta$ by adopting $|\zeta_i^{(t)}| = 100$ for all time steps in SGP and HGP and by using the true $\bar{p}_{ij}$ and $\bar{\delta}_{j\to i}$ for all $i, j \in [n]$. We computed $d_{\boldsymbol{\lambda}}\bar{F}(\boldsymbol{x}^*, \boldsymbol{\lambda})$ from the explicit computation of $\bar{\boldsymbol{B}}(\boldsymbol{I} - \bar{\boldsymbol{A}})^{-1}\boldsymbol{c}^y + \boldsymbol{c}^\lambda$ using expected values of $\bar{p}_{ij}$ and $\bar{\delta}_{j\to i}$ for all $i, j \in [n]$. The HGP was conducted to obtain $\boldsymbol{v}^{(m)}$ after the iterations of SGP using $M = 500$ and the alternative samplings, i.e., $\tilde{\boldsymbol{A}}^{(2m+1)}$ and $\tilde{\boldsymbol{B}}^{(2m)}$ for $m = 0, \ldots, M - 1$.

Fig. 2 shows VR-HGP with $(\alpha,\beta) = (0.9, 0.1)$ provided the smallest estimation error and the larger number of estimation rounds tends to have smaller error. However HGP, which is a special case of VR-HGP with parameters $(\alpha,\beta) = (1.0, 0.0)$, failed to attain smaller error than the well-tuned VR-HGP with $(\alpha,\beta) = (0.9, 0.1)$. This larger estimation error was also observed in experiments with different random seeds. We also observed that HGP could not reduce the estimation error after around $m = 5$ indicating the larger number of rounds does not always help the better estimation in HGP on stochastic communication networks.

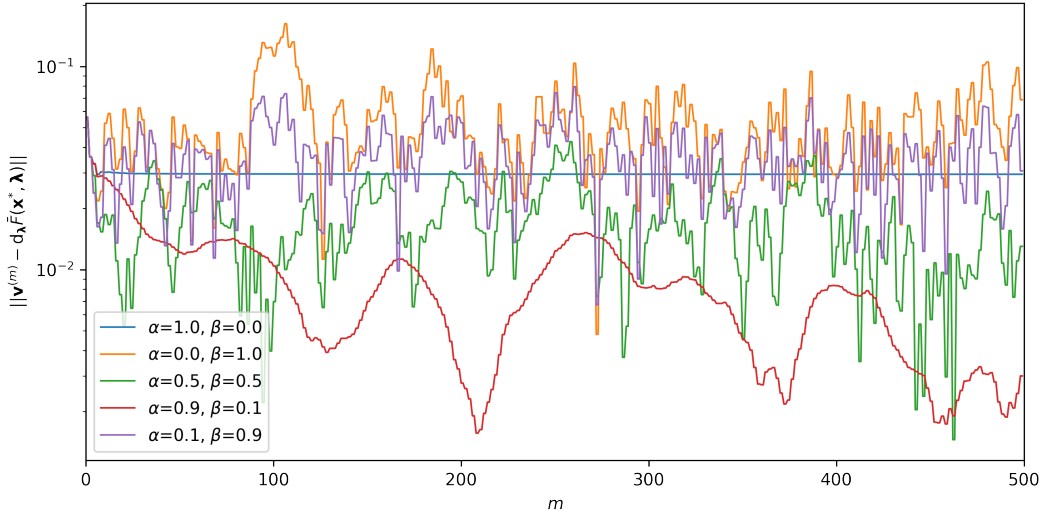

Figure 2: $\ell_2$ norm between the estimation of VR-HGP $\boldsymbol{v}^{(m)}$ and the true hyper-gradient $\mathrm{d}_{\boldsymbol{\lambda}}\bar{F}(\boldsymbol{x}^*, \boldsymbol{\lambda})$ at the $m$-th estimation round with different combinations of $\alpha$ and $\beta$.

## C.7 RELAXATION OF CONVERGENCE TO THE STATIONARY POINT

While VR-HGP relies on the assumption that the unique stationary point $\boldsymbol{y}^*$ is available, a client may only have $\boldsymbol{y}_i^{(T)} \neq \boldsymbol{y}^*$ in a practical case where the inner-problem is solved by a finite $T$ of SGP iterations. We show that this assumption can be relaxed by adopting an extra smoothness assumption below.

**Assumption 10.** There exist finite positive constants $L_y, L_\lambda$ such that for any $i \in [n]$ and for any $\boldsymbol{y}_i, \boldsymbol{y}_i'$,

$$\sup_{\boldsymbol{\lambda}_i, \xi} \|\partial_{\boldsymbol{y}_i} g_i(\boldsymbol{y}_i, \boldsymbol{\lambda}_i, \xi) - \partial_{\boldsymbol{y}_i} g_i(\boldsymbol{y}_i', \boldsymbol{\lambda}_i, \xi)\|_2 \leq L_y \|\boldsymbol{y}_i - \boldsymbol{y}_i'\|,$$

$$\sup_{\boldsymbol{\lambda}_i, \xi} \|\partial_{\boldsymbol{y}_i} h_i(\boldsymbol{y}_i, \boldsymbol{\lambda}_i, \xi) - \partial_{\boldsymbol{y}_i} h_i(\boldsymbol{y}_i', \boldsymbol{\lambda}_i, \xi)\|_2 \leq L_y \|\boldsymbol{y}_i - \boldsymbol{y}_i'\|,$$

$$\sup_{\boldsymbol{\lambda}_i, \xi} \|\partial_{\boldsymbol{\lambda}_i} g_i(\boldsymbol{y}_i, \boldsymbol{\lambda}_i, \xi) - \partial_{\boldsymbol{\lambda}_i} g_i(\boldsymbol{y}_i', \boldsymbol{\lambda}_i, \xi)\|_2 \leq \eta_B L_\lambda \|\boldsymbol{y}_i - \boldsymbol{y}_i'\|,$$

$$\sup_{\boldsymbol{\lambda}_i, \xi} \|\partial_{\boldsymbol{\lambda}_i} h_i(\boldsymbol{y}_i, \boldsymbol{\lambda}_i, \xi) - \partial_{\boldsymbol{\lambda}_i} h_i(\boldsymbol{y}_i', \boldsymbol{\lambda}_i, \xi)\|_2 \leq \eta_B L_\lambda \|\boldsymbol{y}_i - \boldsymbol{y}_i'\|.$$

Below, we show that the error between $\boldsymbol{y}_i^{(T)}$ and $\boldsymbol{y}^*$ induces a bias to Theorem 1.

**Theorem 11.** Let $\tilde{\boldsymbol{v}}^{(M+1)}$ be the estimate of $\boldsymbol{v}^{(M+1)}$ obtained by VR-HGP using $\boldsymbol{y}^{(T)}$ instead of $\boldsymbol{y}^*$. Suppose Assumptions 1–10 hold true, $|\zeta_t^{(2i)}| = |\zeta_t^{(2i+1)}| = b$ for any $t$ and $i$, and $\alpha, \beta \in (0, 1)$. Then, with probability at least $1 - \epsilon$, we have

$$\left\| \tilde{\boldsymbol{v}}^{(M+1)} + \boldsymbol{c}^{\boldsymbol{\lambda}} - \mathrm{d}_{\boldsymbol{\lambda}_j} F(\boldsymbol{y}^*, \boldsymbol{\lambda}) \right\|$$

$$\leq \mu_{\alpha,\beta} \tau \sqrt{\left( \sum_{s,t=1}^{n} \frac{\bar{p}_{ts}^2}{\bar{\delta}_{s\to t}^2} + \frac{4n}{b} \right) \log \frac{n(d_{\boldsymbol{y}} + d_{\boldsymbol{\lambda}})}{\epsilon}} + \frac{\eta_B \|\boldsymbol{c}^{\boldsymbol{y}}\|}{1 - \eta_A} (L_y + L_\lambda) G + \exp(-O(M)),$$

where

$$G = n \sum_{i=1}^{n} \left\| \boldsymbol{y}_i^* - \boldsymbol{y}_i^{(T)} \right\|.$$

*Proof.* To compute $\tilde{\boldsymbol{v}}^{(M+1)}$ using VR-HGP, $\bar{\boldsymbol{A}}$ and $\bar{\boldsymbol{B}}$ are estimated by

$$\tilde{\boldsymbol{A}} = \left[ \mathbb{1}_{ji} \tilde{\boldsymbol{A}}_i^\psi + \bar{p}_{ji} \tilde{\boldsymbol{A}}_j^\varphi \right]_{ji} \in \mathbb{R}^{nd_{\boldsymbol{y}} \times nd_{\boldsymbol{y}}},$$

$$\tilde{\boldsymbol{A}}_i^\psi = \frac{1}{|\zeta_i|} \sum_{\xi \in \zeta_i} \partial_{\boldsymbol{y}_i} g_i(\boldsymbol{y}_i^{(T)}; \boldsymbol{\lambda}_i, \xi) \in \mathbb{R}^{d_{\boldsymbol{y}} \times d_{\boldsymbol{y}}}, \quad \tilde{\boldsymbol{A}}_j^\varphi = \frac{1}{|\zeta_j|} \sum_{\xi \in \zeta_j} \partial_{\boldsymbol{y}_j} h_j(\boldsymbol{y}_j^{(T)}; \boldsymbol{\lambda}_j, \xi) \in \mathbb{R}^{d_{\boldsymbol{y}} \times d_{\boldsymbol{y}}},$$

$$\tilde{\boldsymbol{B}} = \left[ \mathbb{1}_{ji} \tilde{\boldsymbol{B}}_i^\psi + \bar{p}_{ji} \tilde{\boldsymbol{B}}_j^\varphi \right]_{ji} \in \mathbb{R}^{nd_{\boldsymbol{\lambda}} \times nd_{\boldsymbol{y}}},$$

$$\tilde{\boldsymbol{B}}_i^\psi = \frac{1}{|\zeta_i|} \sum_{\xi \in \zeta_i} \partial_{\boldsymbol{\lambda}_i} h_i(\boldsymbol{y}_i^{(T)}; \boldsymbol{\lambda}_i, \xi) \in \mathbb{R}^{d_{\boldsymbol{\lambda}} \times d_{\boldsymbol{y}}}, \quad \tilde{\boldsymbol{B}}_j^\varphi = \frac{1}{|\zeta_j|} \sum_{\xi \in \zeta_j} \partial_{\boldsymbol{\lambda}_j} \varphi_{ji}(\boldsymbol{y}_j^{(T)}; \boldsymbol{\lambda}_j, \xi) \in \mathbb{R}^{d_{\boldsymbol{\lambda}} \times d_{\boldsymbol{y}}}.$$

We can decompose the difference $\tilde{\boldsymbol{A}} - \bar{\boldsymbol{A}}$ and $\tilde{\boldsymbol{B}} - \bar{\boldsymbol{B}}$ as

$$\tilde{\boldsymbol{A}} - \bar{\boldsymbol{A}} = \left( \tilde{\boldsymbol{A}} - \hat{\boldsymbol{A}} \right) + \left( \hat{\boldsymbol{A}} - \bar{\boldsymbol{A}} \right), \qquad \tilde{\boldsymbol{B}} - \bar{\boldsymbol{B}} = \left( \tilde{\boldsymbol{B}} - \hat{\boldsymbol{B}} \right) + \left( \hat{\boldsymbol{B}} - \bar{\boldsymbol{B}} \right).$$

Here, by Assumption 10, we have

$$\left\| \tilde{\boldsymbol{A}} - \hat{\boldsymbol{A}} \right\|_2 \le L_y n \sum_{i=1}^n \left\| \boldsymbol{y}_i^* - \boldsymbol{y}_i^{(T)} \right\| = L_y G, \qquad \left\| \tilde{\boldsymbol{B}} - \hat{\boldsymbol{B}} \right\|_2 \le \eta_B L_\lambda n \sum_{i=1}^n \left\| \boldsymbol{y}_i^* - \boldsymbol{y}_i^{(T)} \right\| = \eta_B L_\lambda G.$$

By using these expressions, we can derive the expression similar to Lemma 5 as

$$\tilde{\boldsymbol{v}}^{(M+1)} - \bar{\boldsymbol{B}} \sum_{i=0}^M \bar{\boldsymbol{A}}^i \boldsymbol{c}^{\boldsymbol{y}}$$

$$= \sum_{i=0}^M \sum_{s,t=1}^n \boldsymbol{X}_{B,st}^{(i)} \boldsymbol{c}^{\boldsymbol{y}} + \sum_{i=0}^M \sum_{t=1}^n \sum_{\xi \in \zeta_s^{(2i)}} \boldsymbol{Y}_{B,t}^{(i,\xi)} \boldsymbol{c}^{\boldsymbol{y}} + \sum_{i=0}^{M-1} \sum_{s,t=1}^n \tilde{\boldsymbol{X}}_{A,st}^{(i)} \boldsymbol{c}^{\boldsymbol{y}} + \sum_{i=0}^{M-1} \sum_{t=1}^n \sum_{\xi \in \zeta_s^{(2i+1)}} \tilde{\boldsymbol{Y}}_{A,t}^{(i,\xi)} \boldsymbol{c}^{\boldsymbol{y}}$$

$$\underbrace{+ \sum_{i=0}^M (\tilde{\boldsymbol{B}}^{(2i)} - \hat{\boldsymbol{B}}) \boldsymbol{R}_3^{(i,M)} \boldsymbol{c}^{\boldsymbol{y}} + \sum_{i=0}^{M-1} \left( \tilde{\boldsymbol{L}}_4^{(i,M)} (\tilde{\boldsymbol{A}}^{(2i+1)} - \hat{\boldsymbol{A}}) \bar{\boldsymbol{A}}^i + \tilde{\boldsymbol{L}}_5^{(i,M)} (\tilde{\boldsymbol{A}}^{(2i+1)} - \hat{\boldsymbol{A}}) \boldsymbol{R}_1^{(i)} \right) \boldsymbol{c}^{\boldsymbol{y}}}_{\text{(Bias)}},$$

where the last line corresponds to the bias induced by the use of $\boldsymbol{y}^{(T)}$ instead of $\boldsymbol{y}^*$, and

$$\tilde{\boldsymbol{L}}_4^{(i,M)} = \sum_{j=i+1}^M \alpha^{M-j+1} \tilde{\boldsymbol{B}}^{(2j)} \left[ \prod_{m=i+1}^{j-1} \tilde{\boldsymbol{A}}^{(2m+1)} \right]$$

$$+ (1-\alpha)(1-\beta) \sum_{j=i+1}^M \alpha^{M-j} \tilde{\boldsymbol{B}}^{(2j)} \sum_{k=i+1}^j \left[ \prod_{m=k}^{j-1} \left( (1-\beta)\boldsymbol{I} + \beta \tilde{\boldsymbol{A}}^{(2m+1)} \right) \right] \left[ \prod_{m=i+1}^{k-1} \tilde{\boldsymbol{A}}^{(2m+1)} \right],$$

$$\tilde{\boldsymbol{L}}_5^{(i,M)} = (1-\alpha)\beta \sum_{j=i+1}^M \alpha^{M-j} \tilde{\boldsymbol{B}}^{(2j)} \left[ \prod_{m=i+1}^{j-1} \left( (1-\beta)\boldsymbol{I} + \beta \tilde{\boldsymbol{A}}^{(2m+1)} \right) \right],$$

$$\tilde{\boldsymbol{X}}_{A,st}^{(i)} = \tilde{\boldsymbol{L}}_4^{(i,M)} \left( \boldsymbol{e}_t \boldsymbol{e}_s^\top \otimes \left( \frac{\delta_{s \to t}^{(2i+1)}}{\bar{\delta}_{s \to t}} - 1 \right) \bar{p}_{ts} \hat{\boldsymbol{A}}_t^{\varphi(2i+1)} \right) \bar{\boldsymbol{A}}^i + \tilde{\boldsymbol{L}}_5^{(i,M)} \left( \boldsymbol{e}_t \boldsymbol{e}_s^\top \otimes \left( \frac{\delta_{s \to t}^{(2i+1)}}{\bar{\delta}_{s \to t}} - 1 \right) \bar{p}_{ts} \hat{\boldsymbol{A}}_t^{\varphi(2i+1)} \right) \boldsymbol{R}_1^{(i)},$$

$$\tilde{\boldsymbol{Y}}_{B,t}^{(i)} = \hat{\boldsymbol{L}}_4^{(i,M)} \sum_{s=1}^n \left( \boldsymbol{e}_t \boldsymbol{e}_s^\top \otimes \frac{1}{|\zeta_t^{(2i+1)}|} \left( \mathbb{1}_{ts} \partial_{\boldsymbol{y}_t} g_t(\boldsymbol{y}_t^*, \boldsymbol{\lambda}_t, \xi) + \bar{p}_{ts} \partial_{\boldsymbol{y}_t} h_t(\boldsymbol{y}_t^*, \boldsymbol{\lambda}_t, \xi) - \mathbb{1}_{ts} \bar{\boldsymbol{A}}_t^\psi - \bar{p}_{ts} \bar{\boldsymbol{A}}_t^\varphi \right) \right) \bar{\boldsymbol{A}}^i$$

$$+ \tilde{\boldsymbol{L}}_5^{(i,M)} \sum_{s=1}^n \left( \boldsymbol{e}_t \boldsymbol{e}_s^\top \otimes \frac{1}{|\zeta_t^{(2i+1)}|} \left( \mathbb{1}_{ts} \partial_{\boldsymbol{y}_t} g_t(\boldsymbol{y}_t^*, \boldsymbol{\lambda}_t, \xi) + \bar{p}_{ts} \partial_{\boldsymbol{y}_t} h_t(\boldsymbol{y}_t^*, \boldsymbol{\lambda}_t, \xi) - \mathbb{1}_{ts} \bar{\boldsymbol{A}}_t^\psi - \bar{p}_{ts} \bar{\boldsymbol{A}}_t^\varphi \right) \right) \boldsymbol{R}_1^{(i)}.$$

Then, we can bound the bias as

$$
\begin{aligned}
\frac{\|(\text{Bias})\|}{\|\boldsymbol{c^y}\|} &\leq \sum_{i=0}^{M} \left\| \tilde{\boldsymbol{B}}^{(2i)} - \hat{\boldsymbol{B}} \right\|_2 \left\| \boldsymbol{R}_3^{(i,M)} \right\|_2 \\
&\quad + \sum_{i=0}^{M-1} \left( \left\| \tilde{\boldsymbol{L}}_4^{(i,M)} \right\|_2 \left\| \tilde{\boldsymbol{A}}^{(2i+1)} - \hat{\boldsymbol{A}} \right\|_2 \left\| \bar{\boldsymbol{A}} \right\|_2^i + \left\| \tilde{\boldsymbol{L}}_5^{(i,M)} \right\|_2 \left\| \tilde{\boldsymbol{A}}^{(2i+1)} - \hat{\boldsymbol{A}} \right\|_2 \left\| \boldsymbol{R}_1^{(i)} \right\|_2 \right) \\
&\leq \eta_B L_\lambda G \frac{1-\alpha}{1-\eta_A} \sum_{i=0}^{M} \alpha^{M-i} \\
&\quad + L_y G \frac{\eta_B}{1-\eta_A} \frac{(1-\alpha)\beta}{\alpha - (1-\beta+\beta\eta_A)} \sum_{i=0}^{M-1} \left( \alpha^{M-i} - (1-\beta+\beta\eta_A)^{M-i} \right) + \exp\left(-O(M)\right) \\
&= \frac{\eta_B}{1-\eta_A}(L_y + L_\lambda)G + \exp\left(-O(M)\right).
\end{aligned}
$$

$\square$

## D  DETAILED EXPERIMENTAL SETTINGS

The experiments in Section 6 followed the settings of EMNIST (Cohen et al., 2017) classification in Marfoq et al. (2021), unless otherwise mentioned.

**Communication networks**  We simulated four communication networks on which the clients perform the distributed learning: fully-connected (`FC`), static undirected (`FixU`), stochastic undirected (`StoU`), and stochastic directed (`StoD`).

`FC` allows clients to communicate with all the other clients in all the time steps, i.e. $\delta_{i \to j}^{(t)} = 1$ for all $i, j \in [n]$ and $t \in \mathbb{N}$. `FixU` uses time-invariant and sparse undirected communication network simulated by a binomial Erdős-Rényi graph (Erdős & Rényi, 1959) with parameter $p = 0.4$ adding the self-loop edges. Following the setting in Marfoq et al. (2021), we generated a doubly stochastic mixing matrix using the Fast Mixing Markov Chain (Boyd et al., 2003) rule. `StoU` uses stochastic and undirected network in which any undirected edge $\delta_{j \to i}^{(t)} = \delta_{i \to j}^{(t)}$ independently realizes at each step with the probability $\bar{\delta}_{j \to i} \in [0, 1]$. In `StoD` each direction of edges $\delta_{j \to i}^{(t)}$ are independently sampled at probability $\bar{\delta}_{j \to i}$, forming stochastic and directed network. `StoD` forms the asymmetric expected mixing matrix given by the `StoD` network is asymmetric representing the communication bias between the clients; some clients may communicate more infrequently than others due to bottlenecks in physical network environments or long computation times of local updates due to poor computational resources. We sampled $\bar{\delta}_{j \to i}$ from the uniform distribution with $[0.4, 0.8]$ both in `StoU` and `StoD`

**Proposed approaches**  We solved personalization of classification models using three different formulation: PDBO-MTL, PDBO-DA, and PDBO-MTL&DA.

For PDBO-DA, we optimize the pseudo sampling rate to recover data augmentation-based personalization (Duan et al., 2019; Zhao et al., 2018). PDBO-DA optimize $\boldsymbol{\lambda}_i^C \in \mathbb{R}^C$ to learn the label-wise weight vector $C\text{Softmax}(\boldsymbol{\lambda}_i) \in [0, C]^C$. In the inner-problem, the losses of instances labeled as $c \in [C]$ are multiplied by the $c$-th element of the weight vector.

PDBO-MTL is obtained by applying PDBO to FedEM Marfoq et al. (2021). PDBO-MTL lets each client train an ensemble classifier that outputs weighted average predictions across $K = 3$ of CNNs. We trained CNN parameters as the inner-problem and optimized the hyperparameters $\boldsymbol{\lambda}_i^K \in \mathbb{R}^K$ to obtain ensemble weight vector $\text{Softmax}(\boldsymbol{\lambda}_i) \in [0, 1]^K$.

PDBO-MTL&DA combines PDBO-DA and PDBO-MTL optimizing $[\boldsymbol{\lambda}_i^{K\top} \ \boldsymbol{\lambda}_i^{C\top}]^\top \in \mathbb{R}^{C+K}$ to obtain both the label-weight and model-weight.

For all $i \in [n]$ in the outer-problem, we ran 20 outer-steps of Adam (Kingma & Ba, 2015) iterations with $(\beta_1, \beta_2) = (0.9, 0.999)$ from the initial hyperparameters $\boldsymbol{0}_C$, $\boldsymbol{0}_K$, and $\boldsymbol{0}_{C+K}$ for PDBO-DA,

Table 2: Parameters for the outer-problems in Section 6

| Network | Method | L2 reg. rate | Hyper-learning rate |
|---|---|---|---|
| FC and FixU | PDBO-DA | 0 | 0.1 |
| | PDBO-MTL | 0.01 | 1.0 |
| | PDBO-MTL&DA | 0.01 for $\boldsymbol{\lambda}_i^K$
0.0005 for $\boldsymbol{\lambda}_i^C$ | 1.0 for $\boldsymbol{\lambda}_i^K$
0.1 for $\boldsymbol{\lambda}_i^C$ |
| StoU and StoD | PDBO-DA | 0 | 0.1 |
| | PDBO-MTL | 0.01 | 0.1 |
| | PDBO-MTL&DA | 0.01 for $\boldsymbol{\lambda}_i^K$
0.0005 for $\boldsymbol{\lambda}_i^C$ | 0.1 for $\boldsymbol{\lambda}_i^K$
0.1 for $\boldsymbol{\lambda}_i^C$ |

PDBO-MTL, and PDBO-MTL&DA, respectively. For Adam optimizer, we adopted different learning rate shown in Table 2 (Hyper-learning rate). We adopt HGP for FC and FixU setting, and VR-HGP with $(\alpha, \beta) = (0.9, 0.1)$ for StoU and StoD settings. Both HGP and VR-HGP ran $M = 200$ estimation steps using iteration Eq. (4b) in all the settings. We also made a practical modification in HGP to sample $\tilde{\boldsymbol{A}}^{(m)}$ and $\tilde{\boldsymbol{B}}^{(m)}$ together at the single $m$-th round, which leads the same length of the Neumann series with the half sampling costs of the original HGP, although they are no more unbiased. For all the approaches the cases and for all $i \in [n]$, we used the average cross-entropy loss over the local train dataset of the $i$-th node and L2 regularization loss of $\boldsymbol{\lambda}_i$ for $F_i$ with the rates shown in Table 2 (L2 reg. rate). We reported the mean test accuracy of an intermediate step that had maximum validation accuracy (i.e., early stopping) which was sampled independently from the train dataset as described in Appendix D.

**Baseline approaches** We compared our approaches with baselines for each communication setting.

For FC and FixU settings, we compared with several personalization approaches: a personalized model trained only on the local dataset (Local), FedAvg with local tuning (FedAvg+) (Jiang et al., 2019), Clustered-FL (Sattler et al., 2020), pFedMe (T Dinh et al., 2020), and centralized and decentralized version of FedEM adopted in Marfoq et al. (2021). We also trained global models using SGP (Nedić & Olshevsky, 2016; Assran et al., 2019) and FedProx (Li et al., 2020). From the fact that SGP recovers FedAvg and DSGD on FC and FixU, respectively, we treat them as equivalent approaches. All the approaches on FC and FixU followed the training procedure with epoch-wise communication in Marfoq et al. (2021) while using Eq. (4b) for HGP computation. And any method ran on StoU and StoD adopted the SGP iteration (Eq. (4b)) with $T = 600$ steps, batch size $|\zeta_i| = 128$, L2 regularization with 0.001 decay. For SGP StoU and StoD, we adopted the learning rate $\alpha_i = 0.05$ for SGP, Local, and PDBO-DA, $\alpha_i = 0.25$ for PDBO-MTL and PDBO-MTL&DA. Those learning rates were scheduled to be multiplied by 0.1 at $t = 500, 550$. As we have no baseline ensemble model approach (i.e. FedEM) to be compared to our PDBO-MTL and PDBO-MTL&DA in StoU and StoD, we also examined our performance improvement from the initial hyperparameter. We confirmed PDBO-MTL and PDBO-MTL&DA improved their test accuracy from the initial hyperparameter both in StoU and StoD, confirming the performance gain of PDBO-MTL and PDBO-MTL&DA from SGP were not solely due to their differences in architectures and learning rates.

**Dataset and model** We adopted the procedure of generating a federated version of EMNIST in Marfoq et al. (2021) except for train and validation split. In our experiments, we consider 10% of the EMNIST dataset as in that were partitioned according to Dirichlet allocation of parameter $\alpha = 0.4$ over $n = 100$ clients as in Marfoq et al. (2021). We randomly selected 20% of the obtained dataset to make a validation dataset. We use the validation dataset only for the early stopping in outer-optimization of PDBO-DA, PDBO-MTL, and PDBO-MTL&DA. We trained the same CNN in Marfoq et al. (2021) for all the baselines with a single model and PDBO-DA, and for base-predictor of FedEM, PDBO-MTL, and PDBO-MTL&DA.

Table 3: Comparison of the gradient-based PDBO, CDBO, and CDBO-Local.

| Study | Bilevel problem | Communication network | Hyper-gradient | No $O\left(d_{\boldsymbol{x}} \times d_{\boldsymbol{\lambda}}\right)$ and $O\left(d_{\boldsymbol{x}} \times d_{\boldsymbol{x}}\right)$ | |
| --- | --- | --- | --- | --- | --- |
| | | | | in communication | in computation |
| **Ours** | PDBO | Stochastic directed | GlobalGrad | ✓ | ✓ |
| Chen et al. (2022) | CDBO | Static undirected | ClientGrad | | |
| Gao et al. (2022) | CDBO | Static undirected | LocalGrad | ✓ | ✓ |
| Yang et al. (2022) | CDBO | Static undirected | GlobalGrad | | |
| Tarzanagh et al. (2022) | CDBO | Centralized | GlobalGrad | ✓ | ✓ |
| Li et al. (2022) | CDBO-Local | Centralized | LocalGrad | ✓ | ✓ |
| Liu et al. (2022) | CDBO-Local | Static undirected | LocalGrad | ✓ | ✓ |
| Lu et al. (2022) | CDBO-Local | Static undirected | LocalGrad | ✓ | ✓ |

# E  GRADIENT-BASED DISTRIBUTED BILEVEL OPTIMIZATION

We compare concurrent studies of distributed bilevel optimization (Chen et al., 2022; Tarzanagh et al., 2022; Gao et al., 2022; Yang et al., 2022; Li et al., 2022; Liu et al., 2022; Lu et al., 2022) in terms of problem settings, applicability on communication networks, hyper-gradient value to estimate, and complexity in communication and computation.

**Bilevel problem setting**  We categorize them into two problems (Bilevel problem in Table 3): the consensus distributed bilevel optimization (CDBO) (Chen et al., 2022; Tarzanagh et al., 2022; Gao et al., 2022; Yang et al., 2022) and CDBO with the local inner-problem (CDBO-Local) (Li et al., 2022; Liu et al., 2022; Lu et al., 2022).

CDBO pursue consensus also in outer-problem, which can be obtained by imposing $\boldsymbol{\lambda}_i = \boldsymbol{\lambda}_j$ for all $i, j \in [n]$ on PDBO outer-problem (Eq. (5-left)):

$$\min_{\substack{\boldsymbol{\lambda}_i \\ \boldsymbol{\lambda}_i = \boldsymbol{\lambda}_j, \forall j}} \frac{1}{n} \sum_{i=1}^{n} F_i\left(\boldsymbol{x}_i^*\left(\boldsymbol{\lambda}_1, \ldots, \boldsymbol{\lambda}_n\right), \boldsymbol{\lambda}_i\right), \text{ s.t. } \boldsymbol{x}_i^* = \underset{\substack{\boldsymbol{x}_i \\ \boldsymbol{x}_i = \boldsymbol{x}_j, \forall j}}{\arg\min} \frac{1}{n} \sum_{i=1}^{n} \mathbb{E}_{\xi_i}\left[f_i\left(\boldsymbol{x}_i, \boldsymbol{\lambda}_i; \xi_i\right)\right], \quad (39)$$

Chen et al. (2022); Tarzanagh et al. (2022); Gao et al. (2022); Yang et al. (2022) applied CDBO to hyperparameter (e.g. L2 regularization coefficient) optimization.

While CDBO-Local also requires consensus in the outer-problem as in CDBO, its inner-problem is a local optimization problem in which optimal parameters are independent of each other client, unlike PDBO and CDBO:

$$\min_{\substack{\boldsymbol{\lambda}_i \\ \boldsymbol{\lambda}_i = \boldsymbol{\lambda}_j, \forall j}} \frac{1}{n} \sum_{i=1}^{n} F_i\left(\boldsymbol{x}_i^*\left(\boldsymbol{\lambda}_i\right), \boldsymbol{\lambda}_i\right), \text{ s.t. } \boldsymbol{x}_i^* = \underset{\boldsymbol{x}_i}{\arg\min} \mathbb{E}_{\xi_i}\left[f_i\left(\boldsymbol{x}_i, \boldsymbol{\lambda}_i; \xi_i\right)\right], \quad (40)$$

Lu et al. (2022) demonstrated the ability of CDBO-Local problem to handle personalization tasks. However, no client in CDBO-Local can benefit from the others in the inner loop for better generalization. We note that in our PDBO, both outer and inner problems are optimized from the global information; the inner-parameter is trained for consensus among the clients and the outer parameter is optimized to improve the total performance across all the clients.

**Communication networks**  The communication networks can be categorized into stochastic directed, static undirected, and centralized (Communication network in Table 3).

Studies for CDBO (Chen et al., 2022; Gao et al., 2022; Yang et al., 2022) and CDBO-Local (Liu et al., 2022; Lu et al., 2022) suppose the communication networks are static and undirected. More specifically, they assume the weighted mixing matrix $\boldsymbol{P}^{(t)}$ to be a double-stochastic matrix at all time steps $t \in \mathbb{N}$ for the consensus of DSGD in the outer-problem (Liu et al., 2022; Lu et al., 2022) (i.e. $\boldsymbol{x}_i = \boldsymbol{x}_j, \forall i, j \in [n]$), and both in the outer-problem and inner-problem (Chen et al., 2022; Gao et al., 2022; Yang et al., 2022) (i.e. $\boldsymbol{x}_i = \boldsymbol{x}_j, \boldsymbol{\lambda}_i = \boldsymbol{\lambda}_j, \forall i, j \in [n]$).

Tarzanagh et al. (2022); Liu et al. (2022) addresses the consensus in the outer-problem by adopting centralized communication settings so that the single global hyperparameter are shared among the clients at every step.

Our HGP is the only method that runs even on stochastic and directed communication networks.

In terms of the consensus, we can relax the assumption of the static undirected communication in Chen et al. (2022); Gao et al. (2022); Yang et al. (2022); Liu et al. (2022); Lu et al. (2022) to the stochastic and directed networks by replacing DSGD with SGP for the inner-loop and outer-loop. However, in terms of the hyper-gradient estimation, we cannot naively replace the communication networks setting as discussed in Section 4.2.

**Hyper-gradient to estimate**    Both PDBO and CDBO require hyper-gradient estimation as they involve the interaction of clients in the inner-problem. However, the estimated hyper-gradient varies among the studies, so we categorize them into GlobalGrad, ClientGrad, and LocalGrad (Hyper-gradient in Table 3). Our HGP and Yang et al. (2022); Tarzanagh et al. (2022) aim at estimating the gradient of the average outer-objective across the client with respect to the hyperparameter of the client (GlobalGrad), i.e. $\mathrm{d}_{\boldsymbol{\lambda}_i} \bar{F}\left(\boldsymbol{x}^*\left(\boldsymbol{\lambda}\right), \boldsymbol{\lambda}\right) \in \mathbb{R}^{d_{\boldsymbol{\lambda}}}$.

Chen et al. (2022) estimate slightly different hyper-gradient, that is, the gradient of client outer-objective with respect to the hyperparameter of the client (ClientGrad), i.e. $\mathrm{d}_{\boldsymbol{\lambda}_i} F_i\left(\boldsymbol{x}_i^*\left(\boldsymbol{\lambda}\right), \boldsymbol{\lambda}_i\right)$. Unlike GlobalGrad, ClientGrad only lets the client know how the perturbation on the client's hyperparameter changes its own outer-objective. Thus the gradient step of the client hyperparameter using ClientGrad is not supposed to improve the performance of the others, which is not the case with GlobalGrad.

Gao et al. (2022) estimates the LocalGrad which is equivalent to the hyper-gradient estimation of SGD that estimates $\mathrm{d}_{\boldsymbol{\lambda}_i} F_i\left(\boldsymbol{x}_i^*\left(\boldsymbol{\lambda}_i\right), \boldsymbol{\lambda}_i\right)$. LocalGrad differs from ClientGrad because LocalGrad needs no communication because the optimal inner-parameter $\boldsymbol{x}_i^*$ is only parameterized by its hyperparameter $\boldsymbol{\lambda}_i$.

**Complexity in communication and computation**    For a fair comparison, we compare the complexity of communication and computation between methods that intend to estimate the same hyper-gradient. Note that we only focus on the requirement of computation or communication for the full Jacobian matrix as it is dominant in decentralized hyper-gradient estimation (rightmost two columns of Table 3).

No approach for LocalGrad involves the full Jacobian computation and communication as they can naively adopt efficient algorithms such as backward mode. For GlobalGrad, the algorithm proposed by Yang et al. (2022) is complex both in computation and communication as they involve computations and communications of full Jacobian matrix ($O\left(d_{\boldsymbol{y}} \times d_{\boldsymbol{\lambda}}\right)$) and Hessian matrix ($O\left(d_{\boldsymbol{y}} \times d_{\boldsymbol{y}}\right)$). Tarzanagh et al. (2022) and our HGP enjoys reasonable complexity because these methods avoid computation and communication of full Jacobian by using Jacobian-vector products.

# F    DETAILED ALGORITHMS

We provide an algorithm Alg. 1 which describes a case of PDBO in which outer-problem is solved by local SGD. We also describe the complete algorithms of SGP (Alg. 2) formulated by Eq. (2), HGP (Alg. 4) formulated by Eq. (12), VR-HGP (Alg. 5), formulated in Section 4.2 (Variance reduction), and the exact recurrent backpropagation (Alg. 3) formulated by Eq. (11). All the algorithms above are expected to run locally at every $i$-th client, showing how all clients collaboratively solve the PDBO (Eq. (7)) without any central orchestration.

For a better understanding, we describe below special notes on several lines in the algorithms that characterize our approach.

**Outer-loop in PDBO**    Let $\boldsymbol{\lambda}_i^{(s)}$ be a hyperparameter of the $i$-th client at the $s$-th outer-step. PDBO runs multiple outer-steps for $s = 0, \ldots, S - 1$ from a given initial hyperparameter $\boldsymbol{\lambda}_i^{(0)}$. Alg. 1 supposes $\boldsymbol{\lambda}_i^{(s)}$ is updated locally by SGD step (Line 7 in Alg. 1). As the output of HGP can be seen as an unbiased estimate of stochastic gradient, the convergence property of outer steps is simply given by the common convergence property of SGD whose noise is characterized by Theorem 1. We can also use other optimizers such as Adam (Kingma & Ba, 2015) for outer-steps, as we adopted in our experiments (Section 6).

**HGP vs. Exact recurrent backpropagation**    We explain the difference between HGP (Eq. (12)) and exact recurrent backpropagation (Eq. (11)) in algorithmic perspective.

As mentioned in Section 4.2 decentralization of exact recurrent backpropagation is impossible on directed communication networks. In exact recurrent backpropagation Line 9 and Line 17 in Alg. 3 require a client to receive the intermediate backpropagation vector $\boldsymbol{u}_j^{(m)}$ from clients such that $\delta_{i \to j}^{(m)} = 1$, indicating the $i$-th client needs to receive the message from whom the $i$-th client sent messages. This is possible only when the communication network is undirected or synchronized.

In our HGP, Line 9 and Line 17 in Alg. 4 let the $i$-th client to receive $\boldsymbol{u}_j^{(m)}$ from clients such that $\delta_{j \to i}^{(m)} = 1$, thus any client simply receives the information from all the client who is able to send to $i$. The estimation bias incurred by this simple modification is corrected according to the expected sending weight $\bar{p}_{ij} = \mathbb{E}_\delta[p_{ij}(\delta_i)]$ and receiving frequency $\bar{\delta}_{j \to i} = \mathbb{E}_\delta[\delta_{j \to i}]$ estimated through inner SGP iterations (Line 7 and Line 12 in Alg. 2).

Note that both HGP and the exact recurrent backpropagation enjoys cheap time complexity since the computations related to Jacobians, $\partial_{\boldsymbol{\lambda}_i} \boldsymbol{\varphi}_i\left(\boldsymbol{y}_i^*; \boldsymbol{\lambda}_i, \zeta_i^{(2m)}\right)\boldsymbol{u}_j^{(m)}$ and $\partial_{\boldsymbol{\lambda}_i} \boldsymbol{\psi}_i\left(\boldsymbol{y}_i^*; \boldsymbol{\lambda}_i, \zeta_i^{(2m)}\right)\boldsymbol{u}_i^{(m)}$, can be locally computed by Jacobian-vector product.

---

**Algorithm 1:** PDBO with SGD ran by the $i$-th client

---

**Input:** $\boldsymbol{y}_i^{(0)}, \boldsymbol{\lambda}_i^{(0)}, \alpha, \beta$

1 **foreach** $s = 0, \dots, S - 1$ **do**

    `// Solve the inner-problem`

2     $\boldsymbol{y}_i^{(T)}, \left\{\left(\bar{p}_{ij}, \bar{\delta}_{j \to i}\right)\right\}_{j \in [n]} \leftarrow \underbrace{\text{Alg. 2}\left(\boldsymbol{y}_i^{(0)}, \boldsymbol{\lambda}_i^{(s)}\right)}_{\text{SGP}}$

    `// Hyper-gradient estimation`

3     **if** $\alpha = 1$ **then**

4         $\widehat{\mathrm{d}_{\boldsymbol{\lambda}_i}}\bar{F} \leftarrow \underbrace{\text{Alg. 4}\left(\boldsymbol{y}_i^{(T)}, \boldsymbol{\lambda}_i^{(s)}, \left\{\left(\bar{p}_{ij}, \bar{\delta}_{j \to i}\right)\right\}_{j \in [n]}\right)}_{\text{HGP}}$

5     **else**

6         $\widehat{\mathrm{d}_{\boldsymbol{\lambda}_i}}\bar{F} \leftarrow \underbrace{\text{Alg. 5}\left(\boldsymbol{y}_i^{(T)}, \boldsymbol{\lambda}_i^{(s)}, \left\{\left(\bar{p}_{ij}, \bar{\delta}_{j \to i}\right)\right\}_{j \in [n]}, \alpha, \beta\right)}_{\text{VR-HGP}}$

    `// Run a local SGD outer-step`

7     $\boldsymbol{\lambda}_i^{(s+1)} \leftarrow \boldsymbol{\lambda}_i^{(s)} - \beta_i \widehat{\mathrm{d}_{\boldsymbol{\lambda}_i}}\bar{F}$

8 **return** $\boldsymbol{\lambda}_i^{(S)}$

---

---

**Algorithm 2:** SGP ran by the $i$-th client

---

**Input:** $\boldsymbol{y}_i^{(0)}, \boldsymbol{\lambda}_i$

// Initialize empirical estimates

1   $\bar{p}_{ij} \leftarrow 0, \ \forall j \in [n]$

2   $\bar{\delta}_{j \to i} \leftarrow 0, \ \forall j \in [n]$

3   **foreach** $t = 0, \ldots, T-1$ **do**

     // Sample a minibatch and communication edges

4      Sample $\zeta_i^{(t)}$ and $\delta_i^{(t)} = [\delta_{i \to 1}^{(t)} \ \cdots \ \delta_{i \to n}^{(t)}]^\top$

5      **foreach** $j$ s.t. $\delta_{i \to j}^{(t)} = 1$ **do**

6         Send $p_{ij}(\delta_i^{(t)}) \boldsymbol{\varphi}_i \left( \boldsymbol{y}_i^{(t)}; \boldsymbol{\lambda}_i, \zeta_i^{(t)} \right)$ to the $j$-th node

7         $\bar{p}_{ij} \mathrel{+}= p_{ij}(\delta_i^{(t)})$

8      $\boldsymbol{y}_i^{(t+1)} \leftarrow \boldsymbol{0}_{d_{\boldsymbol{y}}}$

9      **foreach** $j$ s.t. $\delta_{j \to i}^{(t)} = 1$ **do**

10        Receive $p_{ji}(\delta_j^{(t)}) \boldsymbol{\varphi}_j \left( \boldsymbol{y}_j^{(t)}; \boldsymbol{\lambda}_j, \zeta_j^{(t)} \right)$ from the $j$-th node

11        $\boldsymbol{y}_i^{(t+1)} \mathrel{+}= p_{ji}(\delta_j^{(t)}) \boldsymbol{\varphi}_j \left( \boldsymbol{y}_j^{(t)}; \boldsymbol{\lambda}_j, \zeta_j^{(t)} \right)$

12        $\bar{\delta}_{j \to i} \mathrel{+}= 1$

13      $\boldsymbol{y}_i^{(t+1)} \mathrel{+}= \boldsymbol{\psi}_i \left( \boldsymbol{y}_i^{(t)}; \boldsymbol{\lambda}_i, \zeta_i^{(t)} \right)$

                                     Eq. (2)

  // Normalize empirical estimates

14   $\bar{p}_{ij} \leftarrow \frac{1}{T} \bar{p}_{ij}, \ \forall j \in [n]$

15   $\bar{\delta}_{j \to i} \leftarrow \frac{1}{T} \bar{\delta}_{j \to i}, \ \forall j \in [n]$

16   **return** $\boldsymbol{y}_i^{(T)}, \left\{ \left( \bar{p}_{ij}, \bar{\delta}_{j \to i} \right) \right\}_{j \in [n]}$

---

---

**Algorithm 3:** (**Maybe impossible**) Exact recurrent backpropagation ran by the $i$-th client

**Input:** $\boldsymbol{y}_i^*, \boldsymbol{\lambda}_i$

    `// Compute` $i$`-th block of` $\partial_{\boldsymbol{y}}\bar{F}(\boldsymbol{x}^*, \boldsymbol{\lambda})$ `denoted by` $\langle \boldsymbol{c^y}\rangle_i$

1  $\boldsymbol{u}_i^{(0)} \leftarrow \frac{1}{n}\partial_{\boldsymbol{y}_i}F_i(\boldsymbol{x}_i^*, \boldsymbol{\lambda}_i)$

    `// Compute` $i$`-th block of` $\partial_{\boldsymbol{\lambda}}\bar{F}(\boldsymbol{x}^*, \boldsymbol{\lambda})$ `denoted by` $\langle \boldsymbol{c^\lambda}\rangle_i$

2  $\boldsymbol{v}_i^{(0)} \leftarrow \frac{1}{n}\partial_{\boldsymbol{\lambda}_i}F_i(\boldsymbol{x}_i^*, \boldsymbol{\lambda}_i)$

3  **foreach** $m = 0, \ldots, M-1$ **do**

        `// Sample a minibatch and communication edges`

4       Sample $\zeta_i^{(2m)}$ and $\delta_i^{(2m)} = [\delta_{i \to 1}^{(2m)} \;\; \cdots \;\; \delta_{i \to n}^{(2m)}]^\top$

5       **foreach** $j$ s.t. $\delta_{i \to j}^{(2m)} = 1$ **do**

6          Send $\boldsymbol{u}_i^{(m)}$ to the $j$-th node

7       $\boldsymbol{v}_i^{(m+1)} \leftarrow \boldsymbol{0}_{d_\lambda}$

8       **foreach** $j$ s.t. $\delta_{i \to j}^{(2m)} = 1$ **do**

9          Receive $\boldsymbol{u}_j^{(m)}$ from the $j$-th node

10         $\boldsymbol{v}_i^{(m+1)} \mathrel{+}= p_{ij}(\delta_i^{(2m)})\partial_{\boldsymbol{\lambda}_i}\varphi_i\left(\boldsymbol{y}_i^*; \boldsymbol{\lambda}_i, \zeta_i^{(2m)}\right)\boldsymbol{u}_j^{(m)}$

11      $\boldsymbol{v}_i^{(m+1)} \mathrel{+}= \partial_{\boldsymbol{\lambda}_i}\psi_i\left(\boldsymbol{y}_i^*; \boldsymbol{\lambda}_i, \zeta_i^{(2m)}\right)\boldsymbol{u}_i^{(m)} + \boldsymbol{v}_i^{(m)}$

12      Sample $\zeta_i^{(2m+1)}$ and $\delta_i^{(2m+1)} = [\delta_{i \to 1}^{(2m+1)} \;\; \cdots \;\; \delta_{i \to n}^{(2m+1)}]^\top$

13      **foreach** $j$ s.t. $\delta_{i \to j}^{(2m+1)} = 1$ **do**

14         Send $\boldsymbol{u}_i^{(m)}$ to the $j$-th node

15      $\boldsymbol{u}_i^{(m+1)} \leftarrow \boldsymbol{0}_{d_y}$

16      **foreach** $j$ s.t. $\delta_{i \to j}^{(2m+1)} = 1$ **do**

17         Receive $\boldsymbol{u}_j^{(m)}$ from the $j$-th node

18         $\boldsymbol{u}_i^{(m+1)} \mathrel{+}= p_{ij}(\delta_i^{(2m+1)})\partial_{\boldsymbol{y}_i}\varphi_i\left(\boldsymbol{y}_i^*; \boldsymbol{\lambda}_i, \zeta_i^{(2m+1)}\right)\boldsymbol{u}_j^{(m)}$

19      $\boldsymbol{u}_i^{(m+1)} \mathrel{+}= \partial_{\boldsymbol{y}_i}\psi_i\left(\boldsymbol{y}_i^*; \boldsymbol{\lambda}_i, \zeta_i^{(2m+1)}\right)\boldsymbol{u}_i^{(m)}$

20  **return** $\boldsymbol{v}_i^{(m)}$

21

Iteration of $\boldsymbol{v}_i^{(m+1)}$ in Eq. (11)

Iteration of $\boldsymbol{u}_i^{(m+1)}$ in Eq. (11)

---

**Algorithm 4:** HGP ran by the $i$-th client

---

**Input:** $\boldsymbol{y}_i^*, \boldsymbol{\lambda}_i, \left\{ \left( \bar{p}_{ij}, \bar{\delta}_{j \to i} \right) \right\}_{j \in [n]}$

    `// Compute `$i$`-th block of `$\partial_{\boldsymbol{y}} \bar{F}\left(\boldsymbol{x}^*, \boldsymbol{\lambda}\right)$` denoted by `$\langle \boldsymbol{c}^{\boldsymbol{y}} \rangle_i$

**1**   $\boldsymbol{u}_i^{(0)} \leftarrow \frac{1}{n} \partial_{\boldsymbol{y}_i} F_i\left(\boldsymbol{x}_i^*, \boldsymbol{\lambda}_i\right)$

    `// Compute `$i$`-th block of `$\partial_{\boldsymbol{\lambda}} \bar{F}\left(\boldsymbol{x}^*, \boldsymbol{\lambda}\right)$` denoted by `$\langle \boldsymbol{c}^{\boldsymbol{\lambda}} \rangle_i$

**2**   $\boldsymbol{v}_i^{(0)} \leftarrow \frac{1}{n} \partial_{\boldsymbol{\lambda}_i} F_i\left(\boldsymbol{x}_i^*, \boldsymbol{\lambda}_i\right)$

**3**   **foreach** $m = 0, \ldots, M-1$ **do**

        `// Sample a minibatch and communication edges`

**4**       Sample $\zeta_i^{(2m)}$ and $\delta_i^{(2m)} = [\delta_{i \to 1}^{(2m)} \;\; \cdots \;\; \delta_{i \to n}^{(2m)}]^\top$

**5**       **foreach** $j$ s.t. $\delta_{i \to j}^{(2m)} = 1$ **do**

**6**          Send $\boldsymbol{u}_i^{(m)}$ to the $j$-th node

**7**       $\boldsymbol{v}_i^{(m+1)} \leftarrow \boldsymbol{0}_{d_{\boldsymbol{\lambda}}}$

**8**       **foreach** $j$ s.t. $\delta_{j \to i}^{(2m)} = 1$ **do**

**9**          Receive $\boldsymbol{u}_j^{(m)}$ from the $j$-th node

**10**          $\boldsymbol{v}_i^{(m+1)} \mathrel{+}= \frac{\bar{p}_{ij}}{\bar{\delta}_{j \to i}} \partial_{\boldsymbol{\lambda}_i} \boldsymbol{\varphi}_i\left(\boldsymbol{y}_i^*; \boldsymbol{\lambda}_i, \zeta_i^{(2m)}\right) \boldsymbol{u}_j^{(m)}$

**11**       $\boldsymbol{v}_i^{(m+1)} \mathrel{+}= \partial_{\boldsymbol{\lambda}_i} \boldsymbol{\psi}_i\left(\boldsymbol{y}_i^*; \boldsymbol{\lambda}_i, \zeta_i^{(2m)}\right) \boldsymbol{u}_i^{(m)} + \boldsymbol{v}_i^{(m)}$

               $\left. \begin{array}{c} \\ \\ \\ \\ \end{array} \right\}$ Iteration of $\boldsymbol{v}_i^{(m+1)}$ in Eq. (12)

**12**       Sample $\zeta_i^{(2m+1)}$ and $\delta_i^{(2m+1)} = [\delta_{i \to 1}^{(2m+1)} \;\; \cdots \;\; \delta_{i \to n}^{(2m+1)}]^\top$

**13**       **foreach** $j$ s.t. $\delta_{i \to j}^{(2m+1)} = 1$ **do**

**14**          Send $\boldsymbol{u}_i^{(m)}$ to the $j$-th node

**15**       $\boldsymbol{u}_i^{(m+1)} \leftarrow \boldsymbol{0}_{d_{\boldsymbol{y}}}$

**16**       **foreach** $j$ s.t. $\delta_{j \to i}^{(2m+1)} = 1$ **do**

**17**          Receive $\boldsymbol{u}_j^{(m)}$ from the $j$-th node

**18**          $\boldsymbol{u}_i^{(m+1)} \mathrel{+}= \frac{\bar{p}_{ij}}{\bar{\delta}_{j \to i}} \partial_{\boldsymbol{y}_i} \boldsymbol{\varphi}_i\left(\boldsymbol{y}_i^*; \boldsymbol{\lambda}_i, \zeta_i^{(2m+1)}\right) \boldsymbol{u}_j^{(m)}$

**19**       $\boldsymbol{u}_i^{(m+1)} \mathrel{+}= \partial_{\boldsymbol{y}_i} \boldsymbol{\psi}_i\left(\boldsymbol{y}_i^*; \boldsymbol{\lambda}_i, \zeta_i^{(2m+1)}\right) \boldsymbol{u}_i^{(m)}$

               $\left. \begin{array}{c} \\ \\ \\ \\ \end{array} \right\}$ Iteration of $\boldsymbol{u}_i^{(m+1)}$ in Eq. (12)

**20**   **return** $\boldsymbol{v}_i^{(m)}$

**21**

---

---

**Algorithm 5:** VR-HGP ran by the $i$-th client

**Input:** $\boldsymbol{y}_i^*, \boldsymbol{\lambda}_i, \left\{ \left( \bar{p}_{ij}, \bar{\bar{\delta}}_{j \to i} \right) \right\}_{j \in [n]}, \alpha, \beta$

// Compute $i$-th block of $\partial_{\boldsymbol{y}} \bar{F} \left( \boldsymbol{x}^*, \boldsymbol{\lambda} \right)$ denoted by $\langle \boldsymbol{c^y} \rangle_i$

1   $\boldsymbol{u}_i^{(0)} \leftarrow \frac{1}{n} \partial_{\boldsymbol{y}_i} F_i \left( \boldsymbol{x}_i^*, \boldsymbol{\lambda}_i \right)$

// Compute $i$-th block of $\partial_{\boldsymbol{\lambda}} \bar{F} \left( \boldsymbol{x}^*, \boldsymbol{\lambda} \right)$ denoted by $\langle \boldsymbol{c^\lambda} \rangle_i$

2   $\boldsymbol{v}_i^{(0)} \leftarrow \frac{1}{n} \partial_{\boldsymbol{\lambda}_i} F_i \left( \boldsymbol{x}_i^*, \boldsymbol{\lambda}_i \right)$

3   $\boldsymbol{c}_i^{\boldsymbol{y}} \leftarrow \boldsymbol{u}_i^{(0)}$

4   $\boldsymbol{c}_i^{\boldsymbol{\lambda}} \leftarrow \boldsymbol{v}_i^{(0)}$

5   **foreach** $m = 0, \ldots, M - 1$ **do**

     // Sample a minibatch and communication edges

6      Sample $\zeta_i^{(2m)}$ and $\delta_i^{(2m)} = [\delta_{i \to 1}^{(2m)} \quad \cdots \quad \delta_{i \to n}^{(2m)}]^\top$

7      **foreach** $j$ s.t. $\delta_{i \to j}^{(2m)} = 1$ **do**

8         $\lfloor$ Send $\boldsymbol{u}_i^{(m)}, \boldsymbol{w}_i^{(m)}$ to the $j$-th node

9      $\boldsymbol{v}_i^{(m+1)} \leftarrow \boldsymbol{0}_{d_{\boldsymbol{\lambda}}}$

10     **foreach** $j$ s.t. $\delta_{j \to i}^{(2m)} = 1$ **do**

11       Receive $\boldsymbol{u}_j^{(m)}, \boldsymbol{w}_j^{(m)}$ from the $j$-th node

12       $\boldsymbol{v}_i^{(m+1)} \mathrel{+}= \alpha \left( \frac{\bar{p}_{ij}}{\bar{\delta}_{j \to i}} \partial_{\boldsymbol{\lambda}_i} \boldsymbol{\varphi}_i \left( \boldsymbol{y}_i^*; \boldsymbol{\lambda}_i, \zeta_i^{(2m)} \right) \boldsymbol{u}_j^{(m)} \right)$
$$+ (1 - \alpha) \left( \frac{\bar{p}_{ij}}{\bar{\delta}_{j \to i}} \partial_{\boldsymbol{\lambda}_i} \boldsymbol{\varphi}_i \left( \boldsymbol{y}_i^*; \boldsymbol{\lambda}_i, \zeta_i^{(2m)} \right) \boldsymbol{w}_j^{(m)} \right)$$

13     $\boldsymbol{v}_i^{(m+1)} \mathrel{+}= \alpha \left( \partial_{\boldsymbol{\lambda}_i} \boldsymbol{\psi}_i \left( \boldsymbol{y}_i^*; \boldsymbol{\lambda}_i, \zeta_i^{(2m)} \right) \boldsymbol{u}_i^{(m)} + \boldsymbol{v}_i^{(m)} \right)$
$$+ (1 - \alpha) \left( \partial_{\boldsymbol{\lambda}_i} \boldsymbol{\psi}_i \left( \boldsymbol{y}_i^*; \boldsymbol{\lambda}_i, \zeta_i^{(2m)} \right) \boldsymbol{w}_i^{(m)} \right)$$

14     Sample $\zeta_i^{(2m+1)}$ and $\delta_i^{(2m+1)} = [\delta_{i \to 1}^{(2m+1)} \quad \cdots \quad \delta_{i \to n}^{(2m+1)}]^\top$

15     **foreach** $j$ s.t. $\delta_{i \to j}^{(2m+1)} = 1$ **do**

16       $\lfloor$ Send $\boldsymbol{u}_i^{(m)}, \boldsymbol{w}_i^{(m)}$ to the $j$-th node

17     $\boldsymbol{u}_i^{(m+1)} \leftarrow \boldsymbol{0}_{d_{\boldsymbol{y}}}$

18     $\boldsymbol{w}_i^{(m+1)} \leftarrow \boldsymbol{0}_{d_{\boldsymbol{y}}}$

19     **foreach** $j$ s.t. $\delta_{j \to i}^{(2m+1)} = 1$ **do**

20       Receive $\boldsymbol{u}_j^{(m)}, \boldsymbol{w}_j^{(m)}$ from the $j$-th node

21       $\boldsymbol{u}_i^{(m+1)} \mathrel{+}= \frac{\bar{p}_{ij}}{\bar{\delta}_{j \to i}} \partial_{\boldsymbol{y}_i} \boldsymbol{\varphi}_i \left( \boldsymbol{y}_i^*; \boldsymbol{\lambda}_i, \zeta_i^{(2m+1)} \right) \boldsymbol{u}_j^{(m)}$

22       $\boldsymbol{w}_i^{(m+1)} \mathrel{+}= \beta \left( \frac{\bar{p}_{ij}}{\bar{\delta}_{j \to i}} \partial_{\boldsymbol{y}_i} \boldsymbol{\varphi}_i \left( \boldsymbol{y}_i^*; \boldsymbol{\lambda}_i, \zeta_i^{(2m+1)} \right) \boldsymbol{u}_j^{(m)} \right)$

23     $\boldsymbol{u}_i^{(m+1)} \mathrel{+}= \partial_{\boldsymbol{y}_i} \boldsymbol{\psi}_i \left( \boldsymbol{y}_i^*; \boldsymbol{\lambda}_i, \zeta_i^{(2m+1)} \right) \boldsymbol{u}_i^{(m)}$

24     $\boldsymbol{w}_i^{(m+1)} \mathrel{+}= \beta \left( \partial_{\boldsymbol{y}_i} \boldsymbol{\psi}_i \left( \boldsymbol{y}_i^*; \boldsymbol{\lambda}_i, \zeta_i^{(2m+1)} \right) \boldsymbol{w}_i^{(m)} + \boldsymbol{c}_i^{\boldsymbol{y}} \right) + (1 - \beta) \left( \boldsymbol{w}_i^{(m)} + \boldsymbol{u}_i^{(m+1)} \right)$

25   **return** $\boldsymbol{v}_i^{(m)} + \boldsymbol{c}_i^{\boldsymbol{\lambda}}$

---

Table 4: Test accuracy of personalized models on the simulated stochastic directed communication network (`StoD`) (average clients / 10% percentile).

| | Method | Dataset | | | |
|---|---|---|---|---|---|
| | | CIFAR10 | CIFAR100 | Shakespeare | EMNIST |
| Global | SGP | 75.6 / 69.7 | 38.8 / 31.4 | 28.5 / 26.0 | 79.7 / 72.5 |
| | SGP-MTL | 73.0 / 64.2 | 34.4 / 27.7 | 27.7 / 24.0 | 80.9 / 73.2 |
| Personalized | Local | 62.5 / 41.1 | 33.1 / 25.1 | 25.3 / 17.5 | 73.7 / 63.8 |
| | Local-MTL | 64.2 / 38.3 | 35.2 / 28.0 | 25.5 / 20.0 | 67.3 / 57.5 |
| | **PDBO-DA** | 75.6 / 70.3 | 42.1 / **37.1** | **38.2** / 33.7 | 80.8 / 72.9 |
| | **PDBO-MTL** | 73.6 / 65.2 | 41.4 / 35.1 | 38.0 / **34.9** | 81.6 / **75.0** |
| | **PDBO-MTL&DA** | **77.2 / 73.0** | **43.3** / 36.7 | 37.7 / 33.0 | **82.2** / 74.5 |

## G   ADDITIONAL EXPERIMENTS

We conducted personalization benchmarks on different tasks: image classification (CIFAR10 and CIFAR100 (Krizhevsky, 2009)), language modeling (Shakespeare (Caldas et al., 2018; McMahan et al., 2017)), and handwritten character recognition (EMNIST (Cohen et al., 2017)) on a simulated stochastic directed communication network.

### G.1   SETTINGS

We ran our approaches, PDBO-DA, PDBO-MTL, and PDBO-MTL&DA, and Local and SGP for baselines which are explained in Section 6. We adopted the stochastic directed network `StoD` for a simulated communication network with the same setting as in Section 6. For each approach, we solve different tasks on corresponding datasets: CIFAR10, CIFAR100, Shakespeare, and EMNIST.

**Tasks**   For image classification on CIFAR10, we distributed samples with the same labels across clients according to a symmetric Dirichlet distribution with parameter 0.4, as in Marfoq et al. (2021); Wang et al. (2019), to create a federated version. We used 40% of the total data as the train and validation dataset in a 3:1 ratio and the rest as the test dataset. We also tested image classification using CIFAR100 exploiting the availability of "coarse" and "fine" labels, using a two-stage Pachinko allocation method (Li & McCallum, 2006) as in Reddi et al. (2020); Marfoq et al. (2021), to distribute 900, 300, and 1800 sized train, validation, test datasets to each client, respectively. Pachinko allocation ran with the parameters adopted in Marfoq et al. (2021). For both CIFAR10 and CIFAR100, we set $n = 20$ and trained MobileNet-v2 (Sandler et al., 2018), implemented in TorchVision(Marcel & Rodriguez, 2010), with an additional linear layer.

The Shakespeare dataset was naturally divided by assigning all lines from the same character to the same client as in Marfoq et al. (2021); McMahan et al. (2017). From 728 characters, we randomly selected $n = 20$ characters and assigned each of them to a client. We trained two stacked-LSTM layers with 256 hidden units followed by a densely-connected layer, to predict the next character from a sequence of 200 English characters as input. The model embeds 80 characters into a learnable 8-dimensional embedding space. For each client, we used 80% of lines as the train and validation dataset in a 3:1 ratio and the rest as the test dataset. The lines are split from the beginning in the order train, validation, and test to simulate the practical time dependence between datasets.

The settings of handwritten character recognition on EMNIST (Cohen et al., 2017) are described in Section 6 and Appendix D.

**Approaches**   For PDBO-DA, PDBO-MTL, and PDBO-MTL&DA, we adopted the same strategies and parameters in Section 6. PDBO-DA optimizes weights vector of loss which elements correspond to labels (characters) of EMNIST, CIFAR10, and CIFAR100 (Shakespeare). PDBO-MTL optimizes ensemble weights of predictions of 3 models of each task, and PDBO-MTL&DA simultaneously optimizes the outer-parameters of PDBO-DA and PDBO-MTL. Except for reducing $M$ to 20 rounds for efficiency, we adopted the same VR-HGP setting in Section 6.

For baselines, due to the absence of personalization methods applicable to stochastic directed communication networks, Local and SGP were adopted. We also trained an ensemble model with uniform prediction weights for both baselines, Local-MTL and SGP-MTL to fairly compare the performance difference between the baselines and our approaches. This allows us to exclude architectural differences from the reasons for performance improvements.

**Results and discussions**  Table 4 shows the average test accuracy with weights proportional to local test dataset sizes. We observed that our approaches PDBO-DA, PDBO-MTL, and PDBO-MTL&DA improved accuracy from baselines on all tasks with a few exceptions: PDBO-MTL on CIFAR10 and PDBO-DA on EMNIST. PDBO-MTL&DA out performed on CIFAR10, CIFAR100, and EMNIST in average accuracy, confirming the simultaneous optimization of different parameters is effective on complex tasks. Even on the next character prediction task on time series data, all of our approaches improved the performance from baselines, indicating our gradient-based PDBO is effective in a variety of tasks. Note that the performance improvements did not come from the architectural differences in the models (single model or ensemble model) since PDBO-MTL and PDBO-MTL&DA outperformed SGP-MTL and Local-MTL. Accuracy at the 10% percentile are also improved from the baselines in all the tasks and our approaches, which validated that clients fairly benefited from our personalization.

## H  Hyper-gradient Estimation by Iterative Differentiation

Iterative differentiation is categorized into forward (ITD-Forward) and backward (ITD-Backward) (Franceschi et al., 2017). ITD-Forward and ITD-Backward are advantageous to AID as they do not assume convexity on the loss. In this section, we show that ITD-Forward and ITD-Backward suffer from physical limitations and large complexity in communications.

ITD-Forward and ITD-Backward compute the hyper-gradient by recursively tracing back all inner-loops, which is performed to obtain the trained parameter, $\boldsymbol{y}^{(T)}$, without requiring $\boldsymbol{y}^{(T)}$ to be the stationary point. To apply ITD-Forward and ITD-Backward to our setting, we suppose $\boldsymbol{y}^{(T)}$ is obtained by $T$ iterations of Eq. (2). Considering iterations for $t = 0, \ldots, T-1$, concatenated hyper-gradient of ITD-Forward and ITD-Backward can be given as

$$\mathrm{d}_{\boldsymbol{\lambda}}^{\mathrm{ITD}} \bar{F}(\boldsymbol{x}^{(T)}, \boldsymbol{\lambda}) := \sum_{t=0}^{T-1} \hat{\boldsymbol{B}}^{(t)} \prod_{s=t+1}^{T-1} \hat{\boldsymbol{A}}^{(s)} \boldsymbol{c}^{\boldsymbol{x}} + \boldsymbol{c}^{\boldsymbol{\lambda}}. \tag{41}$$

ITD-Forward and ITD-Backward are differentiated by how they compute Eq. (41).

**Forward mode iterative differentiation (ITD-Forward)**  Let $\boldsymbol{U}^{(t)} := \sum_{t'=0}^{t-1} \hat{\boldsymbol{B}}^{(t')} \prod_{s=t'+1}^{t-1} \hat{\boldsymbol{A}}^{(s)}$. To compute Eq. (41), ITD-Forward updates the matrix $\boldsymbol{U}^{(t)} \in \mathbb{R}^{nd_{\boldsymbol{\lambda}} \times nd_{\boldsymbol{y}}}$. After initializing by $\boldsymbol{U}^{(0)} = \boldsymbol{O}_{nd_{\boldsymbol{\lambda}} \times nd_{\boldsymbol{y}}}$, the following iterations for $t = 0, \ldots, T-1$,

$$\boldsymbol{U}^{(t+1)} = \boldsymbol{U}^{(t)} \hat{\boldsymbol{A}}^{(t)} + \hat{\boldsymbol{B}}^{(t)} \tag{42}$$

provides the hyper-gradient by $\mathrm{d}_{\boldsymbol{\lambda}}^{\mathrm{ITD}} \bar{F}(\boldsymbol{x}^{(T)}, \boldsymbol{\lambda}) = \boldsymbol{U}^{(K)} \boldsymbol{c}^{\boldsymbol{x}} + \boldsymbol{c}^{\boldsymbol{\lambda}}$.

We then consider a decentralized algorithm ran by the $i$-th client to obtain the $i$-th block of concatenated hyper-gradient $\mathrm{d}_{\boldsymbol{\lambda}_i}^{\mathrm{ITD}} \bar{F}(\boldsymbol{x}^{(T)}, \boldsymbol{\lambda}) := \langle \mathrm{d}_{\boldsymbol{\lambda}}^{\mathrm{ITD}} \bar{F}(\boldsymbol{x}^{(T)}, \boldsymbol{\lambda}) \rangle_i \in \mathbb{R}^{d_{\boldsymbol{\lambda}}}$. By letting the $i$-th client update column block matrices $\boldsymbol{U}_{ki}^{(t)} := \langle \boldsymbol{U}^{(t)} \rangle_{ki}$ for all $k \in [n]$, a decentralized algorithm of ITD-Forward can be written as follows.

---

**Forward mode iterative differentiation (ITD-Forward)**

$\boldsymbol{U}_{ki}^{(0)} \leftarrow \boldsymbol{O}_{d_{\boldsymbol{\lambda}} \times d_{\boldsymbol{y}}}, \forall k \in [n]$

$$\boldsymbol{U}_{ki}^{(t+1)} = \sum_{j \in [n]} \delta_{j \to i}^{(t)} \boldsymbol{U}_{kj}^{(t)} \left\langle \hat{\boldsymbol{A}}^{(t)} \right\rangle_{ji} + \delta_{k \to i}^{(t)} \left\langle \hat{\boldsymbol{B}}^{(t)} \right\rangle_{ki}, \forall k \in [n] \quad (43)$$

for $t = 0, 1, 2, \ldots, T-1$

$$\mathrm{d}_{\boldsymbol{\lambda}_i}^{\mathrm{ITD}} \bar{F}(\boldsymbol{x}^{(T)}, \boldsymbol{\lambda}) \leftarrow \sum_{j \in [n]} \boldsymbol{U}_{ij}^{(T)} \boldsymbol{c}_j^{\boldsymbol{x}} + \boldsymbol{c}_i^{\boldsymbol{\lambda}} \tag{44}$$

---

The $i$-th node can compute the first term of the right hand of Eq. (43) because Jacobian-matrix product $\boldsymbol{U}_{kj}^{(t)}\langle\hat{\boldsymbol{A}}^{(t)}\rangle_{ji}$ is always receivable from the $j$-th client even when edges are directed; when $\boldsymbol{U}_{kj}^{(t)}\langle\hat{\boldsymbol{A}}^{(t)}\rangle_{ji} \neq \boldsymbol{O}_{d_{\boldsymbol{\lambda}}\times d_{\boldsymbol{y}}}$ the communication edge for receiving exists ($\delta_{j\to i}^{(t)}=1$) and $\boldsymbol{U}_{kj}^{(t)}\langle\hat{\boldsymbol{A}}^{(t)}\rangle_{ji}=\boldsymbol{O}_{d_{\boldsymbol{\lambda}}\times d_{\boldsymbol{y}}}$ otherwise. In the same manner, the second term can be computed receiving $\langle\hat{\boldsymbol{B}}^{(t)}\rangle_{ki}$ from the $k$-th node. Note that when we choose to let the $i$-th client update row block matrices $\boldsymbol{U}_{ik}^{(t)}:=\langle\boldsymbol{U}^{(t)}\rangle_{ik}$ for all $k\in[n]$ required to update $\boldsymbol{U}_{ik}^{(t)}$ are not guaranteed on stochastic communication networks.

The difficulty of the forward mode is its communication cost. To update $\boldsymbol{U}_{ki}^{(t)}$, the any $j$-th client needs to send $\boldsymbol{U}_{kj}^{(t)}\langle\hat{\boldsymbol{A}}^{(t)}\rangle_{ji}\in\mathbb{R}^{d_{\boldsymbol{\lambda}}\times d_{\boldsymbol{y}}}$ to the $i$-th client. Because both $d_{\boldsymbol{\lambda}}$ and $d_{\boldsymbol{y}}$ can be large for practical models such as deep neural networks, communicating $O(d_{\boldsymbol{\lambda}}\times d_{\boldsymbol{y}})$ parameters is prohibitive in general.

The other problem is the last step of computing the sum $\sum_{j\in[n]}\boldsymbol{U}_{ij}^{(T)}\boldsymbol{c}_j^{\boldsymbol{x}}$ in Eq. (44). This requires communicating with all the clients. The $i$-th client, therefore, needs to wait for several communication rounds until it can receive the Jacobian-vector product $\boldsymbol{U}_{ij}^{(T)}\boldsymbol{c}_j^{\boldsymbol{x}}\in\mathbb{R}^{d_{\boldsymbol{\lambda}}}$ from all $j\in[n]$.

**Backward mode iterative differentiation (ITD-Backward)**   ITD-Backward computes Eq. (41) in the reverse time sequence of ITD-Forward, i.e., $t=T-1,\ldots,0$, resulting in having an iteration similar to Eq. (10). Let $\boldsymbol{u}^{(t)}=\prod_{s=t}^{T-1}\hat{\boldsymbol{A}}^{(s)}\boldsymbol{c}^{\boldsymbol{y}}$ and $\boldsymbol{v}^{(t)}=\sum_{s=t+1}^{T-1}\hat{\boldsymbol{B}}^{(s)}\boldsymbol{u}^{(s)}+\boldsymbol{c}^{\boldsymbol{\lambda}}$. By initializing $\boldsymbol{u}^{(T)}\leftarrow\boldsymbol{c}^{\boldsymbol{y}}$ and $\boldsymbol{v}^{(T)}\leftarrow\boldsymbol{c}^{\boldsymbol{\lambda}}$, and by the following iterations for $t=T-1,\ldots,0$,

$$\begin{cases}\boldsymbol{v}^{(t)}\leftarrow\hat{\boldsymbol{B}}^{(t+1)}\boldsymbol{u}^{(t+1)}+\boldsymbol{v}^{(t+1)},\\\boldsymbol{u}^{(t)}\leftarrow\hat{\boldsymbol{A}}^{(t+1)}\boldsymbol{u}^{(t+1)},\end{cases}$$

we obtain the hyper-gradient estimate as $\widehat{\mathrm{d}_{\boldsymbol{\lambda}}\bar{F}}\leftarrow\boldsymbol{v}^{(0)}$.

Mathematically, the decentralized algorithm of ITD-Backward for the $i$-th client can be written as

---
**Backward mode iterative differentiation (ITD-Backward)**

$$\boldsymbol{u}_i^{(T)}\leftarrow\langle\boldsymbol{c}^{\boldsymbol{y}}\rangle_i,\quad\boldsymbol{v}_i^{(T)}\leftarrow\langle\boldsymbol{c}^{\boldsymbol{\lambda}}\rangle_i$$
$$\begin{cases}\boldsymbol{v}_i^{(t)}\leftarrow\sum_{j=1}^n\delta_{i\to j}^{(t+1)}\left\langle\hat{\boldsymbol{B}}^{(t+1)}\right\rangle_{ij}\boldsymbol{u}_j^{(t+1)}+\boldsymbol{v}_i^{(t+1)},\\\boldsymbol{u}_i^{(t)}\leftarrow\sum_{j=1}^n\delta_{i\to j}^{(t+1)}\left\langle\hat{\boldsymbol{A}}^{(t+1)}\right\rangle_{ij}\boldsymbol{u}_j^{(t+1)}\end{cases}$$
$$\text{for }t=T-1,T-2,\ldots,0$$
$$\mathrm{d}_{\boldsymbol{\lambda}_i}^{\mathrm{ITD}}\bar{F}(\boldsymbol{x}^{(T)},\boldsymbol{\lambda})\leftarrow\boldsymbol{v}_i^{(0)}.$$

---

In centralized bilevel optimization, iterations of $t=T-1,\ldots,0$ are realized by storing the intermediate parameters and indices of every minibatch during the training (Franceschi et al., 2017) to recover all iterations after $T$ inner-gradient descent steps. However, when we use SGP iterations for inner-steps, the stochastic network does not guarantee to reproduce the communication edges which clients experienced during the training, making the computation of the decentralized ITD-Backward infeasible.

Moreover, ITD-Backward also requires undirected edges similar to the exact recurrent backpropagation as pointed out in Section 4.2 (Exact backpropagation requires undirected edges).

We thus conclude that ITD-Backward requires the communication network to be restricted to static and undirected in order to function.

