# OpenReview forum: "Personalized Decentralized Bilevel Optimization over Stochastic and Directed Networks"
_ICLR.cc/2023/Conference — Submitted to ICLR 2023_

### Official Review · Reviewer_TFeg · 2022-10-23

**Confidence:** 4
**Correctness:** 3
**Technical Novelty And Significance:** 2
**Empirical Novelty And Significance:** 2
**Recommendation:** 5

**Clarity, Quality, Novelty And Reproducibility:**

The quality, clarity and originality of the work need to be improved. See Strength And Weaknesses.

**Strength And Weaknesses:**

Pros:
1. This paper considers the general setting in decentralized training, i.e., the network is stochastic and directed.

 2. The formulation of PDBO is more general than previous papers, as discussed in Section E and Table 3 of the appendix.

3.  HGP method does not require computing exact Hessian or Jacobian, which is an improvement comparing to previous CDBO papers.

Cons:

1. Reformulating Eq. (5) as Eq. (6) is reasonable in practice (and is true under Assumption 2). However, in theory technically speaking the condition $y_i^* = y_i^{(T)}$ in Assumption 2 is unusual. It would be good if the reformulation (Eq. (6)) of Eq. (5) can be theoretically justified (e.g., via some error estimation) without assuming $y_i^*=y_i^{(T)}$ in Assumption 2.

2. (Assumption 2) The authors claim "To apply implicit differentiation, we introduce the following assumption as in Grazzi et al. (2020)", but is "$y_i^* = y_i^{(T)}$" assumed in Grazzi et al. (2020)? It seems that in Grazzi et al. (2020) they only require $y^*$ to be unique instead of accessibility in $T$ steps. "$y_i^*=y_i^{(T)}$" seems to be a strong assumption that is not included in papers about non-asymptotic convergence of bilevel optimization. Besides, carefully analyzing the error incurred by $\|y_i^* -y_i^{(T)}\|^2$ is not a trivial problem in bilevel optimization literature. For example, Chen et al. (2022) uses a novel way to deal with this error (see Lemma 3 therein and the order of $T$ in Theorem 1 and Proposition 2) and successfully reduce the sample complexity comparing previous works on bilevel optimization.

3. (Theorem 1) This theorem characterizes the hypergradient estimation error. How can we apply this in real PDBO algorithms to solve (5)? Should we just use $v^{(\lambda)} + c^{\lambda}$ as the hypergradient and then use single-agent gradient descent, or should we still run some decentralized algorithms to solve (5)? It is unclear how we should use the estimation (i.e., $v^{(\lambda)} + c^{\lambda}$) mentioned in Theorem 1 to run PDBO. It would be better if there is another theorem characterizing the convergence results (e.g., sample complexity) of PDBO using this estimation.

4. (Mini-batch) We need to use mini-batch method (i.e., setting $b$ in Theorem 1 to be large enough) for HGP. However, some of the previous CDBO papers (e.g., Yang et al. (2022)) do not require mini-batch method.

Questions:

 ("Suppose the i-th client is responsible for computing the i-th block of $u^{(m+1)}$ and $v^{(m+1)}$, ...", Section 4.2.) What happens after all the agents finish the computation of their own blocks? Is there a central node collecting all the blocks to obtain $v^{(M)}$?

References:

[Chen et al., 2022] T. Chen, Y. Sun, and W. Yin. Closing the gap: Tighter analysis of alternating stochastic gradient
methods for bilevel problems. Advances in Neural Information Processing Systems, 34, 2021.\\

[Grazzi et al., 2020] R. Grazzi, L. Franceschi, M. Pontil, and S. Salzo. On the iteration complexity of hypergradient
computation. In International Conference on Machine Learning, pages 3748–3758. PMLR, 2020.\\

[Yang et al., 2022]S. Yang, X. Zhang, and M. Wang. Decentralized gossip-based stochastic bilevel optimization over
communication networks. arXiv preprint arXiv:2206.10870, 2022.

**Summary Of The Paper:**

This paper studies Personalized Decentralized Bilevel Optimization (PDBO). Based on Stochastic Gradient Push (SGP) method, the authors propose Hyper-Gradient Push (HGP) method to estimate the hypergradient and analyze the estimation error of its variance-reduced version (VR-HGP). On top of the hypergradient estimation, they conduct experiments to solve PDBO problems and compare its performance to some existing algorithms.


**Summary Of The Review:**

Overall, I feel that the proposed method has several benefits than previous ones, but there are quite a few limitations and questions left. I tend to reject this time, but am open to increase the evaluation based on the other reviewers' comments and the authors' response.

---

> ### Author Response · Authors · 2022-11-19
> **Response to Reviewer TFeg (1/3)**
>
> ## Updates of the paper
>
> We thank the reviewer for the constructive comments. Motivated by your comments and questions, we improved the paper by,
>
> - Replacing the **assumption of  $\boldsymbol{y}\_i^∗=\boldsymbol{y}\_i^{(T) }$** with assumptions on **loss convexity** and **network connectivity** (Section 3 and 4)
> - Adding theoretical analysis of the **estimation error incurred by $\Vert \boldsymbol{y}\_i^∗ - \boldsymbol{y}\_i^{(T) }\Vert$**  in Appendix C.7
> - Adding **pseudo algorithms** of PDBO, HGP, and VR-HGP in Appendix F
>
> We also added the following parts according to the comments by other reviews:
>
> - **Extended experimental results** **on CIFAR10, CIFAR100,** and **Shakespeare** in Appendix G
> - **Discussion on the applicability of other hyper-gradient estimation methods** on stochastic directed networks in Related work and Appendix H
>
> ---
>
> ## Responses
>
> ### [Question] “*What happens after all the agents finish the computation of their own blocks? Is there a central node collecting all the blocks to obtain $\boldsymbol{v}^{(M) }$?“*
>
> **Each client simply runs its local gradient step, such as SGD, and there is no need for the central client.**
>
> Recall that **the $i$-th client only needs the $i$-th block** of concatenated hyper-gradient $\boldsymbol{v}^{(M) }$ (denoted by $\langle \boldsymbol{v}^{(M)}\rangle \_i \approx \frac{\mathrm{d}\bar{F}(\boldsymbol{x}^*,\boldsymbol{\lambda})}{\mathrm{d}\boldsymbol{\lambda}\_i}$) to update its hyperparameter $\boldsymbol{\lambda}\_i$, as described in the beginning of Sec 4.2. Our specific choice "*Suppose the $i$-th client is responsible for computing the $i$-th block of $\boldsymbol{u}^{(m+1)}$ and $\boldsymbol{v}^{(m+1)}$ , ...”* allows each node to consequently obtain hyper-gradient block they need. Again, as the outer-optimization step of the $i$-th client is independent of other hyper-gradient blocks $\langle \boldsymbol{v}^{(M)}\rangle\_{j,\forall j\neq i}$ , **no central server or additional communications are required**.
>
> We note that this property that a client can obtain the required hyper-gradinet block $\langle \boldsymbol{v}^{(M)}\rangle \_i$ by simple iterations **is tied to our choice of recurrent backpropagation** for the computation of the estimator. We found that the forward-mode propagation, as the conterpart, necessarily requires additional communication rounds to collect blocks of hyper-gradient as shown in Appendix H. We added this note in Related work of the updated paper.
>
> We added the pseudo algorithms of PDBO (our bilevel optimization) and (hyper-gradient estimation) in Appendix F. Since these algorithms are written from the client’s perspective, we hope they help your understanding of the comments above.

---

> > ### Author Response · Authors · 2022-11-19
> > **Response to Reviewer TFeg (2/3)**
> >
> >
> >
> > ### [Con. 1] Theoretical justification of reformulation Eq. (5) as Eq. (6)
> >
> > ### [Con. 2]  $\boldsymbol{y}\_i^∗=\boldsymbol{y}\_i^{(T) }$ seems to be too strong Assumption.
> >
> > > *Reformulating Eq. (5) as Eq. (6) is reasonable in practice (and is true under Assumption 2). However, in theory technically speaking the condition  $\boldsymbol{y}\_i^∗=\boldsymbol{y}\_i^{(T) }$ in Assumption 2 is unusual. … . It would be **good if the reformulation (Eq. (6)) of Eq. (5) can be theoretically justified** (e.g., via some error estimation) without assuming $\boldsymbol y\_i^∗=\boldsymbol y\_i^{(T) }$ in Assumption 2.*
> > >
> >
> > > ***" $y\_i^∗=y\_i^{(T) }$" seems to be a strong assumption** that is not included in papers about non-asymptotic convergence of bilevel optimization. … Besides, carefully **analyzing the error incurred by  $\boldsymbol{y}\_i^∗=\boldsymbol{y}\_i^{(T) }$ is not a trivial** **problem** in bilevel optimization literature.*
> > >
> >
> > Motivated by your comment, we replaced the inner-problem of reformulated PDBO with the stationary condition of  $\boldsymbol{y}\_i^∗$ (new Eq. (7)) following Grazzi et al., 2020. Then we **justify reformulation from equivalency of the optimum and the stationary point of SGP, which is ensured by** **weaker assumptions (convexity & connectivity) than the assumption of  $\boldsymbol{y}\_i^∗=\boldsymbol{y}\_i^{(T) }$.**
> >
> > For **Con. 1,** reformulating PDBO is justified from the mathematical equivalencies between the stationary point of SGP (Eq. (2)) and the optimum of distributed learning in Eq. (1). We ensure this equivalency by introducing assumptions weaker than  $\boldsymbol{y}\_i^∗=\boldsymbol{y}\_i^{(T) }$ : **every client cost $f\_i$ is strongly-convex** (new Assumption 1) and **the stochastic network is strongly connected in expectation** (new Assumption 2). The new Assumptions 1 and 2 are adopted in the original papers of SGP [Nedić, et al. (2014)] and Push-sum [Bénézit, et al (2010)], respectively. We omitted assumption of  $\boldsymbol{y}\_i^∗=\boldsymbol{y}\_i^{(T) }$ from the updated paper.
> >
> > Regarding **Con. 2,** since $\boldsymbol{y}\_i^{*}$ is practically approximated by  $\boldsymbol{y}\_i^{(T)}$, the consequence of $T$ iterations of SGP, we provided an estimation error bound of hyper-gradient including the error  $\Vert \boldsymbol{y}\_i^∗ - \boldsymbol{y}\_i^{(T) }\Vert$ (Theorem 11 in Appendix C.7), which results in more general bound than Theorem 1. Theorem 11 indicates that the error in inner-parameter $\Vert \boldsymbol{y}\_i^∗ - \boldsymbol{y}\_i^{(T) }\Vert$ brings estimation bias which can linearly grow as $\Vert \boldsymbol{y}\_i^∗ - \boldsymbol{y}\_i^{(T) }\Vert$ increases, finding it non-trivial as you pointed.
> >
> > We finally re-organized and improved the paper having the following structure:
> >
> > - Section 2 (Problem formulation) reformulates PDBO by the stationary point of inner-iteration  $\boldsymbol{y}\_i^∗$ and ensures its existence from additional assumptions on convexity and network connection
> > - Section 5 (Experiments) substitutes  $\boldsymbol{y}\_i^∗$ =  $\boldsymbol{y}\_i^{(T)}$ as a practical approximation, and empirically justifies this approximation by confirming improvements in personalization performance.
> > - Appendix C.7 (Relaxation of convergence to  $\boldsymbol{y}\_i^∗$) theoretically bounds the estimation error of the hyper-gradient incurred by $\Vert \boldsymbol{y}\_i^∗ - \boldsymbol{y}\_i^{(T) }\Vert$ caused by the practical approximation in the experiments.

---

> > > ### Author Response · Authors · 2022-11-19
> > > **Response to Reviewer TFeg (3/3)**
> > >
> > >
> > > ### [Con. 3] “*How can we apply this* (Theorem 1) *in real PDBO algorithms to solve (5)?”*
> > >
> > > > ***How can we apply this in real PDBO algorithms to solve (5)**? Should we just use $\boldsymbol{v}(\boldsymbol{λ})+\boldsymbol{c}^λ$  as the hypergradient and then use single-agent gradient descent, or should we still run some decentralized algorithms to solve (5)? …
> > > … **It would be better if there is another theorem characterizing the convergence results** (e.g., sample complexity) of PDBO using this estimation.*
> > > >
> > >
> > > **We can apply Theorem 1 to bound the variance of the gradient of outer SGD which solves (5).**
> > >
> > > The outer-problem in Eq. (5) is **simply solved by a gradient method, such as SGD,** from a mathematical perspective, while its computation is split into decentralized clients. Recalling that $\boldsymbol{v}(λ)+\boldsymbol{c}^λ$  is an unbiased estimate of the true gradient $\mathrm{d}\_{\boldsymbol{λ}}\bar{F}(\boldsymbol{x}^∗,\boldsymbol{λ})$, **the outer-optimization can be interpreted as an SGD, whose noise is bounded by Theorem 1**. This makes it possible for us to obtain the convergence property of PDBO from the convergence property of SGD, which is much easier.
> > >
> > > ### [Con. 4] HGP requires minibatch method, but other CDBO works do not.
> > >
> > > > *(Mini-batch) We need to use mini-batch method (i.e., setting $b$ in Theorem 1 to be large enough) for HGP. However, some of the previous CDBO papers (e.g., Yang et al. (2022)) do not require mini-batch method.*
> > > >
> > >
> > > The presence or absence of mini-batch size $b$ is **simply due to** **differences in how randomness is defined**.
> > >
> > > Yang et al. (2022) assume an oracle that can produce a stochastic gradient with a bounded variance in their Assumption 3.4 (iii), without specifying what the randomness is (e.g., input or minibatch). That is, it would be more appropriate to say that the assumption of Yang et al. (2022) is more general than ours, but it does not mean their approach is free from the use of the minibatch method. Indeed, if one expects to realize such an oracle who produces a stochastic gradient on a single input, one needs to control the minibatch size as we did to control the variance.
> > >
> > > ### Other points to cover all comments
> > >
> > > > *Besides, carefully analyzing the error incurred by $\vert \boldsymbol{y}\_i^∗ - \boldsymbol{y}\_i^{(T) }\vert^2$ is not a trivial problem in bilevel optimization literature. For example, Chen et al. (2022) uses a novel way to deal with this error (see Lemma 3 therein and the order of $T$ in Theorem 1 and Proposition 2) and successfully reduce the sample complexity comparing previous works on bilevel optimization.*
> > > >
> > >
> > > As you remark, considering estimation error with sample complexity is not a trivial problem. In our HGP, sampling complexity is typically crucial because it uniquely involves the sampling of communication edges other than data sampling. We obtained the bound of the estimation bias incurred by a given $\vert \boldsymbol{y}\_i^∗ - \boldsymbol{y}\_i^{(T) }\vert^2$ in Theorem 11. It is exciting to consider combining this bound with an asymptotic convergence property of SGP (Nedić, et al. (2016), Theorem 1) to trade off the small estimation bias and sampling complexity. We include addressing those complexities in our future directions.
> > >
> > > > *The authors claim "To apply implicit differentiation, we introduce the following assumption as in Grazzi et al. (2020)", Grazzi et al. (2020)", but is ”$\vert \boldsymbol{y}\_i^∗ - \boldsymbol{y}\_i^{(T) }\vert^2$ " assumed in Grazzi et al. (2020)? It is unusual to assume   $\boldsymbol{y}\_i^∗=\boldsymbol{y}\_i^{(T) }$ in bilevel optimization literatures.*
> > > >
> > >
> > > We thank your correction. We omitted the citation of Grazzi et al. (2020) from the sentence, as we found it is not one of their assumptions.
> > >
> > > ---
> > >
> > > ### References
> > >
> > > - [Grazzi et al. (2020)] R. Grazzi, L. Franceschi, M. Pontil, and S. Salzo. On the iteration complexity of hypergradient computation. In International Conference on Machine Learning, pages 3748–3758. PMLR, 2020.
> > > - [Nedić, et al. (2014)] Nedić, A., & Olshevsky, A. (2014). Distributed optimization over time-varying directed graphs. IEEE Transactions on Automatic Control, 60(3), 601-615.
> > > - [Bénézit, et al (2010)] Bénézit, Florence, et al. "Weighted gossip: Distributed averaging using non-doubly stochastic matrices." 2010 ieee international symposium on information theory. IEEE, 2010.
> > > - [Nedić, et al. (2016)] Nedić, Angelia, and Alex Olshevsky. "Stochastic gradient-push for strongly convex functions on time-varying directed graphs." IEEE Transactions on Automatic Control 61.12 (2016): 3936-3947.
> > > - [Yang et al., 2022] S. Yang, X. Zhang, and M. Wang. Decentralized gossip-based stochastic bilevel optimization over communication networks. arXiv preprint arXiv:2206.10870, 2022.
> > > ---
> > > Update 11/28: Fixed a compilation error of tex in the response to Con. 3.

---

### Official Review · Reviewer_3KYv · 2022-10-24

**Confidence:** 2
**Correctness:** 2
**Technical Novelty And Significance:** 2
**Empirical Novelty And Significance:** 3
**Recommendation:** 3

**Clarity, Quality, Novelty And Reproducibility:**

The paper is very hard to read. Because of its low clarity, I cannot judge the quality and the novelty of the results.
The code for reproducing the results was not submitted and in the empirical section the chosen model architecture is not described.

**Strength And Weaknesses:**

The paper is very hard to read, and hence it is difficult to assess the contributions and correctness of the derivations.

These are some of the reasons that make the paper difficult to parse:
1. Some notation and formulations are inexact, confusing or undefined:
- In Problem 1 and Problem 5, the index $i$ is used for two different thing: first for indicating the client's parameters to be optimized and then for indexing the terms of the sum. Also the size of $\xi_i$ is not provided and $p(\xi_i)$ is not formally defined.
- Right before Equation 2, it is said that "The i-th client in SGP updates its weight $\omega_i$" but Equation 4 do not update this parameter.
- Equation 4, the operation $[a \quad b]$ is not defined. I assumed it was a vector concatenation. Also $p()$ denotes here the message distribution, while it was used before for the local data distribution.
- (minor) In page 3, end of Section 2, $p_{ij}$ should be replaced by $p()$ ($p_{ij}$ cannot be a doubly stochastic matrix as it is defined as a scalar function).
- In Assumption 2, the stationary point is denoted by $y^\star$ while later one is referred to as $y^\star(\lambda)$.
- Overall, the equations are very hard to read because the notation is heavy. The paper would benefit from a simplification. For instance, the update matrices that depend on $y$ (resp. $\lambda$) should be called in a way that remind the reader of this dependence. The Jacobian is called $A$ and its estimate $u$.

2. It is not clear which criteria are used for determining if a set of parameters should be optimized in the outer or inner loop of the bi-level problem. For instance, why is the label weight vector of PDBO-DA considered a hyper-parameter, hence to be optimized in the outer problem, and not the parameters of the neural network? In the end, all the parameters of the model are specialized to the agent.

3. Little intuition or explanation are provided for supporting the various derivation choices. For instance, why are truncated Neumann series and recurrent back-propagation needed to compute the outer gradient?

Finally, the interplay between agent similarity (usually referred to as mixing matrix) and model performance is never discussed in the paper. No assumptions are then made on the model for drawing directed communication edges, as the paper does not try (or guarantee) to recover the true mixing matrix. However, the performance of a personalized model depends on how often/well an agent communicates with similar agents. If existing works make more assumptions on the graph model, it is to guarantee a good estimation of the true mixing matrix, hence a good estimation of true local models.



**Summary Of The Paper:**

The paper proposes a general formulation for learning personalized models in a fully-decentralized context. The problem is formulated as a bi-level optimization problem, where the outer-objective updates the local hyper-parameters given the optimal local parameters of the inner objective. The algorithm for optimizing such problem in a decentralized way is then derived, supposing that the communication graph is fully connected but only a few directed communication edges are active at each iteration.

**Summary Of The Review:**

The paper would require a major revision in order for the reader to appreciate its contributions. At this stage, the significance and potential impact of the work are limited because of the poor clarity of the paper and because of the lack of discussion or guarantees on the recovery of the true mixing matrix depending on the choice of random communciation model.

---

> ### Author Response · Authors · 2022-11-19
> **The initial response to Reviewer 3KYv (1/3)**
>
> ## Updates of the paper
>
> We thank the reviewer for the constructive comments. Motivated by your comments and questions, we improved the paper by,
>
> - **Improved clarity** **of derivation** in Section 2 by adding detailed citations and intuitions behind the derivations
> - Adding **discussion on the applicability of other hyper-gradient estimation methods** on stochastic directed networks in Related work and Appendix H
> - Adding **pseudo algorithms** of PDBO, HGP, and VR-HGP in Appendix F
>
> We also made the following updates according to the comments by other reviews:
>
> - Replacing the **assumption of  $\boldsymbol{y}\_i^∗=\boldsymbol{y}\_i^{(T) }$** with assumptions on **loss convexity** and **network connectivity** (Section 3 and 4)
> - Adding theoretical analysis of the **estimation error incurred by $\Vert \boldsymbol{y}\_i^∗ - \boldsymbol{y}\_i^{(T) }\Vert$**  in Appendix C.7
> - Adding **Extended experimental results** **on CIFAR10, CIFAR100,** and **Shakespeare** in Appendix G
>
> ---
>
> ## Responses
>
> For the quick response, below, we picked and respond to the major points in your comments. We will respond to the following comment shortly after this post.
>
> - *“It is not clear which criteria are used for determining if a set of parameters should be optimized in the outer or inner loop of the bi-level problem.”* (in Weakness, reason 2)
>
> ### Lack of discussion on the recovery of the true mixing matrix (in Weaknesses)
>
> > *Finally, the interplay between **agent similarity (usually referred to as mixing matrix) and model performance is never discussed in the paper.** No assumptions are then made on the model for drawing directed communication edges, as the paper does not try (or guarantee) to recover the true mixing matrix. However, the **performance of a personalized model depends on how often/well an agent communicates with similar agents.** If existing works make more assumptions on the graph model, it is to guarantee a good estimation of the true mixing matrix, hence a good estimation of true local models.*
> >
>
> **We respectfully disagree with this weakness**, as **there is** **no existing study that attempts to recover a mixing matrix that represents both similarity of clients and the communication frequency**, to the best of our knowledge.
>
> **Without any citations, it is difficult to interpret what our paper misses.**
>
> What makes things confusing is, matrices of size $n\times n$ appear in different contexts of distributed learning. As we suppose the raised weakness could be raised based on some confusion of the following different $n\times n$ matrices, let us clarify what each matrix represents and does ***not*** represent.
>
> 1. **Weighted adjacency matrix** (**Ours**)
>
>     What it represents: **Communication availability or frequency**
>
>     What it does **not** represent: **Client similarity**
>
>     Weighted adjacency matrix ($\in [0,1]^{n\times n}$) represents the physical connectivity and weights multiplied by the exchanged parameters, which we and common existing works [Bianchi, et al. (2013), Nedic, et al. (2014)] use. Note that regardless of the choice of weight $p\_{ij}$ and frequency of communication (i.e., frequency of $p\_{ij}$ to be positive) the clients’ parameters converge to the identical (consensus) parameter, under the common assumptions that the graph is connected at a regular time interval and the matrix is doubly [Bianchi, et al. (2013)] or column stochastic [Nedic, et al. (2014)].
>
>     Therefore, the weighted adjacency matrix represents no similarity between $i$ and $j$, and the model parameters are intended to converge to the global (non-personalized) parameter.

---

> > ### Author Response · Authors · 2022-11-19
> > **The initial response to Reviewer 3KYv (2/3)**
> >
> >
> >
> > 2. **Task relationship matrix**
> >
> >     What it represents: **Client similarity**
> >
> >     What it does **not** represent: **Communication availability and frequency**
> >
> >     Task relationship matrix  ($\in [0,1]^{n\times n}$) [Smith, et al. (2017) and Vanhaesebrouck, et al. (2017)] can represent task similarity and is intended to provide good personalized models. However, this matrix does not represent the connectivity between clients. The most similar counterparts to our work in a fully-decentralized setting would be Vanhaesebrouck, et al. (2017) and Zantedeschi, et al. (2020). However, neither of them recovers the personalized model by how clients communicate. Vanhaesebrouck, et al. (2017) use an $n \times n$ matrix that represents a “*weighted connected graph”.* which represents client similarity. However, each element in the matrix is a regularization coefficient of a norm between parameters of two clients, thus it does not represents any communications among clients. Zantedeschi, et al. (2020) use a ”*collaboration graph”* which is expected to express client similarity. However their collaboration graph is used only for communication efficiency, i.e., similar clients in the collaboration graph communicate more often than the others. Their personalization is based on the optimization of a weighted combination of base predictors, similar to our MTL-PDBO.
> >
> >     In summary, the task relationship matrix is used either to provide regularizations of parameter norms for personalized models or to reduce communication complexities.
> >
> > 3. **Cluster graph topology**
> >
> >     What it represents: **Client similarity and (*possibly*) communication availability**
> >
> >     What it does **not** represent: **Communication frequency**
> >
> >     Clustering clients is a typical personalization approach in a centralized setting. The cluster of clients can be (mathematically) interpreted as a task to find the best graph topology ($\in \{0,1\}^{n\times n}$) consists of cluster subgraphs in each of which clients are similar in data distributions. The true cluster is normally estimated by some assumptions as you mentioned; for instance, Sattler, et, al. (2020) build clusters based on the similarity in gradients. Since clients communicate (commonly via a host server) only with other clients in the same cluster, we can see cluster graph represents the client similarity and implies physical connectivity. **However, the personalization of models is still independent of *“how often/well”* clients communicate.**
> >
> >     We also emphasize that the estimation of the true cluster heavily relies on the existence of the central host server. In our fully-decentralized setting, in which a client has little information about the other clients, we could not find any article that finds a good cluster graph by optimizing communication graph topologies. Such a method is, at least, not a common approach that needs to be discussed. In addition, we suppose this will be especially difficult when communication topology is directed.
> >
> >
> > ### Little intuition or explanation for supporting the derivation choices (in Weaknesses, reason 3)
> >
> > > *Little intuition or explanation are provided for supporting the various derivation choices. For instance, why are truncated Neumann series and recurrent back-propagation needed to compute the outer gradient?*
> > >
> >
> > For ease of understanding derivations, **we added detailed citations** in Section 4 specifying the corresponding part on which our derivation of the estimator is based. **We also simplified several derivations in Section 4 with the help of cited existing works**.
> >
> > To support the choice of hyper-gradient estimation approach from the existing variants, we explained the reason for our choice of recurrent backpropagation in Related work. In related work, we pointed out the other common approaches are infeasible or inefficient in our stochastic and directed communication setting.

---

> > > ### Author Response · Authors · 2022-11-19
> > > **The initial response to Reviewer 3KYv (3/3)**
> > >
> > >
> > > ### *“Some notation and formulations are inexact, confusing or undefined”* (in Weakness, reason 1)
> > >
> > > We thank the corrections and comments on the formulations. They helped us to make our paper more concise and clear. Let us respond to each raised point.
> > >
> > > > *In Problem 1 and Problem 5, the index i is used for two different thing: first for indicating the client's parameters to be optimized and then for indexing the terms of the sum.*
> > > >
> > >
> > > We accordingly changed the index notation in problem formulations.
> > >
> > > > *Also the size of $ξ\_i$ is not provided and $p(ξ\_i)$ is not formally defined.*
> > > >
> > >
> > > We omitted the notation of $p(ξ\_i)$ as it can be confused with weight function $p\_{ij}$
> > >
> > > > *Right before Equation 2, it is said that "The  $i$-th client in SGP updates its weight $ω\_i$" but Equation 4 do not update this parameter.*
> > > >
> > >
> > > It does update. Eq. (2) with Eq. (4) updates $ω\_i$ by $\omega\_i \leftarrow \sum\_{j\in[n]}p\_{ji}(\delta \_j^{(t)})\omega\_j$.
> > >
> > > > *Equation 4, the operation $[a\quad b]$  is not defined. I assumed it was a vector concatenation.*
> > > >
> > >
> > > We added explanations of  $[a~~  b]$ before Eq. (2) and in other places where it is used.
> > >
> > > > *Also $p()$ denotes here the message distribution, while it was used before for the local data distribution.*
> > > >
> > >
> > > We omitted the notation of $p(ξ\_i)$ as it can be confused with $p\_{ij}(\cdot)$. For a recap, we do not introduce *message distribution* in Section 2. $p\_{ij}(\cdot)$ is introduced to denote the scalar function whose output is multiplied by the inner-parameter.
> > >
> > > > *(minor) In page 3, end of Section 2,  $p\_{ij }$ should be replaced by $p()$ ($p\_{ij}$ cannot be a doubly stochastic matrix as it is defined as a scalar function).*
> > > >
> > >
> > > Use of $p\_{ij}$ is intended. For clarity, we added an explanation about the relation between $p\_{ij}$ and the mixing matrix after Eq. (3) and the end of Section 2.
> > >
> > > > *In Assumption 2, the stationary point is denoted by $y^⋆$ while later one is referred to as  $y^⋆(λ)$.*
> > > >
> > >
> > > In the updated paper, we let $(\boldsymbol{\lambda})$ follow only after concatenated parameters $\boldsymbol{y}^*$, to trade-off between the concise notation and clarity of parameter dependence.
> > >
> > > > *Overall, the equations are very hard to read because the notation is heavy. The paper would benefit from a simplification. For instance, the update matrices that depend on $y$ (resp. $λ$) should be called in a way that remind the reader of this dependence. The Jacobian is called $A$ and its estimate $u$.*
> > > >
> > >
> > > Thank you for your suggestion. We have improved the notation in Section 2 so that readers recall the dependencies between $\boldsymbol{x}$, $\boldsymbol{y}$, and $\boldsymbol{\lambda}$. For the other notations, such as $\boldsymbol{A}$ and $\boldsymbol{u}$, we tried to clarify the dependence on parameters such as $\boldsymbol{y}$ or $\boldsymbol{\lambda}$, but found it makes the notation more complicated. As the typical interest of the proposed method is the communication of such values, we kept these notations as they were.
> > >
> > > ### “*The code for reproducing the results was not submitted and in the empirical section the chosen model architecture is not described.”* (in Reproducibility)
> > >
> > > For reproducibility, we uploaded the code with the updated paper for reproducing the results, including the additional experiments.
> > >
> > > ---
> > >
> > > ### References
> > >
> > > - [Bianchi, et al. (2013)] Bianchi, Pascal, Gersende Fort, and Walid Hachem. "Performance of a distributed stochastic approximation algorithm." IEEE Transactions on Information Theory 59.11 (2013): 7405-7418.
> > > - [Nedic, et al. (2014)] Nedić, A., & Olshevsky, A. (2014). Distributed optimization over time-varying directed graphs. IEEE Transactions on Automatic Control, 60(3), 601-615.
> > > - [Smith, et al. (2017)] Smith, Virginia, et al. "Federated multi-task learning." Advances in neural information processing systems 30 (2017).
> > > - [Vanhaesebrouck, et al. (2017)] Vanhaesebrouck, Paul, Aurélien Bellet, and Marc Tommasi. "Decentralized collaborative learning of personalized models over networks." Artificial Intelligence and Statistics. PMLR, 2017.
> > > - [Zantedeschi, et al. (2020)] Zantedeschi, Valentina, Aurélien Bellet, and Marc Tommasi. "Fully decentralized joint learning of personalized models and collaboration graphs." International Conference on Artificial Intelligence and Statistics. PMLR, 2020.
> > > - [Sattler, et, al. (2020)] Sattler, Felix, Klaus-Robert Müller, and Wojciech Samek. "Clustered federated learning: Model-agnostic distributed multitask optimization under privacy constraints." *IEEE transactions on neural networks and learning systems* 32.8 (2020): 3710-3722.

---

> > > > ### Author Response · Authors · 2022-11-20
> > > > **The second response to cover all the points**
> > > >
> > > > ## “*It is not clear which criteria are used for determining if a set of parameters should be optimized in the outer or inner loop of the bi-level problem.”* (in Weakness, reason 2)
> > > >
> > > > **First, if a set of parameters are chosen by a developer's intention, the criterion is clear from the problem formulation of PDBO:**
> > > >
> > > > - **Inner-parameter: a parameter that is shared among clients (i.e., consensus) for better generalization**
> > > > - **Outer-parameter: a parameter that should vary among clients for better personalization performance**
> > > >
> > > > **Second, how the set of parameters should be chosen is not so clear, but this is not our inherent weakness but rather a general problem of personalization. However, our approach is easier than the others because if developers are not sure which parameter should be chosen, they can simply optimize all of them and let PDBO decide which helps the personalization,** without specifying a single set of parameters as existing methods need.
> > > >
> > > > The **first** point can answer the following questions.
> > > >
> > > > > *For instance, why is the label weight vector of PDBO-DA considered a hyper-parameter, hence to be optimized in the outer problem, and not the parameters of the neural network?*
> > > > >
> > > >
> > > > This is because our choice of label weight vector and neural network parameters intends to mitigate label distribution heterogeneity to obtain well-generalized neural network parameters.
> > > >
> > > > Label weight vector thus should be trained for each client as the outer-parameter, because we intended to mitigate the label distribution heterogeneity. Similarly, the neural network parameters should be the inner-parameter since we intended to obtain a well-generalized model with the help of consensus, as existing FedAvg and SGP do.
> > > >
> > > > Note that if a developer intends to train personalized classification layers and the well-generalized consensus feature extractor, the neural network parameters (i.e., classification layer) can be the outer-parameter as described in Section 3 after Eq. (5).
> > > >
> > > > > *In the end, all the parameters of the model are specialized to the agent.*
> > > > >
> > > >
> > > > We interpret it says assigning all the parameters to out-parameter $\boldsymbol{\lambda}_i$ with an empty inner-parameter. This case will be equivalent to letting clients simply train their model parameters locally using gradient methods (e.g., SGD or Adam).
> > > >
> > > > As you remarked in the second question, the choice of the inner-parameter and outer-parameter is arbitral, and it is unclear which parameters should be chosen and assigned to the inner- or outer-parameter without any intention of a developer. The **second** point should answer this unclarity; it is a general and fundamental problem of any personalization. **Existing work is also not free from this arbitrability and unclarity**, **because there is no single criterion that tells which personalization method should be chosen** **from various approaches** (e.g., model-ensemble personalization or over-/under-sampling) for a specific personalization task. Thus, developers need to choose a single approach based on their assumption or intention for the task (e.g., Sattler, et, al. (2020) requires the developer to assume that any client belongs to clusters and to assume the number of clusters). **The advantage of our approach is that developers can choose multiple possible assumptions and let PDBO decide which helps the personalization**. Our PDBO-DA&MTL with label weight vector and ensemble weight of 3 CNNs is an instance of choosing two assumptions: label distribution is heterogeneous and a client data distribution is a mixture distribution of 3 base distributions [Marfoq, et al. (2021)]. This is much easier than choosing a single personalization method with a single assumption.
> > > >
> > > > ---
> > > >
> > > > ### References
> > > >
> > > > - [Marfoq, et al. (2021)] Marfoq, Othmane, et al. "Federated multi-task learning under a mixture of distributions." *Advances in Neural Information Processing Systems* 34 (2021): 15434-15447.
> > > > - [Sattler, et, al. (2020)] Sattler, Felix, Klaus-Robert Müller, and Wojciech Samek. "Clustered federated learning: Model-agnostic distributed multitask optimization under privacy constraints." *IEEE transactions on neural networks and learning systems* 32.8 (2020): 3710-3722.

---

### Official Review · Reviewer_Mrqc · 2022-10-24

**Confidence:** 3
**Correctness:** 3
**Technical Novelty And Significance:** 2
**Empirical Novelty And Significance:** Not applicable
**Recommendation:** 5

**Clarity, Quality, Novelty And Reproducibility:**

- paper is hard to follow. the authors better to provide some intuition or explanation for some derivations

- unable to verify. code is not provided but the authors claimed in the paper to release code after review process

**Strength And Weaknesses:**

Strength:
- This study proposed a gradient-based PDBO, which reduces most personalization approaches to the optimization of hyperparameters possessed by each client.

Weaknesses:
- Assumption 2 is a bit too strong to require $y_i^*=y_i^{(T)}$ make it unpractical. The authors should provide some bound estimation for it.
- Empirical results using the settings of EMNIST classification is a bit too simple.


**Summary Of The Paper:**

The authors propose a gradient-based bilevel optimization that reduces most personalization approaches to the optimization of client-wise hyperparameters. Then the authors propose a decentralized algorithm to estimate gradients with respect to the hyperparameters, which can run even on stochastic and directed communication networks. Empirical results demonstrated that the gradient-based bilevel optimization enabled combining existing personalization approaches which led to state-of-the-art performance, confirming it can perform on multiple simulated communication environments including a stochastic and directed network.

**Summary Of The Review:**

This study proposed a gradient-based PDBO, which reduces most personalization approaches to the optimization of hyperparameters possessed by each client. The approach is not practical since the assumption is too strong, and the experiment using EMNIST classification is a bit too simple.

---

> ### Author Response · Authors · 2022-11-19
> **Response to reviewer Mrqc (1/2)**
>
>
> ## Updates of the paper
>
> We thank the reviewer for the constructive comments. Motivated by your comments and questions, we improved the paper by,
>
> - Replacing the **assumption of  $\boldsymbol{y}\_i^∗=\boldsymbol{y}\_i^{(T) }$** with assumptions on **loss convexity** and **network connectivity** (Section 3 and 4)
> - Adding Theoretical analysis of the **estimation error incurred by $\Vert \boldsymbol{y}\_i^∗ - \boldsymbol{y}\_i^{(T) }\Vert$**  in Appendix C.7
> - Adding **Extended experimental results** **on CIFAR10, CIFAR100,** and **Shakespeare** in Appendix G
> - **Improving the clarity** of derivation in Section 2 by adding detailed citations and intuitions behind the derivations
> - Adding **pseudo algorithms** of PDBO, HGP, and VR-HGP in Appendix F
>
> We also made the following updates according to the comments by other reviews:
>
> - Adding **Discussion on the applicability of other hyper-gradient estimation methods** on stochastic directed networks in Related work and Appendix H
>
> ---
>
> ## Responses
>
> ### [Weakness 1] “*Assumption 2 is a bit too strong to require* $\boldsymbol{y}\_i^∗=\boldsymbol{y}\_i^{(T)}$ *make it unpractical.”*
>
> > *Assumption 2 is a bit too strong to require $y\_i^∗=y\_i^{(T)}$ make it unpractical. The authors should provide some bound estimation for it.*
> >
>
> **We relaxed the assumption of $\boldsymbol{y}\_i^∗ = \boldsymbol{y}\_i^{(T) }$ by,**
>
> 1. **Replacing the assumption**  $\boldsymbol{y}\_i^∗=\boldsymbol{y}\_i^{(T) }$ **with weaker assumptions** of loss convexity and network connectivity (new Assumption 1 and 2)
> 2. Reformulate the PDBO by the stationary point of the SGP iteration (new Eq. (7))
> 3. **Providing the bound of an estimation bias incurred by $\Vert \boldsymbol{y}\_i^∗ - \boldsymbol{y}\_i^{(T) }\Vert$** (Theorem 11 in Appendix C.7).
>
> Regarding **(1.)** and **(2.)**, motivated by your comment, we replaced the inner-problem of reformulated PDBO with the stationary condition of  $\boldsymbol{y}\_i^∗$ (new Eq. (7)) following Grazzi et al., 2020. Then reformulating PDBO (new Eq. (7)) is justified from the mathematical equivalencies between the stationary point of SGP (Eq. (2)) and the optimum of distributed learning in Eq. (1). **We ensure this equivalency by introducing assumptions weaker than**  $\boldsymbol{y}\_i^∗=\boldsymbol{y}\_i^{(T) }$: every client cost $f\_i$ is strongly-convex (new Assumption 1) and the stochastic network is strongly connected in expectation (new Assumption 2). The new Assumptions 1 and 2 are adopted in the original papers of SGP [Nedić, et al. (2014)] and the push-sum [Bénézit, et al (2010)], respectively.
>
> Besides the replacement of formulation and assumptions **(1.** and **2.)**, $\boldsymbol{y}\_i^{*}$ is practically approximated by  $\boldsymbol{y}\_i^{(T)}$ (the consequence of $T$ iterations of SGP). We thus provided estimation error bound of hyper-gradient considering the error $\Vert \boldsymbol{y}\_i^∗ - \boldsymbol{y}\_i^{(T) }\Vert$ in **(3.)** (Theorem 11 in Appendix C.7), **which results in more general and practical bound than Theorem 1**. Theorem 11 indicates that the error in inner-parameter $\Vert \boldsymbol{y}\_i^∗ - \boldsymbol{y}\_i^{(T) }\Vert$ brings estimation bias which linearly decreases as $\Vert \boldsymbol{y}\_i^∗ - \boldsymbol{y}\_i^{(T) }\Vert$ decreases.
>
> We also note that **our experiments demonstrated the practicality** of our approach. **Our experiment settings in Section 6 and Appendix G provide neither convergence nor convexity**, as we used deep learning models trained by a finite number of SGP iterations. Even in these difficult settings, our approach succeeded in improving the personalization performance.
>
> ### [Weakness 2] “*Empirical results using the settings of EMNIST classification is a bit too simple.”*
>
> > *Empirical results using the settings of EMNIST classification is a bit too simple.*
> >
>
> We conducted additional experiments of personalization on **natural image classifications (CIFAR10 and CIFAR100)** with Mobilenetv2 and **next character prediction on Shakespeare dataset trained by Stacked-LSTM**.
>
> We observed the improvements in our approaches from baselines in all additional tasks.
>
> ### [Reproducibility] Code is not provided
>
> > *unable to verify. code is not provided but the authors claimed in the paper to release code after review process*
> >
>
> We uploaded the code with the updated paper for reproducing the results, including the additional experiments.
>
> ### [Clarity, Quality, and Novelty] “*paper is hard to follow.”*
>
> > *paper is hard to follow. the authors better to provide some intuition or explanation for some derivations*
> >
>
> For ease of understanding derivations, **we added detailed citations** in Section 4 specifying the corresponding part on which our derivation of the estimator is based. **We also simplified several derivations in Section 4 with the help of cited existing works**.

---

> > ### Author Response · Authors · 2022-11-19
> > **Response to reviewer Mrqc (2/2)**
> >
> >
> > ### References
> >
> > - [Grazzi et al. (2020)] R. Grazzi, L. Franceschi, M. Pontil, and S. Salzo. On the iteration complexity of hypergradient computation. In International Conference on Machine Learning, pages 3748–3758. PMLR, 2020.
> > - [Nedić, et al. (2014)] Nedić, A., & Olshevsky, A. (2014). Distributed optimization over time-varying directed graphs. IEEE Transactions on Automatic Control, 60(3), 601-615.
> > - [Bénézit, et al (2010)] Bénézit, Florence, et al. "Weighted gossip: Distributed averaging using non-doubly stochastic matrices." 2010 ieee international symposium on information theory. IEEE, 2010.

---

### Official Review · Reviewer_b9iT · 2022-10-25

**Confidence:** 3
**Correctness:** 3
**Technical Novelty And Significance:** 3
**Empirical Novelty And Significance:** 2
**Recommendation:** 5

**Clarity, Quality, Novelty And Reproducibility:**

This paper is well-written in general. The comparison with related work on decentralized bilevel optimization is clear, though the relevance to previous hyper-gradient estimation methods and push-sum methods need to be clarified in more detail. In addition, several assumptions in theoretical analysis need to be further justified (please refer to the above **Weaknesses**).

**Strength And Weaknesses:**

**Strengths:**
+ The discovery of the failure of standard backpropagation for hyper-gradient computation over directed networks is interesting, and the use of push-sum strategy sounds novel.
+ This paper is well-written in general and easy to follow.
+ The derivation of the estimation error bound is technically sound.

**Weaknesses:**
- Some assumptions in this paper need to be justified.
I wonder if the assumption of knowing the receiving frequency in advance is reasonable. Although this paper said that it can be estimated by simulating several rounds, it will bring estimation error to the computation of hyper-gradients. Does the main theorem take such an error into consideration?
In addition, please justify Assumption 5, which said that the edges are “undirected” in expectation. I wonder if this assumption still holds for standard (deterministic) directed networks.
- The relevance to some related work should be clarified.
Since the proposed method relies on the previously proposed recurrent backpropagation method, HPG should be compared with Grazzi’s work in detail, especially from the theoretical aspect. Specifically, what is the difference between using the sending edges and the receiving edges in principle? Will it bring any difficulty in theoretical analysis? Besides the error bound for VR-HGP, can you provide a bound for the standard HGP and compare it with Grazzi’s?
Another related line of work is the push-sum method. It seems that the core innovation of HPG is to introduce such a method to the optimization of the outer-problem. Hence, I wonder what is the main difference from applying the method to the inner-problem as in previous works (perhaps more complex theoretical analysis)?


**Summary Of The Paper:**

This paper studies the problem of personalized bilevel optimization in a decentralized distributed learning system over stochastic and directed underlying communication networks. The major contribution of this paper is to propose an effective scheme termed HGP to compute the hyper-gradient, i.e., the gradient of hyperparameters in the outer-problem, in a decentralized way. Specifically, this paper first shows the failure of standard backpropagation in hyper-gradient computation on a directed network, then introduces a push-sum communication strategy to tackle the biasness of the hyper-gradient estimate. Moreover, the variance reduction technique is introduced to handle the large variance in the push-sum communication. This paper further derives an upper bound for the estimation error of the proposed method and evaluates it effectiveness empirically.

**Summary Of The Review:**

This paper contributes an effective scheme for hyper-gradient computation in PDBO over stochastic and directed networks, where the use of push-sum communication to make an unbiased estimate is interesting. However, the relevance to previous hyper-gradient computation methods and push-sum communication methods should be explained in more detail, and several assumptions in theoretical analysis need to be justified.

---

> ### Author Response · Authors · 2022-11-19
> **The initial response to Reviewer b9iT**
>
> ## Updates of the paper
>
> We thank the reviewer for the constructive comments. We added and improved the following parts according to the suggestions from the reviewers.
>
> - Adding **Extended experimental results** **on CIFAR10, CIFAR100,** and **Shakespeare** in Appendix G
> - Adding theoretical analysis of the **estimation error incurred by $\Vert \boldsymbol{y}\_i^∗ - \boldsymbol{y}\_i^{(T) }\Vert$**  in Appendix C.7
> - Replacing the **assumption of  $\boldsymbol{y}\_i^∗=\boldsymbol{y}\_i^{(T) }$** with assumptions on **loss convexity** and **network connectivity** (Section 3 and 4)
> - Adding **pseudo algorithms of PDBO, HGP, and VR-HGP** in Appendix F
> - Adding **Discussion on the applicability of other hyper-gradient estimation methods** on stochastic directed networks in Related work and Appendix H
>
> ---
>
> ## Responses
>
> For the quick response, below, we picked and respond to important parts of your comments. We will respond to the following comment shortly after this post.
>
> - Weakness 2: *“The relevance to some related work should be clarified”*
>
> ### [Weakness 1-a] Justification of the assumption that clients know receiving frequency in advance
>
> > *Some assumptions in this paper need to be justified. **I wonder if the assumption of knowing the receiving frequency in advance is reasonable**.*
> >
>
> **It is reasonable because receiving frequency can be estimated accurately in practice**; **the estimation error can be disregarded as it diminishes at $O(\frac{1}{\sqrt{T}})$ and the number of SGP iterations $T$ needs to be large for the convergence of the inner-optimization.**
>
> The assumption of edge independence allows edge realization to simply follow the Bernoulli distribution having the estimation error diminishing at $O(\frac{1}{\sqrt{T}})$. This estimation error is disregarded in practice since $T$ is typically large in modern distributed learning for deep models; the number of inner iterations of FedAvg or SGP has to be large enough for convergence and better performance. We also note that we used the estimated value of receiving frequency in our experiments, and under the setting of $T=600$, our personalization seemed not to be harmed by this estimation error. In the updated paper, we explicitly described that every client in our experiments used the receive frequency that it estimated through the inner-iterations.
>
> > *Although this paper said that it can be estimated by simulating several rounds, it will bring estimation error to the computation of hyper-gradients. **Does the main theorem take such an error into consideration?***
> >
>
> Theorem 1 does not take the estimation error of receiving frequency into account. We agree that adding the estimation error of the receiving frequency certainly makes Theorem 1 more general. However, from the experimental observations that such an error has only negligible impacts on the result, we decided to omit it from Theorem 1 because it is not essential, and decided to focus on the essential factor, i.e., the effect of variance reduction.
>
> ### [Weakness 1-b]  Justification of Assumption 5 (*“undirected” in expectation*)
>
> > *In addition, **please justify Assumption 5**, which said that the edges are “undirected” in expectation.*
> >
>
> **The assumption of *“undirected in expectation”* is commonly adopted by previous studies for its practicality.**
>
> As the stochastic directed edge is a mathematical model of the random gossip protocol [Bénézit, et al. (2010)], if the edge from $i$ to $j$ is realized, it is practical to suppose there is a physical connection between $i$ and $j$. Thus another direction of  $j$ to $i$ is expected to realize in the long time span. Several studies of push-sum methods [Bénézit, et al. (2010), Rezaienia, et al. (2019)] also assume undirected in expectation.
>
> > *I wonder if this assumption still holds for standard (deterministic) directed networks.*
> >
>
> Deterministic directed networks are not cases of Assumption 5. However, it rarely appears in practice for the reason explained above.
>
> ---
>
> ### References
>
> - Bénézit, Florence, et al. "Weighted gossip: Distributed averaging using non-doubly stochastic matrices." 2010 ieee international symposium on information theory. IEEE, 2010.
> - Rezaienia, Pouya, et al. "Push-sum on random graphs: almost sure convergence and convergence rate." *IEEE Transactions on Automatic Control*  65.3 (2019): 1295-1302.

---

> > ### Author Response · Authors · 2022-11-20
> > **The second response to cover all the points (1/2)**
> >
> > ### [Weakness 2-a] Relevance to previous hyper-gradient computation methods
> >
> > > *Since the proposed method relies on the previously proposed recurrent backpropagation method, **HGP should be compared with Grazzi’s work in detail, especially from the theoretical aspect**.*
> > >
> >
> > Thank you for shedding light on the theoretical aspect of HGP.
> >
> > In this paper, we did not include the comparison of the theoretical difference between Grazzi’s work because **the recurrent backpropagation is impossible to compute in a decentralized manner on the stochastic directed network**, as explained in Section 4.2. **While one can recover Grazzi’s work, which considers deterministic iterations, as a special case of HGP by restricting the problem into static undirected communication and deterministic losses, this is out of our main scope of stochastic directed communication**. We thus prioritized tackling the problem of bounding estimation variance incurred by stochastic directed communication, which has never been discussed in the previous hyper-gradient computation methods.
> >
> > Instead of the theoretical aspects, we added discussions of the applicability of existing methods from **practical aspects** in Related work and Appendix H. In related work, we point out the other common approaches are infeasible or inefficient in our stochastic and directed communication setting.
> >
> > > ***Specifically, what is the difference between using the sending edges and the receiving edges in principle? Will it bring any difficulty in theoretical analysis?***
> > >
> >
> > The difference between using sending edges (Exact recurrent backpropagation) and receiving edges (HGP) is **their possibility of decentralized computation in principle**; recurrent backpropagation using sending edges is impossible on the stochastic directed network (Section 4.2).
> >
> > Regarding the difficulty in theoretical analysis, **the estimation error bound of HGP and VR-HGP is not a trivial extension** **of existing methods** because HGP and VR-HGP require us to consider how replacing the sending edge with receiving edge brings the variance in estimation.
> >
> > > *Besides the error bound for VR-HGP, can you provide a bound for the standard HGP and compare it with Grazzi’s?*
> > >
> >
> > The error bound of the standard HGP can be found in Theorem 10 in Appendix C.4. Regarding the comparison of the error between HGP and Grazzi’s work, we prioritized putting emphasis on the variance rather than comparing the bound of HGP with that of an impossible algorithm (Grazzi’s recurrent backpropagation) as explained above. For the comparison with other hyper-gradient estimation th
> >
> > ### [Weakness 2-b] Relevance to previous push-sum methods
> >
> > > *Another related line of work is the* ***push-sum method***. *It seems that* ***the core innovation of HGP is to introduce such a method to the optimization of the outer-problem***. *Hence, I wonder* ***what is the main difference from applying the method to the inner-problem as in previous works*** *(perhaps more complex theoretical analysis)?*
> > >
> >
> > **It is difficult to answer since the premise of the question seems to be incorrect.**
> >
> > Let us summarize the problem setting of our PDBO below. To clarify the point of your question, it will be appropriate to think about the layers PDBO’s problem setting encompasses; PDBO’s aim is to solve both the inner- **(2)** and outer-optimization **(3)** on the stochastic and directed networks **(1)**.
> >
> > 1. **Communication layer: Stochastic and directed edges**
> > 2. **Inner-optimization layer: SGP (the push-sum method) [Nedic, et al. (2016)]**
> > 3. **Outer-optimization layer: Gradient method (e.g., SGD or Adam)**
> >
> > > *It seems that **the core innovation of HGP is to introduce such a method*** **(push-sum)** ***to the optimization of the outer-problem**.*
> > >
> >
> > As in **(2)**, **the push-sum method (SGP) was adopted to pose our inner-problem,** as SGP addresses the distributed learning (inner-optimization) on stochastic directed settings **(1)**. Hence in **(3)**, we focus on solving outer-optimization by introducing a hyper-gradient estimation method that runs on stochastic directed settings **(1)** and that enables a gradient method for outer-optimization. Finally, we proposed hyper-gradient estimation (HGP) by replacing the sending and receiving edges and corrected its estimation bias by receiving frequency. **We thus do not use the push-sum algorithm for outer-optimization.**

---

> > > ### Author Response · Authors · 2022-11-20
> > > **The second response to cover all the points (2/2)**
> > >
> > > > *Hence, I wonder **what is the main difference from applying the method to the inner-problem as in previous works** (perhaps more complex theoretical analysis)?*
> > > >
> > >
> > > In our view above, your question can be interpreted as either of the following. However, we suspect none of them is your intention.
> > >
> > > - *What is the main difference from applying* **“the push-sum method”** *to the inner-problem?*
> > >
> > >     → That is SGP.
> > >
> > > - *What is the main difference from applying* **“the hyper-gradient estimation method”** *to the inner-problem?*
> > >
> > >     → Hyper-gradient is only relevant in bilevel optimization.
> > >
> > > - *What is the main difference from applying* **“the gradient method”** *to the inner-problem?*
> > >
> > >     → It just recovers the local training of clients, bringing a more simple theoretical analysis.
> > >
> > >
> > > We hope to continue the discussion after agreeing on the premise of your question.
> > >
> > > ---
> > > ### References
> > > - [Nedic, et al. (2016)] Nedić, Angelia, and Alex Olshevsky. "Stochastic gradient-push for strongly convex functions on time-varying directed graphs." IEEE Transactions on Automatic Control 61.12 (2016): 3936-3947.
> > > - [Grazzi] R. Grazzi, L. Franceschi, M. Pontil, and S. Salzo. On the iteration complexity of hypergradient computation. In International Conference on Machine Learning, pages 3748–3758. PMLR, 2020.

---

### Author Response · Authors · 2022-11-29
**Dear Reviewers**

We would like to appreciate your time and great efforts in reviewing our paper and providing constructive feedback. They helped us to improve the clarity of the paper.

We finished the initial responses at the beginning of the 2nd discussion stage.

**As the discussion period ends soon, we are happy if you can express any comments on our responses to continue discussions.**

In particular, we are curious about the discussions on “Relevance to previous push-sum methods” (from Reviewer b9iT) and “Recovering the true mixing matrix” (from Reviewer 3KYv) as their premise of the questions are needed to be clarified to address your concerns.

Kind regards,

Paper2391 Authors

---

### Decision · Program_Chairs · 2023-01-20

**Decision:**

Reject

**Justification For Why Not Higher Score:**

Two of the reviewers were in favor of rejecting the paper.

**Justification For Why Not Lower Score:**

-

**Metareview: Summary, Strengths And Weaknesses:**

This paper studies the problem of Personalized Decentralized Bilevel Optimization (PDBO). The authors first introduce a gradient-based
bilevel optimization that reduces most personalization approaches to the optimization of client-wise hyperparameters.  Based on Stochastic Gradient Push (SGP) method, the authors propose Hyper-Gradient Push (HGP) method to estimate the hypergradient and analyze the estimation error of its variance-reduced version (VR-HGP). Also, experimental results are provided to solve PDBO problems and compare its performance to some existing algorithms.

The reviewers had some initial concerns, many of which were alleviated by the author response, but some still remain (please see the reviews). In particular, the reviewers still felt unconvinced about the basic assumptions as well as how strong is the role of mini-batches. Once these concerns are addressed (in a new version), the paper will provide a set of nice contributions.